# Post-Training with Policy Gradients: Optimality and the Base Model Barrier

**Alireza Mousavi-Hosseini** [1 2]   **Murat A. Erdogdu** [1 2 3]

## Abstract

We study post-training linear autoregressive models with outcome and process rewards. Given a context $x$, the model must predict the response $y \in \mathcal{Y}^N$, a sequence of length $N$ that satisfies a $\gamma$ margin condition, an extension of the standard separability to sequences. We prove that on test samples where the base model achieves a non-trivial likelihood $\alpha$, a variant of policy gradient (PG) can achieve likelihood $1 - \varepsilon$ with an essentially minimax optimal number of reward queries $\tilde{\mathcal{O}}((\alpha^{-1} + \varepsilon^{-1})/\gamma^2)$. However, a barrier arises for going beyond the support of the base model. We prove that the overall expected error after post-training with outcome rewards is governed by a property of the base model called the *Likelihood Quantile* (LQ), and that variants of PG, while minimax optimal, may require a number of reward queries exponential in $N$ to go beyond this support, regardless of the pre-training algorithm. To overcome this barrier, we study post-training with a process reward model, and demonstrate how PG variants in this setting avoid the curse of dimensionality in $N$ via dependence on a token-level LQ. Along the way, we prove that under the margin condition, SGD with adaptive learning rate (LR) achieves a near optimal test error for statistical learning, and PG with adaptive LR achieves a near optimal number of mistakes for online learning while being computationally efficient whenever possible, both of which may be of independent interest.

## 1. Introduction

Outcome-based reinforcement learning (RL) has been a promising approach to enable large language models (LLMs) to go beyond their pre-training data. RL has been particularly successful in domains such as math and coding where one is able to *verify* the correctness of the model's response, without requiring a complete ground-truth solution (Guo et al., 2025; Team et al., 2025). However, the extent to which RL can create new knowledge that is missing from the base model is unclear. While some works argue that RL with outcome rewards can genuinely enhance the base model (Wen et al., 2026), others have observed that the effect of RL post-training has mostly been to sharpen the base distribution, and the fine-tuned model is not able to generate responses that are outside the support of the base model (Yue et al., 2025; Wu et al., 2025; Shao et al., 2025). Indeed, explicitly sampling from the sharpened base distribution can match or outperform RL (Karan & Du, 2026), which suggests that RL cannot successfully improve upon the base model unless it is already highly capable.

On the theoretical side, prior works have attempted to characterize the benefits of RL post-training compared to supervised fine-tuning (SFT) (Setlur et al., 2025b) or on top of pre-training (Tsilivis et al., 2026; Kim et al., 2025). While providing positive results for RL, these works leave out the role of the base model. Foster et al. (2025) demonstrate that a notion of *coverage* is needed for computationally efficient exploration with RL. However, their definition does not depend on the final desired likelihood, the probability that the model generates a correct response to a context. Chen et al. (2025a) study coverage profile which characterizes the probability of success with best-of-N sampling, and consider how log-likelihood maximization improves this coverage. However, the consequences of this definition for RL post-training, which is used to achieve the best-of-N performance using a single generation, is less clear. Chen et al. (2025b) study outcome-based online RL, with a focus on sample complexity, while the computational efficiency of their algorithm is unclear. Importantly, the works above leave open the study of practical algorithms for post-training, specifically PG variants such as proximal policy optimization (PPO) (Schulman et al., 2017) and group relative policy op-

[1]University of Toronto [2]Vector Institute [3]Numurho. Correspondence to: Alireza Mousavi-Hosseini <mousavi@cs.toronto.edu>.

*Proceedings of the 43$^{rd}$ International Conference on Machine Learning*, Seoul, South Korea. PMLR 306, 2026. Copyright 2026 by the author(s).

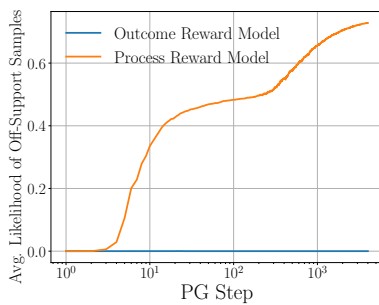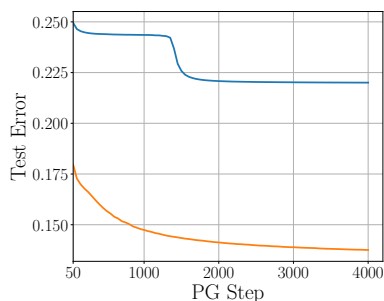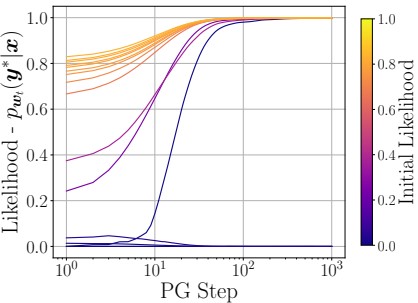

*Figure 1.* The evolution of the model's likelihood to generate the correct response over different contexts throughout PG. **(Left)** The average likelihood over samples with initial likelihood $\approx 0$ under the base model. On-policy PG with PRM is able to improve this average while with ORM stays at 0. **(Center)** The comparison between the expected test error with ORM where the error plateaus at a threshold, and with PRM where the error continues to decrease. **(Right)** The likelihood of individual samples throughout PG with ORM, where the color denotes initial likelihood. Experiment details are presented in Section 6.

timization (GRPO) (Shao et al., 2024). In this work, we aim to answer the following question in a theoretically tractable setting:

**Q1.** *How do the number of reward queries and policy gradient steps in post-training depend on the base model for on- and off-support samples?*

We answer the above question by contrasting the on- and off-support cases. For sequences of length $N$, the probability that the uniform policy generates a correct response is exponentially small in $N$. If the base policy from pre-training achieves a probability of correct response (likelihood) that is non-trivial, i.e. at most polynomially small in $N$, we consider this sample to be on the base model support, and the complexities above remain polynomial. However, if the base policy is not significantly better than a uniform distribution on a sample, then the sample is off-support and improving its likelihood may require exponentially many iterations. This phenomenon is demonstrated in Figure 1. While this gives us a per-sample answer to the above question, a natural next step is to investigate the expected error over the entire distribution:

**Q2.** *Can RL post-training achieve a significantly smaller expected test error compared to the base model while being computationally efficient?*

We answer the above question for outcome and process rewards. For outcome reward models, we show that in the worst-case, any post-training algorithm may require exponentially many reward queries to significantly improve upon the base model obtained by SGD. However, this exponential dependence on the sequence length can be alleviated by using a process reward model, where after generating each token, the learner can query the reward to verify the correctness of the response so far.

**1.1. Our Contributions**

Our main contributions are as follows:

- In Section 3.1, we study post-training with outcome rewards and access to a base model $q$, with the goal of predicting a response $\boldsymbol{y}^*(\boldsymbol{x}) = (y_1^*(\boldsymbol{x}), \ldots, y_N^*(\boldsymbol{x}))$, where $y_i \in \{1, \ldots, k\}$, to an input context $\boldsymbol{x}$, which satisfies a $\gamma$ margin assumption. We show that conditioned on the base model achieving a non-trivial likelihood $\alpha$ on a sample $(\boldsymbol{x}, \boldsymbol{y}^*(\boldsymbol{x}))$, a variant of policy-gradient can boost this likelihood and get expected error $\varepsilon$ with $\tilde{\mathcal{O}}(1/(\alpha\gamma^2\varepsilon))$ iterations. As a consequence, we show that PG with a uniform behavior policy can achieve the minimax optimal mistake bound $\tilde{\mathcal{O}}(k^N/\gamma^2)$ in online learning.

- In Section 3.2, we show that for the overall expected error to go below $\varepsilon$, the number of reward queries depends on a property of the base model, which we refer to as the *Likelihood Quantile* (LQ). This creates a fundamental *barrier of the base model*; if the base model is pre-trained with SGD, PG requires exponentially many reward queries to go below the SGD error rate.

- In Section 4, we demonstrate how using process rewards instead of outcome rewards can alleviate the above problem, with the number of reward queries instead depending linearly on $N$ and another property of the base model, called the *Token-Level Likelihood Quantile*, which is linear in $k$ in the worst-case.

- In Section 5, we establish the minimax optimality of the policy gradient variants explored in Sections 3 and 4. We also rule out the possibility of significant algorithmic improvements beyond SGD in pre-training for better LQ; the minimax optimal LQ is of order $k^{-N}$ unless enough labeled samples are given for an SGD pre-trained model to have very low test error of order $k^{-N}$.

**1.2. Additional Related Works**

**Online Linear Classification with Bandit Feedback.** The problem of post-training a policy with outcome rewards can be cast as an instance of a contextual bandit problem. For the case of $N = 1$, there has been a rich literature

on learning a linear multiclass classifier with bandit feedback. The seminal work of Kakade et al. (2008) introduced the Banditron algorithm, a modification of the classical Perceptron algorithm, that only achieves a mistake bound $\tilde{\mathcal{O}}(k^2/\gamma^2)$. Daniely & Helbertal (2013) show that a mistake bound of $\tilde{\mathcal{O}}(k/\gamma^2)$ is information-theoretically optimal, and present an inefficient algorithm to achieve it. Beygelzimer et al. (2019) provide a computationally efficient algorithm that only achieves the $\tilde{\mathcal{O}}(k/\gamma^2)$ rate under a stronger notion of margin. Others have considered more general data distributions and achieved a relative mistake bound of order $\sqrt{T}$, see e.g. van der Hoeven et al. (2021) and references therein.

In contrast, we introduce a simple online algorithm as a variant of PG with a uniform behavior policy that achieves the minimax optimal mistake bound $\tilde{\mathcal{O}}(k^N/\gamma^2)$, thus achieving the desirable $\tilde{\mathcal{O}}(k/\gamma^2)$ when $N = 1$ while also being computationally efficient per iteration. Thus, we answer the open question of achieving such an efficient online learning algorithm raised in (Beygelzimer et al., 2019).

**SGD for Separable Data.**   The study of SGD on logistic loss for linearly separable data first gained attention through its connection to the max-margin SVM and its implicit bias Soudry et al. (2018); Ji & Telgarsky (2019). Extensions to multiclass settings have been considered in (Ravi et al., 2024; Schliserman & Koren, 2025). Our analysis for both SGD and PG is more similar to Shamir (2021). In contrast, we significantly go beyond the binary setting by considering autoregressive models and adapting the margin assumption to sequences of labels, where each label has $k \geq 2$ choices.

**Policy Gradient Analysis.**   A comprehensive study of theoretical properties of policy gradient methods in tabular settings was carried out by Agarwal et al. (2021). However, their guarantees for PG with softmax parameterization are only asymptotic. Further works have made this guarantee non-asymptotic and extended beyond the tabular setting (Zhang et al., 2020; Mei et al., 2020; Zhang et al., 2021; Liu et al., 2020). We remark however that none of these results are directly applicable in our setting, since their generality prevents establishing tight bounds. We instead carry out an analysis tailored to the 0-1 reward structure for post-training an autoregressive model.

**Notation.**   For quantities $a$ and $b$, we use $a \lesssim b$ or $a = \mathcal{O}(b)$ to denote $a \leq Cb$ for some absolute constant $C > 0$, and define $a \gtrsim b$ and $a = \Omega(b)$ similarly. We use $\tilde{\mathcal{O}}$ and $\tilde{\Omega}$ to hide polylogarithmic factors. We use $f(\varepsilon) = o(1)$ to denote $f(\varepsilon) = \mathcal{O}(1/\log(1/\varepsilon))$, a stricter version of the usual definition $\lim_{\varepsilon \downarrow 0} f(\varepsilon) = 0$. We use the notation $\boldsymbol{y}_{1:i} = (y_j)_{j=1}^i$, and $\boldsymbol{y}_{1:0}$ is an empty sequence. We use $\mathrm{Clip}(x, a, b) := ((x \vee a) \wedge b)$ for $x, a, b \in \mathbb{R}$. For a set $\mathcal{Y}$, we define $\mathcal{Y}^+ := \bigcup_{i=1}^N \mathcal{Y}^i$. We use $\boldsymbol{e}_y$ to denote the $y$th standard basis of $\mathbb{R}^k$.

## 2. Autoregressive Linear Models and Pre-Training

Given access to a verifier, post-training can be performed using outcome rewards. This approach yields a contextual bandit formulation: given a prompt $\boldsymbol{x} \in \mathcal{X}$, the model $p_{\boldsymbol{W}}(\cdot \mid \boldsymbol{x})$ generates a response $\boldsymbol{y} = (y_1, \ldots, y_N) \in \mathcal{Y}^N$ of length $N$, and receives a reward $r(\boldsymbol{x}, \boldsymbol{y})$ from an outcome reward model (ORM). For simplicity, we fix the length of the response, while our arguments can easily be adapted to dynamic lengths. We assume $|\mathcal{Y}| = k$. For many verifiers this reward is binary, i.e. $r(\boldsymbol{x}, \boldsymbol{y}) = 1$ if $\boldsymbol{y}$ is a correct answer to $\boldsymbol{x}$, and $r(\boldsymbol{x}, \boldsymbol{y}) = 0$ otherwise. There are also settings where a process reward model (PRM) can provide intermediate signal to the learner. We study this formulation in the subsequent section.

We specify the generative model $p_{\boldsymbol{w}}(\cdot \mid \boldsymbol{x})$ using the decomposition $p_{\boldsymbol{w}}(\boldsymbol{y} \mid \boldsymbol{x}) = \prod_{i=1}^N p_{\boldsymbol{w}}(y_i \mid \boldsymbol{x}, \boldsymbol{y}_{1:i-1})$, where

$$p_{\boldsymbol{w}}(y_i \mid \boldsymbol{x}, \boldsymbol{y}_{1:i-1}) = \frac{\exp\left(\langle \boldsymbol{w}, \phi(\boldsymbol{x}, \boldsymbol{y}_{1:i-1}, y_i) \rangle\right)}{\sum_{y' \in \mathcal{Y}} \exp\left(\langle \boldsymbol{w}, \phi(\boldsymbol{x}, \boldsymbol{y}_{1:i-1}, y') \rangle\right)}.$$

Here, $\phi : \mathcal{X} \times \mathcal{Y}^+ \to \mathbb{R}^D$ is the feature map. Current models typically implement $\phi$ using a deep Transformer (Vaswani et al., 2017). The above formulation simplifies the analysis by only focusing on the final linear layer while freezing the feature map $\phi$.

We assume that each prompt $\boldsymbol{x}$ has a unique correct response $\boldsymbol{y}^*(\boldsymbol{x})$ that satisfies the following assumption.

**Assumption 2.1.**  Suppose there exists $\boldsymbol{w}^* \in \mathbb{R}^D$ such that $\|\boldsymbol{w}^*\|_2 = 1$ and for all $i \in [N]$, $y \neq y_i^*(\boldsymbol{x})$,

$$\langle \boldsymbol{w}^*, \phi(\boldsymbol{x}, \boldsymbol{y}_{1:i}^*(\boldsymbol{x})) \rangle \geq \langle \boldsymbol{w}^*, \phi(\boldsymbol{x}, \boldsymbol{y}_{1:i-1}^*(\boldsymbol{x}), y) \rangle + \gamma,$$

a.s. over $\boldsymbol{x}$ for some $\gamma > 0$. We further assume $\|\phi(\boldsymbol{x}, \boldsymbol{y}_{1:i}^*(\boldsymbol{x}))\|_2 \leq 1$ for all $i \in [N]$, $\boldsymbol{x}$, and $\boldsymbol{y}$.

When $N = 1$, letting $\phi(\boldsymbol{x}, y) = \mathrm{Vec}(\boldsymbol{e}_y \boldsymbol{x}^\top)$ and $\boldsymbol{w} = \mathrm{Vec}(\boldsymbol{W})$ with $\boldsymbol{W} \in \mathbb{R}^{k \times d}$, the above reduces to a standard $\gamma$-margin assumption for $k$-class linear classification, and the autoregressive model reduces to the usual logistic model.

The above only requires separability at the token level while assuming all previous tokens are correct, as opposed to sequence level separability which would be harder to satisfy. With Assumption 2.1, it might be tempting to only train on a single token to recover $\boldsymbol{w}^*$, thus remove dependence on $N$ from the complexity of supervised training. However, such a direction may not be unique, and a direction that separates some tokens might not be useful for the rest.

Before moving to the analysis of post-training, we first recall results about pre-training. During pre-training, we

have access to i.i.d. labeled examples $(\boldsymbol{x}^{(t)}, \boldsymbol{y}^*(\boldsymbol{x}^{(t)}))_{t \geq 0}$, and we can run SGD iterates as

$$\boldsymbol{w}_{t+1} = \boldsymbol{w}_t + \eta_t \nabla \log p_{\boldsymbol{w}_t}(\boldsymbol{y}^*(\boldsymbol{x}^{(t)}) \,|\, \boldsymbol{x}^{(t)}), \quad \text{(SGD)}$$

where $\eta_t$ is a potentially adaptive learning rate. (Chen et al., 2025a) demonstrated that with a constant LR, SGD suffers from a convergence rate that depends linearly on the sequence length $N$. They mitigate this issue by analyzing an Adagrad-type (Duchi et al., 2011) adaptive LR on SGD with a sufficiently large mini-batch size. Under Assumption 2.1, we can study the simpler variant above that operates in a one-pass mode over the pre-training data and does not require batching, hence can also be applied in online settings.

**Proposition 2.2.** *Suppose Assumption 2.1 holds, and let $(\boldsymbol{w}_t)_{t=0}^{T-1}$ denote the iterates of SGD with learning rate schedule $(\eta_t)$. Then,*

1. *(Constant LR) If $\eta_t = 1/(2N)$, then*

$$1 - \mathbb{E}\Big[ p_{\bar{\boldsymbol{w}}_T^{\mathrm{SGD}}}(\boldsymbol{y}^*(\boldsymbol{x}) \,|\, \boldsymbol{x}) \Big] \lesssim \frac{N \log(Tk\gamma^2)^2}{\gamma^2 T}$$

   *where $\bar{\boldsymbol{w}}_T^{\mathrm{SGD}} := \frac{1}{T} \sum_{t=0}^{T-1} \boldsymbol{w}_t$ is the averaged iterate.*

2. *(Adaptive LR) If*

$$\eta_t = \big( 2 + 4\|\nabla \log p_{\boldsymbol{w}_t}(\boldsymbol{y}^*(\boldsymbol{x}_t) \,|\, \boldsymbol{x}_t)\|_2 \big)^{-1},$$

   *then*

$$1 - \frac{1}{T} \sum_{t=0}^{T-1} \mathbb{E}[p_{\boldsymbol{w}_t}(\boldsymbol{y}^*(\boldsymbol{x}) \,|\, \boldsymbol{x})] \lesssim \frac{\log(TNk\gamma^2)^2}{\gamma^2 T}.$$

   *In other words, if we draw $\tau \sim \mathrm{Unif}(\{0, \dots, T-1\})$, then*

$$1 - \mathbb{E}[p_{\boldsymbol{w}_\tau}(\boldsymbol{y}^*(\boldsymbol{x}) \,|\, \boldsymbol{x})] \lesssim \frac{\log(TNk\gamma^2)^2}{\gamma^2 T}.$$

A few remarks are in order. First, it is well-known that the VC dimension of the class of $\gamma$-margin half-spaces in dimension $D \geq 1/\gamma^2$ is at least $\Omega(1/\gamma^2)$ (e.g. can be proved by considering an orthonormal set of points). Therefore, by standard minimax lower bounds on sample complexity using VC dimension in realizable settings (Shalev-Shwartz & Ben-David, 2014, Theorem 6.8), we require at least $\Omega(1/(\gamma^2 \varepsilon))$ samples to achieve a test error $\varepsilon$ in the worst-case for $N = 1$ and $k = 2$, which can thus be extended to all $N$ and $k$. This lower bound can alternatively be derived using the simpler technique we employ in Theorem 5.3. Interestingly, while for $N = \mathcal{O}(1)$ SGD with constant or adaptive LR (almost) achieves the minimax optimal sample complexity, for $N \gg 1$, it is only the adaptive LR that does so, with nearly the same rate as the $N = 1, k = 2$ case.

Moreover, while we state Proposition 2.2 and other results in the paper in a conventional statistical learning setup where data is i.i.d., proofs are obtained by an online analysis and an online-to-batch conversion (see Appendix A for details). As a consequence, an online learner that samples responses $\boldsymbol{y}^{(t)} \sim p_{\boldsymbol{w}_t}(\cdot \,|\, \boldsymbol{x}^{(t)})$ would make at most $\tilde{\mathcal{O}}(1/\gamma^2)$ mistakes in expectation over online (potentially adversarially chosen) data, which nearly matches the classical Perceptron algorithm and is minimax optimal, while covering the broader setting of autoregressive linear models.

## 3. Policy Gradient with Outcome Reward (Bandit Feedback)

For post-training, we only have access to i.i.d. prompts $(\boldsymbol{x}_t)_{t \geq 0}$ as well as a reward model, but not the labels directly. In this section, we assume access to a reward model for the final response generated, i.e. $r : \mathcal{X} \times \mathcal{Y}^N \to \{0, 1\}$ where

$$r(\boldsymbol{x}, \boldsymbol{y}) = \mathbb{1}[\boldsymbol{y} = \boldsymbol{y}^*(\boldsymbol{x})].$$

This stage is typically performed using some policy gradient algorithm such as PPO or GRPO. Here, we study the most basic policy gradient algorithm, also referred to as REIN-FORCE (Williams, 1992). The PG update with outcome rewards is given by

$$\boldsymbol{w}_{t+1} = \boldsymbol{w}_t + \eta_t r_t \nabla \log p_{\boldsymbol{w}_t}(\boldsymbol{y}^{(t)} \,|\, \boldsymbol{x}^{(t)}), \quad \text{(PG-OR)}$$

where $r_t := r(\boldsymbol{x}^{(t)}, \boldsymbol{y}^{(t)})$, $\boldsymbol{y}_t \sim q_t(\cdot \,|\, \boldsymbol{x}^{(t)})$ and $q_t$ is the behavior policy at round $t$ which can be adaptively chosen using past samples. Unless otherwise stated, we assume $\boldsymbol{w}_0 = \boldsymbol{0}$ for simplicity. While in practice the initialization is provided by the base model, in our analysis the benefit of the base model comes from the behavior policy, therefore we can simplify the analysis by initializing at zero. Two remarks are in order.

- For (PG-OR) to be policy gradient in the sense of following the population reward gradient, we have to introduce the importance weights $p_{\boldsymbol{w}_t}(\boldsymbol{y}^{(t)} \,|\, \boldsymbol{x}^{(t)})/q_t(\boldsymbol{y}^{(t)} \,|\, \boldsymbol{x}^{(t)})$ unless we are fully on-policy, i.e. $q_t = p_{\boldsymbol{w}_t}$. We defer the study of this fully on-policy PG to Appendix A.5 since our analysis for it yields a qualitatively similar behavior to subsequent PG variants we study, yet with a suboptimal convergence rate. When $q_t \neq p_{\boldsymbol{w}_t}$, practical algorithms such as PPO clip the importance weight to be close to 1. Our analysis can also accommodate clipped importance weights; in Appendix A.6 we study the following updates

$$\boldsymbol{w}_{t+1} = \boldsymbol{w}_t + \eta_t r_t \rho_t \nabla \log p_{\boldsymbol{w}_t}(\boldsymbol{y}^{(t)} \,|\, \boldsymbol{x}^{(t)})$$

  where $\rho_t = \mathrm{Clip}\big( \frac{p_{\boldsymbol{w}_t}(\boldsymbol{y}^{(t)} \,|\, \boldsymbol{x}^{(t)})}{q_t(\boldsymbol{y}^{(t)} \,|\, \boldsymbol{x}^{(t)})}, 1/\zeta, \zeta \big)$ for some $\zeta \geq 1$. In summary, we can show that all convergence rates will be scaled by a factor of $\zeta^2$. In the main text, we focus the

presentation on the case where the importance weights are removed, i.e. $\zeta = 1$, as it optimizes the convergence rate and achieves minimax optimality, which suggests that following the (unbiased estimate of) population reward gradient in this setting is not strictly necessary.

- Note that (PG-OR) considers a simple advantage estimator with no baseline. Once again, this is motivated by the minimax optimality of the updates. No advantage estimator achieves a better rate with only Assumption 2.1.

### 3.1. Conditional Convergence of PG

We begin with the following conditional guarantee. The proof of all results in this section are stated in Appendix A.

**Theorem 3.1** (PG-OR with Constant LR). *Let* $(\boldsymbol{w}_t)$ *denote the* (PG-OR) *iterates. Suppose Assumption 2.1 holds. For simplicity, assume* $\eta = 1/(2N)$ *and* $\|\boldsymbol{w}_0\|_2 \leq 1/\gamma \cdot \log(NkT\gamma^2)$. *Let* $\boldsymbol{x}$ *denote a test sample. For any* $\alpha \in (0, 1]$, *define the event* $\mathcal{E}_\alpha := \{\min_t q_t(\boldsymbol{y}^*(\boldsymbol{x}) \,|\, \boldsymbol{x}) \geq \alpha\}$ *and* $\pi_\alpha := \mathbb{P}(\mathcal{E}_\alpha)$. *Then,* $\bar{\boldsymbol{w}}_T^{\mathrm{PG}} = \frac{1}{T} \sum_{t=0}^{T-1} \boldsymbol{w}_t$ *satisfies*

$$\mathbb{E}\left[p_{\bar{\boldsymbol{w}}_T^{\mathrm{PG}}}(\boldsymbol{y}^*(\boldsymbol{x}) \,|\, \boldsymbol{x}) \,|\, \mathcal{E}_\alpha\right] \geq 1 - \tilde{\mathcal{O}}\left(\frac{N}{\pi_\alpha \alpha \gamma^2 T}\right). \quad (1)$$

The conditioning above highlights that the test performance on an individual sample after post-training depends on how likely the behavior policy $q_t$ is to generate the correct response for that sample. In the bound above, $\alpha$ denotes this likelihood. If we set $q_t = \mathrm{Unif}$ for all $t$, then $\pi_\alpha = 1$ for $\alpha = k^{-N}$ and $\pi_\alpha = 0$ for any $\alpha > k^{-N}$. The way a base model enters the above bound is through the function $\alpha \mapsto \pi_\alpha$. The slower it decays, the faster the convergence will be for samples that have an initial likelihood at least $\alpha$.

Similar to SGD, we can remove $N$ in the convergence rate above by using adaptive learning rates.

**Theorem 3.2** (PG-OR with Adaptive LR). *Consider the same setting as Theorem 3.1, except we use the adaptive learning rate* $\eta_t = (4 + 2\|\nabla \log p_{\boldsymbol{w}_t}(\boldsymbol{y}^{(t)} \,|\, \boldsymbol{x}^{(t)})\|_2)^{-1}$. *Then, in expectation over the randomness in training, in the test prompt* $\boldsymbol{x}$, *and in* $\tau \sim \mathrm{Unif}(\{0, \ldots, T-1\})$, *we have*

$$\mathbb{E}[p_{\boldsymbol{w}_\tau}(\boldsymbol{y}^*(\boldsymbol{x}) \,|\, \boldsymbol{x}) \,|\, \mathcal{E}_\alpha] \geq 1 - \tilde{\mathcal{O}}\left(\frac{1}{\pi_\alpha \alpha \gamma^2 T}\right), \quad (2)$$

*where we recall* $\mathcal{E}_\alpha := \{\min_t q_t(\boldsymbol{y}^*(\boldsymbol{x}) \,|\, \boldsymbol{x}) \geq \alpha\}$ *and* $\pi_\alpha := \mathbb{P}(\mathcal{E}_\alpha)$ *for any* $\alpha \in (0, 1]$.

*Remark* 3.3 (Online Learning Regret Bound). As discussed earlier, while we stated the guarantee of Theorem 3.2 for i.i.d. data, we can utilize the same proof technique for an arbitrary sequence $\boldsymbol{x}^{(t)}, \boldsymbol{y}^*(\boldsymbol{x}^{(t)})$, where an adversary can choose $\boldsymbol{x}^{(t)}$ and $\boldsymbol{y}^*(\boldsymbol{x}^{(t)})$ for all $t$, with the only constraint being Assumption 2.1 (see Proposition A.7 and its proof).

In particular, if $q_t$ is simply the uniform policy, we have $\pi_\alpha = 1$ for $\alpha = k^{-N}$, thus we can remove conditioning and write

$$\frac{1}{T} \sum_{t=0}^{T-1} \mathbb{E}\left[p_{\boldsymbol{w}_t}(\boldsymbol{y}^*(\boldsymbol{x}^{(t)}) \,|\, \boldsymbol{x}^{(t)})\right] \geq 1 - \tilde{\mathcal{O}}\left(\frac{k^N}{\gamma^2 T}\right).$$

Therefore, the online learning algorithm that draws the response $\boldsymbol{y}^{(t)}$ to $\boldsymbol{x}^{(t)}$ according to $p_{\boldsymbol{w}_t}$ will have an expected number of mistakes $\tilde{\mathcal{O}}(k^N/\gamma^2)$ over $T$ rounds. Indeed, this number of mistakes is minimax optimal (up to log terms) as established in Theorem 5.4. To our knowledge, this is the first online learning algorithm in this setting that is computationally efficient per iteration, while nearly achieving the minimax optimal number of mistakes. Prior works achieved this number of expected mistakes under a stronger notion of separability and only for $N = 1$ (Beygelzimer et al., 2019).

### 3.2. Unconditional Convergence of PG

In the previous section, we conditioned the expected error on the initial likelihoods of test samples. Our focus now is to use those results to bound the expected test error over the entire distribution. For a policy $q$, we define the Likelihood Quantile (LQ) function $\mathcal{Q}_q : \mathbb{R}[0, 1] \to \mathbb{R}[0, 1]$ as

$$\mathcal{Q}_q(\varepsilon) := \sup\left\{\alpha \in [0, 1] : \mathbb{P}_{\boldsymbol{x}}(q(\boldsymbol{y}^*(\boldsymbol{x}) \,|\, \boldsymbol{x}) \leq \alpha) \leq \varepsilon\right\}.$$

Specifically, $\mathcal{Q}_q(\varepsilon)$ provides the $\varepsilon$-quantile of the random variable $q(\boldsymbol{y}^*(\boldsymbol{x}) \,|\, \boldsymbol{x})$, which is the likelihood of the ground-truth data $(\boldsymbol{x}, \boldsymbol{y}^*(\boldsymbol{x}))$ under the base model $q$. As can be seen from Figure 2b, the more concentrated around 1 the random variable $q(\boldsymbol{y}^*(\boldsymbol{x}) \,|\, \boldsymbol{x})$ is (corresponding to longer pre-training), the larger $\mathcal{Q}_q(\varepsilon)$ will be for a fixed $\varepsilon$. With $\pi_\alpha := \mathbb{P}_{\boldsymbol{x}}(q(\boldsymbol{y}^*(\boldsymbol{x}) \,|\, \boldsymbol{x}) \geq \alpha)$, the mapping $\alpha \mapsto 1 - \pi_\alpha$ is the CDF of the likelihood over test samples, and $\mathcal{Q}_q$ is the generalized inverse of this function; it satisfies $\mathcal{Q}_q(1 - \pi_\alpha) \geq \alpha$. Through this observation, LQ is connected to other notions studied in the literature; $1 - \pi_\alpha$ is referred to as the $\alpha$-coverage profile in Chen et al. (2025a). We refer to references therein for other definitions in the literature.

We can examine lower bounds on $\mathcal{Q}_q$ when $q = p_{\bar{\boldsymbol{w}}_n^{\mathrm{SGD}}}$ is obtained by SGD pre-training with $T^{\mathrm{SGD}} = n$ samples:

- With the constant LR $\eta \asymp 1$:

$$\mathcal{Q}_q(\varepsilon) \geq e^{-\tilde{\mathcal{O}}(N/(n\gamma^2 \varepsilon))}$$

- With the adaptive LR $\eta_t \asymp (1 + \|\nabla \log p_{\boldsymbol{w}_t}\|_2)^{-1}$:

$$\mathcal{Q}_q(\varepsilon) \geq \left(1 - \tilde{\mathcal{O}}((n\gamma^2 \varepsilon)^{-1})\right) \vee 0.$$

Further, for the uninformative uniform policy $q$, we have $\mathcal{Q}_q(\varepsilon) = k^{-N}$ for all $\varepsilon < 1$. We can now state unconditional bounds on the accuracy of the post-trained model.

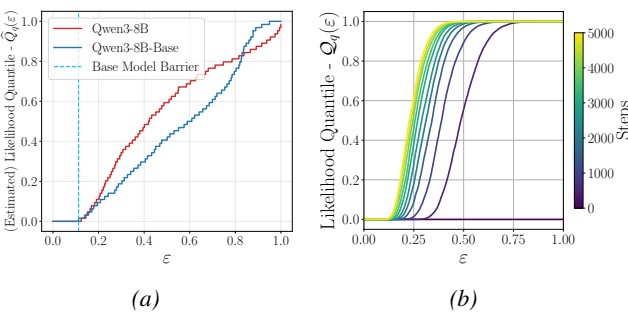

*(a)*           *(b)*

*Figure 2.* (a) Estimated LQ of Qwen3-8B and Qwen3-8B-Base on Math500 and (b) true LQ of our linear autoregressive model trained with Adagrad with varying number of steps on a synthetic task. As the model trains for longer, likelihoods increase and $\mathcal{Q}_q(\varepsilon)$ gets closer to 1 for fixed $\varepsilon$. Details provided in Section 6.

**Corollary 3.4** (Test Error of PG-OR)**.** *Suppose Assumption 2.1 holds. Given a base policy $q_0$, let $q = \frac{1}{2}q_0 + \frac{1}{2}$Unif. Consider* (PG-OR) *with $q_t = q$ for all $t$ and with the adaptive learning rate of Theorem 3.2. Let $\tau \sim$ Unif($\{0, \ldots, T-1\}$). Then, for all $\varepsilon \in (0,1)$, with*

$$T = \tilde{\mathcal{O}}\left( \frac{\min\left( \mathcal{Q}_{q_0}\left((1-o(1))\varepsilon\right)^{-1}, k^N \right)}{\gamma^2 \varepsilon} \right)$$

*iterations, we have* $\mathbb{E}[p_{\boldsymbol{w}_\tau}(\boldsymbol{y}^*(\boldsymbol{x}) \mid \boldsymbol{x})] \geq 1 - \varepsilon$.

*Remark* 3.5. We distinguish between the reciprocal and inverse in our notation; $\mathcal{Q}_{q_0}(\varepsilon)^{-1} = 1/\mathcal{Q}_{q_0}(\varepsilon)$ while $\mathcal{Q}_{q_0}^{-1}(\cdot)$ denotes the inverse function of $\mathcal{Q}_{q_0}(\cdot)$. By monotonicity of $\mathcal{Q}_{q_0}$, we have $\varepsilon \geq 1 - \pi_{k^{-N}} = \mathcal{Q}_{q_0}^{-1}(k^{-N})$ iff $\mathcal{Q}_{q_0}(\varepsilon) \geq k^{-N}$. Therefore, we can express the above bound as

$$T = \begin{cases} \tilde{\mathcal{O}}\left( \frac{\mathcal{Q}_{q_0}\left((1-o(1))\varepsilon\right)^{-1}}{\gamma^2 \varepsilon} \right) & \text{if} \quad \varepsilon \geq \mathcal{Q}_{q_0}^{-1}(k^{-N}) \\ \tilde{\mathcal{O}}\left( \frac{k^N}{\gamma^2 \varepsilon} \right) & \text{otherwise.} \end{cases}$$

$$(3)$$

To understand the base model barrier, consider the case where $q_0$ is given by SGD with adaptive LR on $n$ labeled samples. Then, $\mathcal{Q}_{q_0}^{-1}(k^{-N}) = \tilde{\Omega}(1/(n\gamma^2))$. This suggest that to go below expected error $\tilde{\mathcal{O}}(1/(\gamma^2 n))$, *which is already achieved by the SGD-trained base model*, policy gradient will require exponentially many iterations. One may wonder if this is due to a suboptimal bound on $\mathcal{Q}_{q_0}$ for SGD, or due to a fundamental suboptimality of SGD itself. In Section 5, we rule out both cases by showing a minimax bound of $\mathcal{Q}_q(\varepsilon) \leq \mathcal{O}(k^{-N}/(1 - \gamma^2 n\varepsilon))$ among all algorithms that output $q$ with $n$ labeled i.i.d. samples. If $n \ll 1/(\gamma^2 \varepsilon)$, the upper bound above inevitably depends on $k^N$. If $n \gg 1/(\gamma^2 \varepsilon)$, the SGD pre-trained model already achieves error $\varepsilon$. This phenomenon can also be observed in practice. Figure 2a shows that while for sufficiently large $\varepsilon$ the post-trained Qwen3-8B model has larger LQ than Qwen3-8B-Base, for $\varepsilon$ below a threshold, both models fail

completely and have LQ of zero. This suggests that post-training was unable to extend the model's support beyond regions where the base model already exhibited non-trivial LQ.

The barrier above is not specific to policy gradient and its variants. We demonstrate in Theorem 5.4 that (PG-OR) with the behavior policy described above is minimax optimal (up to log factors) among algorithms that have access to $q_0$ and only make one reward query per context. However, it is possible to improve both sample/iteration complexity and number of reward queries by considering a modification that makes multiple queries per round. Specifically, by choosing $q_t$ according to Algorithm 1, we have the following.

**Corollary 3.6.** *Suppose Assumption 2.1 holds. Given a base policy $q_0$ and for any $\varepsilon \in (0,1)$, consider* (PG-OR) *with adaptive learning rate of Theorem 3.2 and $q_t$ given by Algorithm 1 with $m = \min\left( \lceil \mathcal{Q}_{q_0}\left((1-o(1))\varepsilon\right)^{-1} \rceil, k^N \right)$. Let $T$ and $Q$ denote the total number of iterations (also samples) and reward queries (also queries to $q_0$) respectively, and $\tau \sim$ Unif($\{0, \ldots, T-1\}$). Then, with $Q = \tilde{\mathcal{O}}((m + \varepsilon^{-1})/\gamma^2)$ reward queries and $T = \tilde{\mathcal{O}}(1/(\gamma^2 \varepsilon))$ iterations, we have* $\mathbb{E}[p_{\boldsymbol{w}_\tau}(\boldsymbol{y}^*(\boldsymbol{x}) \mid \boldsymbol{x})] \geq 1 - \varepsilon$.

*Remark* 3.7. Compared to Corollary 3.4, the number of PG iterations and context samples only scale with $\tilde{\mathcal{O}}(1/(\gamma^2 \varepsilon))$. On the other hand, the number of reward queries (and sampling queries from $q_0$) still blows up exponentially for $\varepsilon < \mathcal{Q}_{q_0}^{-1}(k^{-N})$, thus still facing the base model barrier.

*Remark* 3.8. As we prove in Theorem 5.3, both the number of samples and reward queries above are minimax optimal (up to log factors) among learners with access to $q_0$.

Unlike Corollary 3.4, the behavior policy in Corollary 3.6 depends on the current policy $p_{\boldsymbol{w}_t}$, taking advantage of online exploration. If we remove line 1 from Algorithm 1, then the expected number of reward queries will always be exponential in $N$. When $\varepsilon < \mathcal{Q}_{q_0}^{-1}(k^{-N})$, we could choose $m = \infty$ and obtain the same bounds in expectation. In this case, PG and SGD coincide, and we only rely on Algorithm 1 to recover the ground-truth label of $\boldsymbol{x}^{(t)}$.

---

**Algorithm 1** Best-of-$m$ exploration for PG-OR.

**input** $\boldsymbol{x}^{(t)}, p_{\boldsymbol{w}_t}, q_0, m$
1:  $\boldsymbol{y} \sim p_{\boldsymbol{w}_t}(\cdot \mid \boldsymbol{x}^{(t)})$
2:  $j = 1$
3:  **while** $j \leq m$ **and** $r(\boldsymbol{x}^{(t)}, \boldsymbol{y}) = 0$ **do**
4:     $j = j+1, \boldsymbol{y} \sim (\frac{1}{2}q_0 + \frac{1}{2}\text{Unif})(\cdot \mid \boldsymbol{x})$
5:  **end while**
6:  **return** $\boldsymbol{y}$

---

## 4. Policy Gradient with Process Rewards

In the previous section, we demonstrated the hardness of post-training when we only have access to outcome rewards. An alternative is to provide the model with reward feedback throughout the generation process (Setlur et al., 2025a; Cui et al., 2025). While no longer a contextual bandit problem, the policy gradient iterates can be written as

$$\boldsymbol{w}_{t+1} = \boldsymbol{w}_t - \eta_t \nabla_{\boldsymbol{w}} \ell_t(\boldsymbol{w}_t),$$

$$\ell_t(\boldsymbol{w}_t) \coloneqq -\sum_{i=1}^{N} A_i^{(t)} \log p_{\boldsymbol{w}_t}(y_i^{(t)} \mid \boldsymbol{x}, \boldsymbol{y}_{1:i-1}^{(t)}) \quad \text{(PG-PR)}$$

where $\boldsymbol{y}^{(t)} \sim q_t(\cdot \mid \boldsymbol{x}_t)$ is sampled from some behavior policy, and $A_i^{(t)}$ is the advantage of taking action $y_i^{(t)}$ at step $i$ of rollout $t$. Suppose we have a 0-1 reward function $r^* : \mathcal{X} \times \mathcal{Y}^+ \to \{0, 1\}$ that satisfies

$$r^*(\boldsymbol{x}, \boldsymbol{y}_{1:i}) = \mathbb{1}[\boldsymbol{y}_{1:i} = \boldsymbol{y}_{1:i}^*(\boldsymbol{x})], \quad \forall i \in [N]. \quad (4)$$

The most basic estimator for advantage would be the (undiscounted) return from taking action $y_i^{(t)}$, i.e. $A_i^{(t)} = \sum_{j \geq i} r^*(\boldsymbol{x}^{(t)}, \boldsymbol{y}_{1:j}^{(t)})$. However, we can achieve optimal rates by using a simpler advantage $A_i^{(t)} = r^*(\boldsymbol{x}, \boldsymbol{y}_{1:i}^{(t)})$, which is what we consider in the following. This form of reward allows us to efficiently explore sequences with a base model that is only accurate for next token prediction (and not predicting the entire sequence). In particular, we can define the Token-Level Likelihood Quantile function $\mathcal{Q}_q^{\mathrm{TL}} : [0, 1] \to [0, 1]$ as

$$\mathcal{Q}_q^{\mathrm{TL}}(\varepsilon) \coloneqq \sup \big\{ \alpha \in [0, 1] :$$
$$\mathbb{P}_{\boldsymbol{x}} \big( \min_{i \in N} q(y_i^*(\boldsymbol{x}) \mid \boldsymbol{x}, \boldsymbol{y}_{i-1}^*(\boldsymbol{x})) \leq \alpha \big) \leq \varepsilon \big\}$$

Unlike LQ, for a uniform base policy $q$, we have $\mathcal{Q}_q^{\mathrm{TL}}(\varepsilon) = k^{-1}$ for all $\varepsilon < 1$, notably independent of $N$. This definition leads to the following guarantee. The proofs of this section are deferred to Appendix B.

**Theorem 4.1** (PG-PR with Adaptive Learning Rate). *Suppose Assumption 2.1 holds. Given a base policy $q_0$ and for any $\varepsilon \in (0, 1)$, let $q_t$ be the output of Algorithm 2 with $m = \lceil 2(\log N + 1) \cdot \min \big( \mathcal{Q}_{q_0}^{\mathrm{TL}} \big( (1 - o(1))\varepsilon \big)^{-1}, k \big) \rceil$. Let $T$ and $Q$ denote the total number of post-training iterations (also samples) and reward (also sampling from $q_0$) queries respectively, and $\tau \sim \mathrm{Unif}(\{0, \ldots, T-1\})$. Consider* (PG-PR) *iterates with the adaptive learning rate $\eta_t = (4 + 2\|\nabla_{\boldsymbol{w}} \ell_t(\boldsymbol{w}_t)\|)^{-1}$. Then, with $T = \tilde{\mathcal{O}}(1/(\gamma^2 \varepsilon))$ iterations and*

$$Q = \tilde{\mathcal{O}}\Big( \frac{N \min \big( \mathcal{Q}_{q_0}^{\mathrm{TL}} \big( (1 - o(1))\varepsilon \big)^{-1}, k \big)}{\gamma^2} + \frac{1}{\gamma^2 \varepsilon} \Big)$$

*reward queries we have $\mathbb{E}[p_{\boldsymbol{w}_\tau}(\boldsymbol{y}^*(\boldsymbol{x}) \mid \boldsymbol{x})] \geq 1 - \varepsilon$.*

We immediately observe that compared to Corollary 3.6 which required $\tilde{\mathcal{O}}((k^N + \varepsilon^{-1})/\gamma^2)$ reward queries in the worst-case, the above theorem requires only $\tilde{\mathcal{O}}((Nk + \varepsilon^{-1})/\gamma^2)$ reward queries. Further, the requirement on the base model is much milder for this theorem. $q_0$ only needs to generate the next correct token conditioned on a partially correct response. One can observe that for any base model $q_0$, we have $\mathcal{Q}_{q_0}^{\mathrm{TL}}(\varepsilon) \geq \mathcal{Q}_{q_0}(\varepsilon)$. In certain settings, the benefit of process reward models will be significant, and they will enable PG to go beyond the support of the base model and achieve significantly smaller expected test error. One such example is provided below.

**Corollary 4.2.** *Suppose Assumption 2.1 holds, and additionally, the feature map satisfies $\phi(\boldsymbol{x}, \boldsymbol{y}_{1:i}^*(\boldsymbol{x})) = \psi(\boldsymbol{x})$ for some $\psi : \mathbb{R}^d \to \mathbb{R}^D$ and all $i \in [N]$. Let $(\boldsymbol{w}_t^{\mathrm{SGD}})_{t=0}^{n-1}$ denote the SGD iterates with constant learning rate $\eta = 1/(2N)$, and define $q_0 = p_{\bar{\boldsymbol{w}}_n^{\mathrm{SGD}}}$ where $\bar{\boldsymbol{w}}_n^{\mathrm{SGD}} = \frac{1}{n} \sum_{t=0}^{n-1} \boldsymbol{w}_t^{\mathrm{SGD}}$. Let $(\boldsymbol{w}_t^{\mathrm{PG}})_{t=0}^{T-1}$ denote the* (PG-PR) *iterates in the setting of Theorem 4.1, and let $\tau \sim \mathrm{Unif}(\{0, \ldots, T-1\})$.*

1. *To achieve $\mathbb{E}\big[ p_{\bar{\boldsymbol{w}}_n^{\mathrm{SGD}}}(\boldsymbol{y}^*(\boldsymbol{x}) \mid \boldsymbol{x}) \big] \geq 1 - \varepsilon$, we have a supervised sample complexity of $n = \tilde{\mathcal{O}}(N/(\gamma^2 \varepsilon))$.*

2. *To achieve $\mathbb{E}\big[ p_{\boldsymbol{w}_\tau^{\mathrm{PG}}}(\boldsymbol{y}^*(\boldsymbol{x}) \mid \boldsymbol{x}) \big] \geq 1 - \varepsilon$, we have a (post-training) iteration complexity $T = \tilde{\mathcal{O}}(1/(\gamma^2 \varepsilon))$. Further, the sufficient number of reward queries is given by*

$$Q = \begin{cases} \tilde{\mathcal{O}}((N + \varepsilon^{-1})/\gamma^2) & \text{if} \quad n = \tilde{\Omega}(1/(\gamma^2 \varepsilon)) \\ \tilde{\mathcal{O}}((Nk + \varepsilon^{-1})/\gamma^2) & \text{otherwise} \end{cases}.$$

The above clearly demonstrates (PG-PR) with an SGD-trained base policy as an optimal combination of PG and SGD. While learning with (constant LR) SGD alone would require a number of supervised samples scaling linearly with $N$, and doing PG alone would require a number of reward queries scaling with $Nk$; their combination removes the $N$ factor in the supervised sample complexity and the $k$ factor in the number of reward queries. We remark however that this still does not outperform SGD with adaptive LR with the same number of supervised samples. We leave such an attempt to future work.

## 5. Lower Bounds

In this section, we argue about the tightness and optimality of the policy gradient algorithms studied for post-training, and of SGD with adaptive LR for pre-training. To establish post-training optimality, it is natural to define classes of algorithms with access to a base model and a reward function. Our lower bounds cover both online learning settings where the learner receives only one reward feedback per iteration

**Algorithm 2** Best-of-$m$ exploration for PG-PR.

---

**input** $(\boldsymbol{x}^{(t)}, p_{\boldsymbol{w}_t}, q_0, m)$

1: $\boldsymbol{y} \sim p_{\boldsymbol{w}_t}(\cdot \mid \boldsymbol{x}^{(t)})$
2: **if** $r^*(\boldsymbol{x}, \boldsymbol{y}) = 1$ **then**
3:     **return** $\boldsymbol{y}$
4: **end if**
5: **for** $i = 1, \ldots, N$ **do**
6:     $j = 1, y_{i,j} \sim (\frac{1}{2}q_0 + \frac{1}{2}\text{Unif})(\cdot \mid \boldsymbol{x}, \boldsymbol{y}_{1:i-1})$
7:     **while** $j \leq m$ **and** $r^*(\boldsymbol{x}, \boldsymbol{y}_{1:i-1}, y_{i,j}) = 0$ **do**
8:         $j = j + 1, y_{i,j} \sim (\frac{1}{2}q_0 + \frac{1}{2}\text{Unif})(\cdot \mid \boldsymbol{x}, \boldsymbol{y}_{1:i-1})$
9:     **end while**
10:    $y_i = y_{i,j}$
11: **end for**
12: **return** $\boldsymbol{y}$

---

and its goal is to minimize the number of mistakes, and the statistical setting where the learner has access to the reward model and can make multiple queries on the same context, and its goal is to have high accuracy on a new i.i.d. sample.

We are specifically interested in the role the base model plays in the lower bounds. Hence, we formalize the definition of online and statistical learners with access to a base model as an oracle, and we believe these definitions might be useful in subsequent works on this topic.

**Definition 5.1** (Online Learner with Bandit Feedback). An online learner with access to base model $q(\cdot \mid \boldsymbol{x})$ is a potentially randomized algorithm that at every iteration receives $\boldsymbol{x}_t$, predicts $\boldsymbol{y}_t$ (using its history and the base model), receives reward $r(\boldsymbol{x}_t, \boldsymbol{y}_t)$, and proceeds to the next round.

**Definition 5.2** (Statistical Learner with Bandit Feedback). A statistical learner with access to base model $q(\cdot \mid \boldsymbol{x})$ is a potentially randomized algorithm that receives $n$ samples $\{\boldsymbol{x}_t\}_{t=1}^n$, and on sample $t$ makes $m_t$ reward queries $\{r(\boldsymbol{x}_t, \boldsymbol{y}_t^{(i)})\}_{i=1}^{m_t}$ where $\boldsymbol{y}_t^{(i)}$ can be chosen adaptively using access to the base model. The algorithm then constructs the training set $S = \{(\boldsymbol{x}_t, \boldsymbol{y}_t^{(i)}, r(\boldsymbol{x}_t, \boldsymbol{y}_t^{(i)})) : t \in [T], i \in [m_t]\}$ and returns a prediction rule $\hat{\boldsymbol{y}}(S) : \mathcal{X} \to \mathcal{Y}^N$.

The performance of an online learner is measured by the number of mistakes it makes over $T$ rounds, i.e. $\sum_{t=0}^{T-1} \mathbb{1}[\boldsymbol{y}_t \neq \boldsymbol{y}^*(\boldsymbol{x}_t)]$, while the performance of a statistical learner is measured by its expected error over a new test sample, i.e. $\mathbb{P}_{\boldsymbol{x}}(\hat{\boldsymbol{y}}(S)(\boldsymbol{x}) \neq y^*(\boldsymbol{x}))$. Note that (PG-OR) is an example of both algorithms. It can be used to make online predictions as $\boldsymbol{y}_t \sim p_{\boldsymbol{w}_t}(\cdot \mid \boldsymbol{x}_t)$ (assuming the behavior policy does not make additional reward queries), and it can also be used to make predictions on new test samples at the end of training via $\hat{\boldsymbol{y}} \sim p_{\bar{\boldsymbol{w}}}(\cdot \mid \boldsymbol{x})$ where $\bar{\boldsymbol{w}}$ is the averaged iterate.

With the above definitions, we are ready to state the lower bound for statistical learners, which establishes the near

optimality of the PG variant with adaptive learning rate presented in Corollary 3.6. The proofs of this section are deferred to Appendix C.

**Theorem 5.3** (Lower Bound for Statistical Learner under ORM). *Let $(\boldsymbol{x}_t)_{t\geq 0}$ be an i.i.d. sequence drawn from some distribution $\mathcal{D}$. Consider the statistical learner of Definition 5.2, where $r$ is the outcome reward. For every $\gamma > 0$, $\alpha \in [k^{-N}, 0.5]$, $\varepsilon^* \in (0, 1/2)$, $\varepsilon \in (0, 1/2)$, and $D \geq k\lfloor 1/\gamma^2 \rfloor$, there exists a base policy $q$ satisfying*

$$\mathcal{Q}_q(\varepsilon) = \begin{cases} \alpha & \text{if} \quad \varepsilon \geq \varepsilon^* \\ k^{-N} & \text{if} \quad \varepsilon < \varepsilon^* \end{cases} \tag{5}$$

*and a feature map $\phi$, such that for every learner with access to $q$, there exist a distribution $\mathcal{D}$ and $\boldsymbol{w}^*$ that satisfy Assumption 2.1 with margin $\gamma$ and to achieve expected test error less than $\varepsilon$ the learner requires at least $Q = \Omega\big(\mathcal{Q}_q\big((1 + o(1))\varepsilon\big)^{-1}/\gamma^2\big)$ reward queries and at least $T = \Omega(1/(\gamma^2\varepsilon))$ samples.*

We highlight a contrast with the result of (Jia et al., 2025). They show that learning a reward model using trajectory samples can be done efficiently with a polynomial dependence on $N$. Our lower bound however, states that the number of reward queries, either true reward or queries to a learned reward model, may have to scale exponentially with $N$ in the worst-case, which provides a computational, rather than a statistical, barrier.

Next, we establish the near optimality of the online PG variant with adaptive learning rate of Corollary 3.4 among online learners.

**Theorem 5.4** (Lower Bound for Online Learner under ORM). *Consider the same setting as Theorem 5.3, except with the online learner of Definition 5.1. There exists a base policy $q$ satisfying (5) and a feature map $\phi$ such that for every learner, there is a distribution such that to achieve expected test error $\varepsilon$ on a new sample the learner needs at least $T = \Omega\big(\mathcal{Q}_q\big((1 + o(1))\varepsilon\big)^{-1}/(\gamma^2\varepsilon)\big)$ reward queries (equivalently samples). Further, over $T \geq k^N/\varepsilon^*$ rounds the expected number of mistakes is at least $\Omega(k^N/\gamma^2)$.*

### 5.1. Lower Bounds on the Base Model

So far we established that under an ORM, the optimal number of reward queries to achieve expected error $\varepsilon$ scales as $\mathcal{Q}_q(\varepsilon)^{-1}/\gamma^2$. One may hope however that there is a supervised pre-training algorithm that with a modest number of samples $n$ can achieve an LQ that is polynomially (rather than exponentially) small. The following lower bound shows that such an algorithm does not exist in general.

**Theorem 5.5** (Lower Bound for Pre-Training). *Let $S_n := \{(\boldsymbol{x}^{(t)}, \boldsymbol{y}^*(\boldsymbol{x}^{(t)}))\}_{t=1}^n$ be $n$ i.i.d. samples drawn from some distribution $\mathcal{D}$. Consider a learner that receives $S_n$ and*

*outputs a base policy q. For every $n \geq 1$, $\gamma > 0$, $\varepsilon \in (0, 1/(8n\gamma^2))$, and $D \geq k\lfloor 1/\gamma^2 \rfloor$ and every learner, there exists a distribution $\mathcal{D}$ that satisfies Assumption 2.1, and $\mathcal{Q}_q(\varepsilon) \leq k^{-N}/(1 - \mathcal{O}(\gamma^2 n\varepsilon))$.*

The above lower bound implies that to have $\mathcal{Q}_q(\varepsilon) \gg k^{-N}$, any pre-training algorithm needs $n \gg 1/(\gamma^2\varepsilon)$. On the other hand, SGD (with adaptive LR) has an expected error of $\tilde{\mathcal{O}}(1/(\gamma^2 n))$ over $n$ samples. Thus with $n \gg 1/(\gamma^2\varepsilon)$ samples, SGD can already achieve an expected error of order $\varepsilon$, and there will be no significant improvement coming from PG in terms of expected error. This argument shows that the barrier of the base model is not an artifact of our analysis or specific to pre-training with SGD, it is a fundamental property of post-training with outcome rewards.

# 6. Experiments

To validate intuitions from our theory, we perform experiments on synthetic well as a mathematical reasoning tasks.

For the mathematical reasoning task, we use two models from the Qwen3 family (Yang et al., 2025), namely Qwen3-8B-Base which is a pre-trained base model, and Qwen3-8B which has undergone post-training for improved reasoning capabilities. These models are evaluated on the Math-500 benchmark, a dataset consisting of 500 problems from the MATH dataset (Lightman et al., 2024). To estimate the likelihood of generating a correct answer, we generate 64 completions for each problem, and estimate the likelihood as the ratio of correct over all completions. We use temperature 0.7, top-p 0.95, and top-k 50 for generation. The resulting likelihood quantile functions are plotted in Figure 2a.

In the synthetic task, $x$ is drawn according to either a mixture model or uniformly from the hypercube $\{\pm 1\}^d$. For the mixture model, to generate $x$, we sample uniformly from the standard basis vectors, add an isotropic Gaussian noise with per-dimension standard deviation $0.05/\sqrt{d}$ whose norm is clipped at $0.05$, and project the resulting vector on $\mathbb{S}^{d-1}(\sqrt{d})$.

To generate $y^*(x)$, we first fix $W_1^* \in \mathbb{R}^{k \times d}$ and $W_2^* \in \mathbb{R}^{k \times k}$ once by drawing their entries i.i.d. from $\mathcal{N}(0, 1)$. Then $y_1^*(x) = \arg\max(W_1^* x)$ and $y_i^* = \arg\max(W_1^* x + W_2^* e_{y_{i-1}^*(x)})$. Since $y_{i+1}$ can be predicted from $x$ and $y_i$, we can use the feature map $\phi(x, y_{1:i}) = \text{Concat}\left(\text{Vec}(e_{y_i} x^\top), \text{Vec}(e_{y_i} e_{y_{i-1}}^\top)\right)$ if $i > 1$ and $\phi(x, y) = \text{Concat}\left(\text{Vec}(e_y x^\top), 0_{k^2}\right)$ otherwise. Thus $\phi(\cdot, \cdot) \in \mathbb{R}^{dk+k^2}$.

In Figure 1, we use the mixture dataset described above with $N = 128$ and $d = k = 32$. To obtain the base model, we train a linear autoregressive model initialized from zero using Adagrad on negative log-likelihood for 1000 steps with a learning rate of 0.1 and a batch size of 256. We then use this model to initialize on-policy PG with either outcome rewards of Section 3 or process rewards of Section 4. For PG, we use Adagrad with 4000 steps, 0.1 learning rate, and a batch size of 1024. Each batch is a fresh draw of i.i.d. samples during both pre- and post-training.

For Figure 1, we define off-support samples as mixture centers that have a likelihood under the base model smaller than $10^{-12}$. 4 out of 32 centers satisfy this constraint. We plot the average likelihood of generating a correct response over these 4 samples throughout PG training. PG with ORM is unable to boost the likelihood over any of these 4 samples, while the average likelihood improves under PRM. We further plot the expected error over 1024 i.i.d. test samples throughout PG. We also plot the likelihood trajectory of 16 out of 32 mixture centers under ORM, where we can observe samples that start near 0 stay near 0. This is in line with Theorem 3.2. For these samples, the initial likelihood is $\alpha \asymp k^{-N}$, and improving over the base model takes exponentially many iterations. However, samples with larger $\alpha$ show progress throughout PG.

In Figure 2b, we illustrate the evolution of the LQ function over the supervised training trajectory. For this experiment, we use the uniform over hypercube distribution for $x$, and set $N = 128$, $d = 32$, and $k = 10$. We use the Adagrad optimizer with learning rate 1.0 and batch size 256.

The results can be reproduced using https://github.com/mousavih/rlvr-base-model-barrier.

# 7. Conclusion

In this paper, we studied variants of policy gradient for post-training an autoregressive linear model, where the responses satisfy a separability assumption. We proved that with an outcome reward model, PG can efficiently increase the likelihood of samples inside the support of the base model. To go outside this support and achieve test error significantly smaller than the base model, the number of reward queries depends on a property of the base model we call the likelihood quantile, and can be exponential in the sequence length. This issue can be alleviated using a process reward model, where we introduce a token-level likelihood quantile and show that it scales more favorably with sequence length. We further demonstrated the tightness of our analysis, as well as fundamental statistical limitations on LQ, through minimax lower bounds.

While we assume access to accurate process rewards, it is interesting to find algorithms that can learn such a reward with statistical and computational efficiency whenever possible. Another open question is to extend the results of this paper to settings with noisy and non-separable responses, and develop optimal post-training algorithms accordingly.

## Acknowledgements

The authors would like to thank Denny Wu, Juno Kim, and anonymous reviewers for helpful discussions and feedback. MAE was partially supported by the NSERC Grant [2019-06167], the CIFAR AI Chairs program, the CIFAR Catalyst grant, and the Ontario Early Researcher Award.

## Impact Statement

This paper presents work whose goal is to advance the theory of Machine Learning. There are many potential societal consequences of this field in general, none of which we feel must be specifically highlighted here.

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

## A. Omitted Results and Proofs of Section 3

Our core argument for PG upper bounds is that PG can be seen as online gradient descent on a convex loss. Therefore, we state and use the following general fact for online gradient descent.

**Proposition A.1.** *Let $(\ell_t)_{t=0}^{T-1}$ be a sequence of convex and non-negative functions on $\mathbb{R}^D$. Consider online gradient descent updates $(w_t)_{t=0}^{T}$, initialized from $w_0$ and defined as*

$$w_{t+1} = w_t - \eta_t \nabla \ell_t$$

*where $(\eta_t)_{t=0}^{T-1}$ is the learning rate sequence. Then:*

1. *If $\left\| \nabla^2 \ell_t \right\|_2 \leq L$ for some $L > 0$ and all $t$, let $\eta_t = \eta$ where $\eta \in (0, L^{-1})$. Then, for all $v \in \mathbb{R}^D$ we have*

$$\frac{1}{T} \sum_{t=0}^{T-1} \ell_t(w_t) \leq \frac{1}{1 - \eta L} \left( \frac{1}{T} \sum_{t=0}^{T-1} \ell_t(v) + \frac{\|v - w_0\|_2^2}{2\eta T} \right). \tag{6}$$

2. *If $\|\nabla \ell_t\|_2 \leq C\ell_t$ for some $C > 0$ and all $t$, let $\eta_t = \eta/(\lambda + \|\nabla \ell_t\|_2)$ where $\lambda > 0$ and $\eta \in (0, 2C^{-1})$. Then, for all $v \in \mathbb{R}^D$ we have*

$$\frac{1}{T} \sum_{t=0}^{T-1} \frac{\ell_t(w_t)}{1 + C\lambda^{-1} \ell_t(w_t)} \leq \frac{1}{1 - C\eta/2} \left( \frac{1}{T} \sum_{t=0}^{T-1} \ell_t(v) + \frac{\lambda \|v - w_0\|_2^2}{2\eta T} \right). \tag{7}$$

While the constant learning rate case is well-known in the literature, the adaptive learning rate case, inspired by (Chen et al., 2025a), is particularly relevant for gradient descent to maximize log-likelihood on softmax-linear policies where the smoothness constant can grow with the sequence length.

To prove this proposition, we recall the following lemmas.

**Lemma A.2.** *Suppose $f : \mathbb{R}^d \to \mathbb{R}$ satisfies $\left\| \nabla^2 f \right\|_2 \leq L$. Then, $\|\nabla f(x)\|_2^2 \leq 2L(f(x) - \inf f)$ for all $x \in \mathbb{R}^d$.*

*Proof.* By smoothness, for any pair $x, x'$ we have $f(x') \leq f(x) + \langle \nabla f(x), x' - x \rangle + \frac{L}{2} \|x' - x\|_2^2$. The proof is completed by letting $x' = x - \frac{1}{L} \nabla f(x)$. $\square$

**Lemma A.3.** *Suppose $(u_t)_{t=0}^{T-1}$ is an arbitrary sequence of vectors, and starting from $w_0$, define $w_{t+1} := w_t - u_t$ for all $t \leq T - 1$. Then, for any $v$,*

$$\sum_{t=0}^{T-1} \langle u_t, w_t - v \rangle \leq \frac{\|v - w_0\|^2}{2} + \frac{\sum_{t=0}^{T-1} \|u_t\|^2}{2}.$$

*Proof.* For any $t \geq 0$,

$$\|w_{t+1} - v\|^2 = \|w_t - v\|^2 + \|u_t\|^2 - 2\langle u_t, w_t - v \rangle.$$

By rearranging the terms,

$$2\sum_{t=0}^{T-1} \langle u_t, w_t - v \rangle = \sum_{t=0}^{T-1} \|w_t - v\|^2 - \|w_{t+1} - v\|^2 + \sum_{t=0}^{T-1} \|u_t\|^2 \leq \|w_0 - v\|^2 + \sum_{t=0}^{T-1} \|u_t\|^2.$$

$\square$

We can now state the proof of the proposition.

*Proof of Proposition A.1.* By convexity of $\ell_t$, for any $v \in \mathbb{R}^D$ we have $\ell_t(w_t) - \ell_t(v) \leq \langle \nabla \ell_t(w_t), w_t - v \rangle$. Therefore

$$\sum_{t=0}^{T-1} \eta_t(\ell_t(w_t) - \ell_t(v)) \leq \sum_{t=0}^{T-1} \langle \eta_t \nabla \ell_t(w_t), w_t - v \rangle$$

$$\leq \frac{\|v - w_0\|_2^2}{2} + \frac{\sum_{t=0}^{T-1} \eta_t^2 \|\nabla \ell_t(w_t)\|_2^2}{2}. \tag{8}$$

Now, suppose $\left\|\nabla^2 \ell_t\right\|_2 \leq L$. By Lemma A.2 and non-negativity of $\ell_t$, we have $\left\|\nabla \ell_t\right\|_2^2 \leq 2L\ell_t$, consequently

$$\sum_{t=0}^{T-1} \eta_t(\ell_t(\boldsymbol{w}_t) - \ell_t(\boldsymbol{v})) \leq \frac{\|\boldsymbol{v} - \boldsymbol{w}_0\|_2^2}{2} + \sum_{t=0}^{T-1} \eta_t^2 \ell_t(\boldsymbol{w}_t).$$

Recall that in the first case we choose $\eta_t = \eta < 1/L$, therefore by rearranging the above

$$(1 - \eta L) \sum_{t=0}^{T-1} \ell_t(\boldsymbol{w}_t) \leq \sum_{t=0}^{T-1} \ell_t(\boldsymbol{v}) + \frac{\|\boldsymbol{v} - \boldsymbol{w}_0\|_2^2}{2},$$

which finishes the proof of the first case.

For the second case, we have $\eta_t = \eta/(\lambda + \|\nabla \ell_t(\boldsymbol{w}_t)\|_2)$. Thus $\eta_t \|\nabla \ell_t(\boldsymbol{w}_t)\|_2 \leq \eta$. Combining this fact with (8) yields

$$\sum_{t=0}^{T-1} \eta_t(\ell_t(\boldsymbol{w}_t) - \ell_t(\boldsymbol{v})) \leq \frac{\|\boldsymbol{v} - \boldsymbol{w}_0\|_2^2}{2} + \frac{\eta \sum_{t=0}^{T-1} \eta_t \|\nabla \ell_t(\boldsymbol{w}_t)\|_2}{2}$$

$$\leq \frac{\|\boldsymbol{v} - \boldsymbol{w}_0\|_2^2}{2} + \frac{C\eta \sum_{t=0}^{T-1} \eta_t \ell_t(\boldsymbol{w}_t)}{2}.$$

By rearranging the terms,

$$\left(1 - \frac{C\eta}{2}\right) \sum_{t=0}^{T-1} \eta_t \ell_t(\boldsymbol{w}_t) \leq \sum_{t=0}^{T-1} \eta_t \ell_t(\boldsymbol{v}) + \frac{\|\boldsymbol{v} - \boldsymbol{w}_0\|_2^2}{2}$$

$$\leq \sum_{t=0}^{T-1} \frac{\eta \ell_t(\boldsymbol{v})}{\lambda} + \frac{\|\boldsymbol{v} - \boldsymbol{w}_0\|_2^2}{2}.$$

Furthermore, notice that

$$\eta_t \ell_t(\boldsymbol{w}_t) = \frac{\eta \ell_t(\boldsymbol{w}_t)}{\lambda + \|\nabla \ell_t(\boldsymbol{w}_t)\|_2} \geq \frac{\eta \ell_t(\boldsymbol{w}_t)}{\lambda + C\ell_t(\boldsymbol{w}_t)}.$$

As a result

$$\left(1 - \frac{C\eta}{2}\right) \sum_{t=0}^{T-1} \frac{\ell_t(\boldsymbol{w}_t)}{1 + C\lambda^{-1} \ell_t(\boldsymbol{w}_t)} \leq \sum_{t=0}^{T-1} \ell_t(\boldsymbol{v}) + \frac{\lambda \|\boldsymbol{v} - \boldsymbol{w}_0\|_2^2}{2\eta},$$

which concludes the proof of the second case. $\qquad\square$

With this general theorem in hand, we can prove the results of Section 3. Note that Proposition 2.2 is a special case of Theorem 3.1 and Theorem 3.2, where $q_t(\cdot \mid \boldsymbol{x}) = \delta_{\boldsymbol{y}^*(\boldsymbol{x})}$ is the Dirac measure. As a result, we only need to prove the two theorems.

### A.1. Proof of Theorem 3.1

This theorem uses the first case of Proposition A.1. Thus, we need to estimate the smoothness of log-likelihood, achieved by the following lemma.

**Lemma A.4.** *For any $\boldsymbol{w} \in \mathbb{R}^D, \boldsymbol{x} \in \mathbb{R}^d$, and $\boldsymbol{y} \in \mathcal{Y}^N$, we have*

$$\nabla_{\boldsymbol{w}}^2 \log p_{\boldsymbol{w}}(\boldsymbol{y} \mid \boldsymbol{x}) = \sum_{i=1}^{N} -\mathbb{E}_{Y_i}\left[\phi(\boldsymbol{x}, \boldsymbol{y}_{1:i-1}, y)\phi(\boldsymbol{x}, \boldsymbol{y}_{1:i-1}, Y_i)^{\top}\right] + \mathbb{E}_{Y_i}\left[\phi(\boldsymbol{x}, \boldsymbol{y}_{1:i-1}, Y_i)\right] \mathbb{E}_{Y_i}\left[\phi(\boldsymbol{x}, \boldsymbol{y}_{1:i-1}, Y_i)^{\top}\right]$$

*where $Y_i \sim p_{\boldsymbol{w}}(\cdot \mid \boldsymbol{x}, \boldsymbol{y}_{1:i-1})$. As a result, if $\|\phi(\boldsymbol{x}, \boldsymbol{y}_{1:i})\|_2 \leq R$ for all $\boldsymbol{x}, \boldsymbol{y}, i$, then $\left\|\nabla^2 \log p_{\boldsymbol{w}}(\boldsymbol{y} \mid \boldsymbol{x})\right\|_2 \leq NR^2$ for all $\boldsymbol{x}, \boldsymbol{y}$.*

*Proof.* We have

$$\nabla_{\boldsymbol{w}}^2 \log p_{\boldsymbol{w}}(\boldsymbol{y}\,|\,\boldsymbol{x}) = \sum_{i=1}^{N} \nabla_{\boldsymbol{w}}^2 \log p_{\boldsymbol{w}}(y_i\,|\,\boldsymbol{x}, \boldsymbol{y}_{1:i-1}).$$

Further,

$$\nabla_{\boldsymbol{w}} \log p_{\boldsymbol{w}}(y_i\,|\,\boldsymbol{x}, \boldsymbol{y}_{1:i-1}) = \phi(\boldsymbol{x}, \boldsymbol{y}_{1:i-1}, y_i) - \mathbb{E}_{Y_i}\big[\phi(\boldsymbol{x}, \boldsymbol{y}_{1:i-1}, Y_i)\big].$$

Taking a second derivative yields

$$
\begin{aligned}
\nabla_{\boldsymbol{w}}^2 \log p_{\boldsymbol{w}}(y_i\,|\,\boldsymbol{x}, \boldsymbol{y}_{1:i-1}) &= -\sum_{y=1}^{k} \phi(\boldsymbol{x}, \boldsymbol{y}_{1:i-1}, y) \nabla p_{\boldsymbol{w}}(y\,|\,\boldsymbol{x}, \boldsymbol{y}_{1:i-1}) \\
&= -\sum_{y=1}^{k} p_{\boldsymbol{w}}(y\,|\,\boldsymbol{x}, \boldsymbol{y}_{1:i-1}) \phi(\boldsymbol{x}, \boldsymbol{y}_{1:i-1}, y) \nabla \log p_{\boldsymbol{w}}(y\,|\,\boldsymbol{x}, \boldsymbol{y}_{1:i-1}) \\
&= -\mathbb{E}_{Y_i}\big[\phi(\boldsymbol{x}, \boldsymbol{y}_{1:i-1}, Y_i) \nabla \log p_{\boldsymbol{w}}(Y_i\,|\,\boldsymbol{x}, \boldsymbol{y}_{1:i-1})\big] \\
&= -\mathbb{E}_{Y_i}\big[\phi(\boldsymbol{x}, \boldsymbol{y}_{1:i-1}, Y_i) \phi(\boldsymbol{x}, \boldsymbol{y}_{1:i-1}, Y_i)^\top\big] + \mathbb{E}_{Y_i}\big[\phi(\boldsymbol{x}, \boldsymbol{y}_{1:i-1}, Y_i)\big] \mathbb{E}_{Y_i}\big[\phi(\boldsymbol{x}, \boldsymbol{y}_{1:i-1}, Y_i)^\top\big],
\end{aligned}
$$

which completes the proof. $\square$

To handle settings that involve both i.i.d. and online settings, we state a more general version of Theorem 3.1 as a corollary of Proposition A.1, and then show how Theorem 3.1 for i.i.d. data follows from this corollary.

**Proposition A.5.** *Let $(\boldsymbol{w}_t)$ denote the* (PG-OR) *iterates. Suppose the sequence $(\boldsymbol{x}^{(t)}, \boldsymbol{y}^*(\boldsymbol{x}^{(t)}))$ is chosen arbitrarily (in particular not necessarily i.i.d.), with the only constraint being Assumption 2.1. For simplicity, assume $\eta_t = 1/(2N)$ for all $t$ and $\|\boldsymbol{w}_0\|_2 \le 1/\gamma \cdot \log(kT\gamma^2)$. Then,*

$$\mathbb{E}_{\boldsymbol{y}^{(1)},\ldots,\boldsymbol{y}^{(T)}}\left[\frac{1}{T}\sum_{t=0}^{T-1} -q_t(\boldsymbol{y}^*(\boldsymbol{x}^{(t)})\,|\,\boldsymbol{x}^{(t)}) \log p_{\boldsymbol{w}_t}(\boldsymbol{y}^*(\boldsymbol{x}^{(t)})\,|\,\boldsymbol{x}^{(t)})\right] \le \frac{10N\log(kT\gamma^2)^2}{\gamma^2 T}, \tag{9}$$

*where the expectation is over the randomness in sampling $\boldsymbol{y}^{(t)} \sim p_{\boldsymbol{w}_t}(\cdot\,|\,\boldsymbol{x}^{(t)})$.*

*Proof.* Note that since $r(\boldsymbol{x}_t, \boldsymbol{y}_t) = \mathbb{1}[\boldsymbol{y}_t = \boldsymbol{y}^*(\boldsymbol{x}_t)]$, we can rewrite (PG-OR) as

$$\boldsymbol{w}_{t+1} = \boldsymbol{w}_t + \eta_t r(\boldsymbol{x}^{(t)}, \boldsymbol{y}^{(t)}) \nabla \log p_{\boldsymbol{w}_t}(\boldsymbol{y}^{*(t)}\,|\,\boldsymbol{x}^{(t)}).$$

For simplicity, throughout the proof we will use the shorthand $\boldsymbol{y}^{*(t)} := \boldsymbol{y}^*(\boldsymbol{x}^{(t)})$. Define

$$\ell_t(\boldsymbol{w}) := -r(\boldsymbol{x}^{(t)}, \boldsymbol{y}^{(t)}) \log p_{\boldsymbol{w}_t}(\boldsymbol{y}^{*(t)})\,|\,\boldsymbol{x}^{(t)}).$$

Then $\ell_t$ is convex and $\|\nabla^2 \ell_t\|_2 \le N$ by Lemma Lemma A.4 and the fact that $r(\boldsymbol{x}^{(t)}, \boldsymbol{y}^{(t)}) \le 1$. Therefore, for $\eta_t = 1/(2N)$, we can invoke the first case of Proposition A.1 to write

$$\frac{1}{T}\sum_{t=0}^{T-1} -r(\boldsymbol{x}^{(t)}, \boldsymbol{y}^{(t)}) \log p_{\boldsymbol{w}_t}(\boldsymbol{y}^{*(t)})\,|\,\boldsymbol{x}^{(t)}) \le \frac{2}{T}\sum_{t=0}^{T-1} -r(\boldsymbol{x}^{(t)}, \boldsymbol{y}^{(t)}) \log p_{\boldsymbol{v}}(\boldsymbol{y}^{*(t)}\,|\,\boldsymbol{x}^{(t)}) + \frac{2N\|\boldsymbol{v} - \boldsymbol{w}_0\|_2^2}{T} \tag{10}$$

for any $\boldsymbol{v} \in \mathbb{R}^D$. Next, we take expectation over the randomness of $\boldsymbol{y}^{(0)}, \ldots, \boldsymbol{y}^{(T-1)}$ from both sides. In the above, the only random quantities are $(\boldsymbol{y}^{(t)})_{t=0}^{T-1}$ and $(\boldsymbol{w}_t)_{t=0}^{T-1}$, as we are considering an arbitrary sequence of context $(\boldsymbol{x}^{(t)})_{t=0}^{T-1}$. Note that $\boldsymbol{w}_t$ is independent of the draw of $\boldsymbol{y}^{(t)}$, therefore

$$
\begin{aligned}
\mathbb{E}\Big[r(\boldsymbol{x}^{(t)}, \boldsymbol{y}^{(t)}) \log p_{\boldsymbol{w}_t}(\boldsymbol{y}^{*(t)}\,|\,\boldsymbol{x}^{(t)})\Big] &= \mathbb{E}\Big[\log p_{\boldsymbol{w}_t}(\boldsymbol{y}^*(\boldsymbol{x}^{(t)})\,|\,\boldsymbol{x}^{(t)}) \mathbb{E}\Big[r(\boldsymbol{x}^{(t)}, \boldsymbol{y}^{(t)})\,|\,\boldsymbol{w}_t\Big]\Big] \\
&= \mathbb{E}\Big[q_t(\boldsymbol{y}^{*(t)}\,|\,\boldsymbol{x}^{(t)}) \log p_{\boldsymbol{w}_t}(\boldsymbol{y}^{*(t)}\,|\,\boldsymbol{x}^{(t)})\Big].
\end{aligned}
$$

In the above, we used the fact that $\mathbb{E}\big[r(\boldsymbol{x}^{(t)}, \boldsymbol{y}^{(t)}) \,|\, \boldsymbol{w}_t\big] = q_t(\boldsymbol{y}^{*(t)} \,|\, \boldsymbol{x}^{(t)})$. By plugging this into (10), we have

$$\mathbb{E}\left[\frac{1}{T}\sum_{t=0}^{T-1} -q_t(\boldsymbol{y}^{*(t)} \,|\, \boldsymbol{x}^{(t)}) \log p_{\boldsymbol{w}_t}(\boldsymbol{y}^{*(t)} \,|\, \boldsymbol{x}^{(t)})\right] \leq \frac{2}{T}\sum_{t=0}^{T-1} -q_t(\boldsymbol{y}^{*(t)} \,|\, \boldsymbol{x}^{(t)}) \log p_{\boldsymbol{v}}(\boldsymbol{y}^{*(t)} \,|\, \boldsymbol{x}^{(t)}) + \frac{2N\|\boldsymbol{v} - \boldsymbol{w}_0\|_2^2}{T}$$

$$\leq -\frac{2}{T}\sum_{t=0}^{T-1} \log p_{\boldsymbol{v}}(\boldsymbol{y}^{*(t)} \,|\, \boldsymbol{x}^{(t)}) + \frac{2N\|\boldsymbol{v} - \boldsymbol{w}_0\|_2^2}{T}.$$

Let $\boldsymbol{v}^* = R_v \boldsymbol{w}^*$ where $\boldsymbol{w}^*$ is the unit-norm vector of Assumption 2.1, and $r$ is to be chosen later. Then, for any $\boldsymbol{x}$,

$$-\log p_{\boldsymbol{v}}(\boldsymbol{y}^* \,|\, \boldsymbol{x}) = -\sum_{i=1}^{N} \log p_{\boldsymbol{v}}(y_i^* \,|\, \boldsymbol{x}, \boldsymbol{y}_{1:i-1}^*)$$

$$= \sum_{i=1}^{N} \log\left(1 + \sum_{y' \neq y_i^*} e^{R_v \langle \boldsymbol{w}^*, \phi(\boldsymbol{x}, \boldsymbol{y}_{1:i-1}^*, y') - \phi(\boldsymbol{x}, \boldsymbol{y}_{1:i}^*) \rangle}\right)$$

$$\leq Nke^{-R_v\gamma}.$$

Therefore,

$$\mathbb{E}_{\boldsymbol{y}^{(0)},\ldots,\boldsymbol{y}^{(T-1)}}\left[\frac{1}{T}\sum_{t=0}^{T-1} -q_t \log p_{\boldsymbol{w}_t}(\boldsymbol{y}^{*(t)} \,|\, \boldsymbol{x}^{(t)})\right] \leq 2Nke^{-R_v\gamma} + \frac{4N\|\boldsymbol{v}\|_2^2 + 4N\|\boldsymbol{w}_0\|_2^2}{T}$$

$$\leq 2Nke^{-R_v\gamma} + \frac{4NR_v^2 + 4N\|\boldsymbol{w}_0\|_2^2}{T}.$$

We can balance the first two terms by choosing $R_v = \frac{1}{\gamma}\log(kT\gamma^2)$, and obtain

$$\mathbb{E}_{\boldsymbol{y}^{(0)},\ldots,\boldsymbol{y}^{(T-1)}}\left[\frac{1}{T}\sum_{t=0}^{T-1} -q_t \log p_{\boldsymbol{w}_t}(\boldsymbol{y}^{*(t)} \,|\, \boldsymbol{x}^{(t)})\right] \leq \frac{6N}{\gamma^2 T}\log^2(kT\gamma^2) + \frac{2N\|\boldsymbol{w}_0\|_2^2}{T}$$

$$\leq \frac{10N\log^2(kT\gamma^2)}{\gamma^2 T},$$

which completes the proof. $\qquad\square$

To finish the proof of Theorem 3.1, we can employ a simple online-to-batch conversion.

*Proof of Theorem 3.1.* By taking expectation from both sides of (9), we have

$$\mathbb{E}\left[\frac{1}{T}\sum_{t=0}^{T-1} -q_t(\boldsymbol{y}^*(\boldsymbol{x}_t) \,|\, \boldsymbol{x}_t) \log p_{\boldsymbol{w}_t}(\boldsymbol{y}^*(\boldsymbol{x}_t) \,|\, \boldsymbol{x}_t)\right] \leq \frac{10N\log(kT\gamma^2)^2}{\gamma^2 T}.$$

Since $\boldsymbol{x}_t$ is independent of $\boldsymbol{w}_t$ and $q_t$, we can replace $\boldsymbol{x}_t$ with the test sample $\boldsymbol{x}$ that has the same law as $\boldsymbol{x}_t$ and is independent from all $(\boldsymbol{w}_t)$ and $(q_t)$ while preserving the expectation above, and obtain

$$\mathbb{E}\left[\frac{1}{T}\sum_{t=0}^{T-1} -q_t(\boldsymbol{y}^*(\boldsymbol{x}) \,|\, \boldsymbol{x}) \log p_{\boldsymbol{w}_t}(\boldsymbol{y}^*(\boldsymbol{x}) \,|\, \boldsymbol{x})\right] \leq \frac{10N\log(kT\gamma^2)^2}{\gamma^2 T}.$$

Let

$$\mathcal{L} := \frac{1}{T}\sum_{t=0}^{T-1} -q_t(\boldsymbol{y}^*(\boldsymbol{x}) \,|\, \boldsymbol{x}) \log p_{\boldsymbol{w}_t}(\boldsymbol{y}^*(\boldsymbol{x}) \,|\, \boldsymbol{x}).$$

Using the law of total expectation and the fact that $\mathcal{L} \geq 0$, we have $\mathbb{E}[\mathcal{L} \,|\, \mathcal{E}_\alpha]\mathbb{P}(\mathcal{E}_\alpha) \leq \mathbb{E}[\mathcal{L}]$. As a result,

$$\mathbb{E}\left[\frac{1}{T}\sum_{t=0}^{T-1} -q_t(\boldsymbol{y}^*(\boldsymbol{x}) \,|\, \boldsymbol{x}) \log p_{\boldsymbol{w}_t}(\boldsymbol{y}^*(\boldsymbol{x}) \,|\, \boldsymbol{x}) \,|\, \mathcal{E}_\alpha\right] \leq \frac{10N\log(kT\gamma^2)^2}{\pi_\alpha \gamma^2 T}.$$

Note that on $\mathcal{E}_\alpha$, we have $q_t(\boldsymbol{y}^*(\boldsymbol{x}) \,|\, \boldsymbol{x}) \geq \alpha$ for all $t$, thus

$$\mathbb{E}\left[\frac{1}{T}\sum_{t=0}^{T-1} -\log p_{\boldsymbol{w}_t}(\boldsymbol{y}^*(\boldsymbol{x}) \,|\, \boldsymbol{x}) \,|\, \mathcal{E}_\alpha\right] \leq \frac{10N \log(kT\gamma^2)^2}{\pi_\alpha \alpha \gamma^2 T}.$$

Using the convexity of negative log-likelihood and the Jensen inequality, we obtain

$$\mathbb{E}\left[-\log p_{\bar{\boldsymbol{w}}_T^{\mathrm{PG}}}(\boldsymbol{y}^*(\boldsymbol{x}) \,|\, \boldsymbol{x}) \,|\, \mathcal{E}_\alpha\right] \leq \frac{10N \log(kT\gamma^2)^2}{\pi_\alpha \alpha \gamma^2 T}.$$

Finally, we can use the inequality $1 - x \leq -\log x$ for $x > 0$ to complete the proof. $\qquad\square$

### A.2. Proof of Theorem 3.2

For this part, we are going to use the second case of Proposition A.1. To do so, we will take advantage of the structure of the log-likelihood gradient characterized by the following lemma.

**Lemma A.6.** *Let $R = \sup_{\boldsymbol{w},\boldsymbol{x},\boldsymbol{y},i}\|\phi(\boldsymbol{x},\boldsymbol{y}_{1:i})\|_2$. Then, for any $\boldsymbol{w} \in \mathbb{R}^D$, $\boldsymbol{x} \in \mathbb{R}^d$, $\boldsymbol{y} \in \mathcal{Y}^N$, and $i \in [N]$, we have*

$$\left\|\nabla_{\boldsymbol{w}} \log p_{\boldsymbol{w}}(y_i \,|\, \boldsymbol{x}, \boldsymbol{y}_{1:i-1})\right\|_2 \leq -2R \log p_{\boldsymbol{w}}(y_i \,|\, \boldsymbol{x}, \boldsymbol{y}_{1:i-1}).$$

*Therefore, by the triangle inequality, we also have*

$$\left\|\nabla_{\boldsymbol{w}} \log p_{\boldsymbol{w}}(\boldsymbol{y} \,|\, \boldsymbol{x})\right\|_2 \leq -2R \log p_{\boldsymbol{w}}(\boldsymbol{y} \,|\, \boldsymbol{x}).$$

*Proof.* Let us define $p_y = p_{\boldsymbol{w}}(y \,|\, \boldsymbol{x}, \boldsymbol{y}_{1:i-1})$. Then,

$$\begin{aligned}
\left\|\nabla_{\boldsymbol{w}} \log p_{\boldsymbol{w}}(y_i \,|\, \boldsymbol{x}, \boldsymbol{y}_{1:i-1})\right\|_2 &= \left\|\phi(\boldsymbol{x},\boldsymbol{y}_{1:i-1},y_i) - \sum_{y=1}^{k}\phi(\boldsymbol{x},\boldsymbol{y}_{1:i-1},y)p_y\right\|_2 \\
&\leq (1-p_{y_i})\left\|\phi(\boldsymbol{x},\boldsymbol{y}_{1:i-1},y_i)\right\|_2 + \sum_{y \neq y_i} p_y\left\|\phi(\boldsymbol{x},\boldsymbol{y}_{1:i-1},y)\right\|_2 \\
&\leq 2R(1-p_{y_i}) \\
&\leq -2R \log p_{y_i},
\end{aligned}$$

where the last inequality used the fact that $1 - x \leq -\log x$ for $x > 0$. $\qquad\square$

Similar to the previous section, we first prove a more general online guarantee for PG with adaptive learning rate.

**Proposition A.7.** *Let $(\boldsymbol{w}_t)$ denote the (PG-OR) iterates. Suppose the sequence $(\boldsymbol{x}^{(t)}, \boldsymbol{y}^*(\boldsymbol{x}^{(t)}))$ is chosen arbitrarily (in particular not necessarily i.i.d.), with the only constraint being Assumption 2.1. Assume $\eta_t = 1/(2(\lambda + \|\nabla \log p_{\boldsymbol{w}_t}(\boldsymbol{y}^{(t)} \,|\, \boldsymbol{x}^{(t)})\|_2))$ for all $t$ and some $\lambda > 0$, and $\|\boldsymbol{w}_0\|_2 \leq 1/\gamma \cdot \log(NkT\gamma^2/\lambda)$. Then,*

$$\mathbb{E}_{\boldsymbol{y}^{(0)},\ldots,\boldsymbol{y}^{(T-1)}}\left[\frac{1}{T}\sum_{t=0}^{T-1} q_t(\boldsymbol{y}^*(\boldsymbol{x}^{(t)}) \,|\, \boldsymbol{x}^{(t)})\min\left(\tfrac{\lambda}{2}, -\log p_{\boldsymbol{w}_t}(\boldsymbol{y}^*(\boldsymbol{x}^{(t)}) \,|\, \boldsymbol{x}^{(t)})\right)\right] \leq \frac{20\lambda \log(NkT\gamma^2/\lambda)^2}{\gamma^2 T}. \quad (11)$$

*Proof.* We can make a similar observation as in Proposition A.5 about the (PG-OR) update, and rewrite it as

$$\begin{aligned}
\boldsymbol{w}_{t+1} &= \boldsymbol{w}_t + \eta_t r(\boldsymbol{x}^{(t)}, \boldsymbol{y}^{(t)})\nabla \log p_{\boldsymbol{w}_t}(\boldsymbol{y}^{*(t)} \,|\, \boldsymbol{x}^{(t)}) \\
&= \boldsymbol{w}_t - \eta_t \nabla \ell_t(\boldsymbol{w}_t)
\end{aligned}$$

for $\ell_t(\boldsymbol{w}_t) := -r(\boldsymbol{x}^{(t)}, \boldsymbol{y}^{(t)})\log p_{\boldsymbol{w}_t}(\boldsymbol{y}^{*(t)} \,|\, \boldsymbol{x}^{(t)})$, where we recall $\boldsymbol{y}^{*(t)} := \boldsymbol{y}^*(\boldsymbol{x}^{(t)})$. Note that $\ell_t$ is convex, and by Lemma A.6, it satisfies $\|\nabla \ell_t\|_2 \leq 2\ell_t$. For simplicity, define $r_t := r(\boldsymbol{x}^{(t)}, \boldsymbol{y}^{(t)})$ and $\tilde{\ell}_t(\boldsymbol{w}) := -\log p_{\boldsymbol{w}}(\boldsymbol{y}^{*(t)} \,|\, \boldsymbol{x}^{(t)})$. Recall the

learning rate $\eta_t = (\lambda + \left\|\nabla\tilde{\ell}_t(\boldsymbol{w}_t)\right\|_2)^{-1}/2$, and note that $\eta_t$ is only important when $r_t = 1$, in which case $\tilde{\ell}_t = \ell_t$, thus $\eta_t = (\lambda + \|\nabla\ell_t(\boldsymbol{w}_t)\|_2)^{-1}/2$. We can therefore invoke the second case of Proposition A.1 and write

$$\frac{1}{T}\sum_{t=0}^{T-1}\frac{r_t\tilde{\ell}_t(\boldsymbol{w}_t)}{1+2\lambda^{-1}\tilde{\ell}_t(\boldsymbol{w}_t)} \leq \frac{1}{T}\sum_{t=0}^{T-1}\frac{r_t\tilde{\ell}_t(\boldsymbol{w}_t)}{1+2\lambda^{-1}r_t\tilde{\ell}_t(\boldsymbol{w}_t)} \leq \frac{2}{T}\sum_{t=0}^{T-1}r_t\tilde{\ell}_t(\boldsymbol{v}) + \frac{2\lambda\|\boldsymbol{v}-\boldsymbol{w}_0\|_2^2}{T}$$

for any $\boldsymbol{v} \in \mathbb{R}^D$. Similar to the proof of Proposition A.5, we can take expectation over the randomness of $\boldsymbol{y}^{(0)},\ldots,\boldsymbol{y}^{(T-1)}$ from both sides above, and use the independence of $\boldsymbol{y}^{(t)}$ and $\boldsymbol{w}_t$ as well as the fact that $\mathbb{E}\big[r(\boldsymbol{x}^{(t)},\boldsymbol{y}^{(t)})\,\big|\,\boldsymbol{w}_t\big] = q_t(\boldsymbol{y}^{*(t)}\,|\,\boldsymbol{x}^{(t)}) =: q_t$ to write

$$\mathbb{E}_{\boldsymbol{y}^{(0)},\ldots,\boldsymbol{y}^{(T-1)}}\left[\frac{1}{T}\sum_{t=0}^{T-1}\frac{q_t\tilde{\ell}_t(\boldsymbol{w}_t)}{1+2\lambda^{-1}\tilde{\ell}_t(\boldsymbol{w}_t)}\right] \leq \frac{2}{T}\sum_{t=0}^{T-1}q_t\tilde{\ell}_t(\boldsymbol{v}) + \frac{2\lambda\|\boldsymbol{v}-\boldsymbol{w}_0\|_2^2}{T}$$

Note that we have $\frac{\tilde{\ell}_t}{1+2\lambda^{-1}\tilde{\ell}_t} = \frac{(\lambda/2)\tilde{\ell}_t}{\lambda/2+\tilde{\ell}_t} \geq \frac{1}{2}\min(\lambda/2,\tilde{\ell}_t)$. Thus,

$$\mathbb{E}_{\boldsymbol{y}^{(0)},\ldots,\boldsymbol{y}^{(T-1)}}\left[\frac{1}{T}\sum_{t=0}^{T-1}q_t\min(\tfrac{\lambda}{2},\tilde{\ell}_t)\right] \leq \frac{4}{T}\sum_{t=0}^{T-1}q_t\tilde{\ell}_t(\boldsymbol{v}) + \frac{4\lambda\|\boldsymbol{v}-\boldsymbol{w}_0\|_2^2}{T}$$

$$\leq \frac{4}{T}\sum_{t=0}^{T-1}\tilde{\ell}_t(\boldsymbol{v}) + \frac{4\lambda\|\boldsymbol{v}-\boldsymbol{w}_0\|_2^2}{T}.$$

Recall from the proof of Proposition A.5 that for $\boldsymbol{v} = R_v\boldsymbol{w}^*$ we have $\tilde{\ell}_t(\boldsymbol{v}) \leq Nke^{-R_v\gamma}$. Therefore,

$$\mathbb{E}_{\boldsymbol{y}^{(0)},\ldots,\boldsymbol{y}^{(T-1)}}\left[\frac{1}{T}\sum_{t=0}^{T-1}q_t\min(\tfrac{\lambda}{2},\tilde{\ell}_t(\boldsymbol{w}_t))\right] \leq 4Nke^{-R_v\gamma^2} + \frac{4\lambda\|\boldsymbol{v}-\boldsymbol{w}_0\|_2^2}{T}$$

$$\leq 4Nke^{-R_v\gamma^2} + \frac{4\lambda\|\boldsymbol{v}\|_2^2 + 4\lambda\|\boldsymbol{w}_0\|_2^2}{T}$$

$$\leq 4Nke^{-R_v\gamma^2} + \frac{8\lambda R_v^2 + 8\lambda\|\boldsymbol{w}_0\|_2^2}{T}.$$

By choosing $R_v = 1/\gamma \cdot \log(NkT\gamma^2/\lambda)$, we obtain

$$\mathbb{E}_{\boldsymbol{y}^{(0)},\ldots,\boldsymbol{y}^{(T-1)}}\left[\frac{1}{T}\sum_{t=0}^{T-1}q_t\min(\tfrac{\lambda}{2},\tilde{\ell}_t(\boldsymbol{w}_t))\right] \leq \frac{20\lambda\log(NkT\gamma^2/\lambda)^2}{\gamma^2 T},$$

which completes the proof. $\qquad\square$

Similar to the previous section, we can use an online-to-batch conversion to prove Theorem 3.2.

*Proof of Theorem 3.2.* Note that by taking expectation from both sides of (11) and replacing $\boldsymbol{x}_t$ with $\boldsymbol{x}$ (as they are both independent of $\boldsymbol{w}_t$, $q_t$ and have the same law) we obtain

$$\mathbb{E}\left[\frac{1}{T}\sum_{t=0}^{T-1}q_t(\boldsymbol{y}^*(\boldsymbol{x})\,|\,\boldsymbol{x})\min(\tfrac{\lambda}{2},-\log p_{\boldsymbol{w}_t}(\boldsymbol{y}^*(\boldsymbol{x})\,|\,\boldsymbol{x}))\right] \lesssim \frac{\lambda\log(NkT\gamma^2/\lambda)^2}{\gamma^2 T}.$$

Using the inequality $1 - x \leq -\log x$ for all $x$, we obtain

$$\mathbb{E}\left[\frac{1}{T}\sum_{t=0}^{T-1}q_t(\boldsymbol{y}^*(\boldsymbol{x})\,|\,\boldsymbol{x})\min(\tfrac{\lambda}{2},1-p_{\boldsymbol{w}_t}(\boldsymbol{y}^*(\boldsymbol{x})\,|\,\boldsymbol{x}))\right] \lesssim \frac{\lambda\log(NkT\gamma^2/\lambda)^2}{\gamma^2 T}.$$

Now by choosing $\lambda = 2$, we can remove $\min$ and write

$$\mathbb{E}\left[\frac{1}{T}\sum_{t=0}^{T-1} q_t(\boldsymbol{y}^*(\boldsymbol{x})\,|\,\boldsymbol{x})(1 - p_{\boldsymbol{w}_t}(\boldsymbol{y}^*(\boldsymbol{x})\,|\,\boldsymbol{x}))\right] \lesssim \frac{\log(NkT\gamma^2)^2}{\gamma^2 T}.$$

Similar to the proof of Theorem 3.1, we can use the non-negativity of the LHS above along with the law of total expectation to write

$$\mathbb{E}\left[\frac{1}{T}\sum_{t=0}^{T-1} q_t(\boldsymbol{y}^*(\boldsymbol{x})\,|\,\boldsymbol{x})(1 - p_{\boldsymbol{w}_t}(\boldsymbol{y}^*(\boldsymbol{x})\,|\,\boldsymbol{x}))\,|\,\mathcal{E}_\alpha\right] \lesssim \frac{\log(NkT\gamma^2)^2}{\pi_\alpha\gamma^2 T}.$$

Observing that $q_t(\boldsymbol{y}^*(\boldsymbol{x})\,|\,\boldsymbol{x}) \geq \alpha$ on $\pi_\alpha$ concludes the proof. $\qquad\square$

### A.3. Proof of Corollary 3.4

Let $\tilde{\mathcal{E}}_{\tilde{\alpha}} := \{\boldsymbol{x} : q(\boldsymbol{y}^*(\boldsymbol{x})\,|\,\boldsymbol{x}) \geq \tilde{\alpha}\}$. Then, for any $\tilde{\alpha} \in (0, 1]$,

$$\mathbb{E}[1 - p_{\boldsymbol{w}_\tau}(\boldsymbol{y}^*(\boldsymbol{x})\,|\,\boldsymbol{x})] = \mathbb{E}\left[1 - p_{\boldsymbol{w}_\tau}(\boldsymbol{y}^*(\boldsymbol{x})\,|\,\boldsymbol{x})\,|\,\tilde{\mathcal{E}}_{\tilde{\alpha}}\right]\mathbb{P}\left(\tilde{\mathcal{E}}_{\tilde{\alpha}}\right) + \mathbb{E}\left[1 - p_{\boldsymbol{w}_\tau}(\boldsymbol{y}^*(\boldsymbol{x})\,|\,\boldsymbol{x})\,|\,\tilde{\mathcal{E}}_{\tilde{\alpha}}^c\right]\mathbb{P}\left(\tilde{\mathcal{E}}_{\tilde{\alpha}}^c\right). \qquad (12)$$

We will consider two approaches to choose $\tilde{\alpha}$ for bounding this expression.

First, note that

$$\mathbb{P}\left(\tilde{\mathcal{E}}_{\tilde{\alpha}}^c\right) = \mathbb{P}\left(\tfrac{1}{2}q_0(\boldsymbol{y}^*(\boldsymbol{x})\,|\,\boldsymbol{x}) + \tfrac{k^{-N}}{2} < \tilde{\alpha}\right) \leq \mathbb{P}(q_0(\boldsymbol{y}^*(\boldsymbol{x})\,|\,\boldsymbol{x}) \leq 2\tilde{\alpha}) = 1 - \pi_{2\tilde{\alpha}}$$

where we recall $\pi_\alpha := \mathbb{P}(q_0(\boldsymbol{y}^*(\boldsymbol{x})\,|\,\boldsymbol{x}) \geq \alpha)$. Then, we can rewrite (12) as

$$\mathbb{E}[1 - p_{\boldsymbol{w}_\tau}(\boldsymbol{y}^*(\boldsymbol{x})\,|\,\boldsymbol{x})] \leq \mathbb{E}\left[1 - p_{\boldsymbol{w}_\tau}(\boldsymbol{y}^*(\boldsymbol{x})\,|\,\boldsymbol{x})\,|\,\tilde{\mathcal{E}}_{\tilde{\alpha}}\right]\mathbb{P}\left(\tilde{\mathcal{E}}_{\tilde{\alpha}}\right) + 1 - \pi_{2\tilde{\alpha}}. \qquad (13)$$

By choosing $\tilde{\alpha} = \tfrac{1}{2}\mathcal{Q}_{q_0}\left((1 - \tfrac{1}{\log(1/\varepsilon)})\varepsilon\right)$, we have $1 - \pi_{2\tilde{\alpha}} \leq \varepsilon(1 - \tfrac{1}{\log(1/\varepsilon)})$. Combined with the bound from Theorem 3.2 for the first term, we obtain

$$\mathbb{E}[1 - p_{\boldsymbol{w}_\tau}(\boldsymbol{y}^*(\boldsymbol{x})\,|\,\boldsymbol{x})] \leq \tilde{\mathcal{O}}\left(\frac{\tilde{\alpha}^{-1}}{\gamma^2 T}\right) + 1 - \pi_{2\tilde{\alpha}}$$

$$\leq \tilde{\mathcal{O}}\left(\frac{\mathcal{Q}_{q_0}\left((1 - o(1))\varepsilon\right)^{-1}}{\gamma^2 T}\right) + \varepsilon(1 - \tfrac{1}{\log\varepsilon}),$$

where $o(1)$ is hiding the $1/\log(1/\varepsilon)$ factor. By letting $T = \tilde{\mathcal{O}}(\tilde{\alpha}^{-1}/(\gamma^2 T\varepsilon))$, we can derive the first term below $\varepsilon/\log(1/\varepsilon)$, and the total expectation below $\varepsilon$. This gives us a sufficient number of iterations

$$T = \tilde{\mathcal{O}}\left(\frac{\mathcal{Q}_{q_0}\left((1 - o(1))\varepsilon\right)^{-1}}{\gamma^2\varepsilon}\right).$$

For the second approach, note that for $\tilde{\alpha} = \tfrac{k^{-N}}{2}$ we have $\mathbb{P}\left(\tilde{\mathcal{E}}_{\tilde{\alpha}}\right) = 1$. Thus, we can directly use Theorem 3.2 to write $\mathbb{E}[1 - p_{\boldsymbol{w}_\tau}(\boldsymbol{y}^*(\boldsymbol{x})\,|\,\boldsymbol{x})] \leq \tilde{\mathcal{O}}(k^N/(\gamma^2 T))$. Hence, another sufficient number of iterations is given by $T = \tilde{\mathcal{O}}(k^N/(\gamma^2\varepsilon))$. We conclude the proof by considering the minimum between the two approaches. $\qquad\square$

### A.4. Proof of Corollary 3.6

Recall the definitions

$$\mathcal{E}_\alpha := \{\boldsymbol{x} : q_0(\boldsymbol{y}^*(\boldsymbol{x})\,|\,\boldsymbol{x}) \geq \alpha\},$$
$$\tilde{\mathcal{E}}_{\tilde{\alpha}} := \{\boldsymbol{x} : q(\boldsymbol{y}^*(\boldsymbol{x})\,|\,\boldsymbol{x}) \geq \tilde{\alpha}\}.$$

From Theorem 3.2,

$$\mathbb{E}\left[1 - p_{\boldsymbol{w}_\tau}(\boldsymbol{y}^*(\boldsymbol{x})\,|\,\boldsymbol{x})\,|\,\tilde{\mathcal{E}}_{\tilde{\alpha}}\right]\mathbb{P}\left(\tilde{\mathcal{E}}_{\tilde{\alpha}}\right) \leq \tilde{\mathcal{O}}\left(\frac{1}{\tilde{\alpha}\gamma^2 T}\right).$$

By inserting the above bound into (12) we obtain

$$\mathbb{E}[1 - p_{\boldsymbol{w}_\tau}(\boldsymbol{y}^*(\boldsymbol{x}) \,|\, \boldsymbol{x})] \le \tilde{\mathcal{O}}\left(\frac{1}{\tilde{\alpha}\gamma^2 T}\right) + \mathbb{P}\left(\tilde{\mathcal{E}}_{\tilde{\alpha}}^c\right). \tag{14}$$

Let

$$m := \min\left(\left\lceil \mathcal{Q}_{q_0}\left((1 - \tfrac{1}{\log(1/\varepsilon)})\varepsilon\right)^{-1}, k^N\right\rceil\right).$$

We break down the argument based on which term attains the minimum.

- Suppose the minimum is attained by the first term. Since $q(\boldsymbol{y}^*(\boldsymbol{x}) \,|\, \boldsymbol{x}) \ge 1 - (1 - q_0(\boldsymbol{y}^*(\boldsymbol{x}) \,|\, \boldsymbol{x})/2)^m$, with $\alpha = m^{-1}$ we have $\tilde{\alpha} = 1 - e^{-1/2}$. Further,

$$\mathbb{P}(\mathcal{E}_\alpha^c) = 1 - \pi_\alpha \le 1 - \pi_{m^{-1}} = \mathcal{Q}_{q_0}^{-1}(m^{-1}) \le (1 - \tfrac{1}{\log(1/\varepsilon)})\varepsilon.$$

Since $\mathbb{P}\left(\tilde{\mathcal{E}}_{\tilde{\alpha}}^c\right) \le \mathbb{P}(\mathcal{E}_\alpha^c)$, from (14) we have

$$\mathbb{E}[1 - p_{\boldsymbol{w}_\tau}(\boldsymbol{y}^*(\boldsymbol{x}) \,|\, \boldsymbol{x})] \le \tilde{\mathcal{O}}\left(\frac{1}{\gamma^2 T}\right) + \varepsilon(1 - \tfrac{1}{\log(1/\varepsilon)}).$$

Thus $T = \tilde{\mathcal{O}}(1/(\gamma^2\varepsilon))$ is sufficient to ensure the first term is less than $\varepsilon/\log(1/\varepsilon)$ and the overall expectation is less than $\varepsilon$.

- Suppose instead that the minimum is instead attained by $k^N$. Note that

$$q(\boldsymbol{y}^*(\boldsymbol{x}) \,|\, \boldsymbol{x}) \ge 1 - (1 - k^{-N}/2)^m \ge 1 - e^{-1/2}.$$

Hence, this time for $\tilde{\alpha} = 1 - e^{-1/2}$ we have $\mathbb{P}\left(\tilde{\mathcal{E}}_{\tilde{\alpha}}^c\right) = 0$. As a result, (14) reads

$$\mathbb{E}[1 - p_{\boldsymbol{w}_\tau}(\boldsymbol{y}^*(\boldsymbol{x}) \,|\, \boldsymbol{x})] \le \tilde{\mathcal{O}}\left(\frac{1}{\gamma^2 T}\right),$$

and taking $T = \tilde{\mathcal{O}}(1/(\gamma^2\varepsilon))$ is once again sufficient.

For the number of reward queries, we observe that every time $\boldsymbol{y}^{(t)} = \boldsymbol{y}^*(\boldsymbol{x}^{(t)})$ for $\boldsymbol{y}^{(t)} \sim p_{\boldsymbol{w}_t}$, Algorithm 1 makes one reward query, otherwise it makes at most $1 + m$ queries. The (PG-OR) also makes one reward query. Therefore

$$
\begin{aligned}
Q &= 2T + m\,\mathbb{E}\left[\sum_{t=1}^T \mathbb{1}[\hat{\boldsymbol{y}}^{(t)} \ne \boldsymbol{y}^*(\boldsymbol{x}^{(t)})]\right] \\
&= 2T + m\sum_{t=1}^T \mathbb{E}\left[1 - p_{\boldsymbol{w}_t}(\boldsymbol{y}^*(\boldsymbol{x}^{(t)}) \,|\, \boldsymbol{x}^{(t)})\right] \\
&= 2T + m\sum_{t=1}^T \mathbb{E}[1 - p_{\boldsymbol{w}_t}(\boldsymbol{y}^*(\boldsymbol{x}) \,|\, \boldsymbol{x})] & \text{(Using independence of } \boldsymbol{x}^{(t)} \text{ from } \boldsymbol{w}_t) \\
&\le 2T + mT\mathbb{P}\left(\tilde{\mathcal{E}}_{\tilde{\alpha}}^c\right) + m\tilde{\mathcal{O}}(\gamma^{-2}) & \text{(Using (14) and } \tilde{\alpha} = 1 - e^{-1/2}).
\end{aligned}
$$

Recall that $\mathbb{P}\left(\tilde{\mathcal{E}}_{\tilde{\alpha}}^c\right) \le \varepsilon(1 - o(1))$ and $T = \tilde{\mathcal{O}}(1/(\gamma^2\varepsilon))$. Therefore we can combine the middle and last terms on the RHS above, and write

$$Q \le 2T + m\tilde{\mathcal{O}}(\gamma^{-2}) \le \tilde{\mathcal{O}}((m + \varepsilon^{-1})/\gamma^2),$$

finishing the proof. $\qquad\square$

## A.5. Fully On-Policy Policy Gradient

In this section, we will consider (PG-OR) when $q_t = p_{\boldsymbol{w}_t}$ for all $t$ starting from a base model $q_0 = p_{\boldsymbol{w}_0}$. We first state this result and then explain the intuition behind it, and how it is qualitatively related to the observations in Section 3. We then proceed to prove the result.

**Proposition A.8** (Fully Online PG). *Consider the* (PG-OR) *updates under Assumption 2.1, with a base policy $q_0 = p_{\boldsymbol{w}_0}$ and assume $\|\boldsymbol{w}_0\|_2 \lesssim \log(Tk\gamma^2)/\gamma$ for simplicity. Let $\tau \sim \mathrm{Unif}\{0, \dots, T-1\}$). For any $\delta \in (0, 1)$, with a proper choice of learning rate of order $\eta = \tilde{\Theta}(\gamma^2 \delta/N)$, after*

$$T = \tilde{\mathcal{O}}\left(\frac{N}{\gamma^4 \delta^2 \varepsilon} \cdot \min\left(1, \frac{1 - \mathbb{E}[q_0(\boldsymbol{y}^*(\boldsymbol{x}) \mid \boldsymbol{x})]}{\delta}\right)\right)$$

*we have $p_{\boldsymbol{w}_\tau}(\boldsymbol{y}^*(\boldsymbol{x}) \mid \boldsymbol{x}) \geq 1 - \varepsilon$ with probability at least $\mathbb{E}[q_0(\boldsymbol{y}^*(\boldsymbol{x}) \mid \boldsymbol{x})] - \delta$.*

We can observe that with a fixed probability of $\mathbb{E}[q_0(\boldsymbol{y}^*(\boldsymbol{x}) \mid \boldsymbol{x})] - \delta$, online PG can boost the likelihood to be $1 - \varepsilon$, with a rate that although suboptimal, still depends only linearly on the sequence length and polynomially on other parameters. Crucially, the role of the base model here is to determine the probability of success, which can at most be $\mathbb{E}[q_0(\boldsymbol{y}^*(\boldsymbol{x}) \mid \boldsymbol{x})]$. While the above proposition does not characterize which samples at test-time win this lottery ticket, Figure 1 hints that these are samples where the initial likelihood is non-trivial. Further, we expect the above rate to be an artifact of our analysis and possibly improvable.

To prove the above proposition, we make a number of observations.

**Lemma A.9.** *Recall $\sup_{\boldsymbol{w}, \boldsymbol{x}, \boldsymbol{y}, i} \|\phi(\boldsymbol{x}, \boldsymbol{y}_{1:i})\|_2 \leq 1$. We then have $\sup_{\boldsymbol{w}, \boldsymbol{x}, \boldsymbol{y}} \|\nabla_{\boldsymbol{w}}^2 p_{\boldsymbol{w}}(\boldsymbol{y} \mid \boldsymbol{x})\|_2 \lesssim 1$.*

*Proof.* Define the shorthand notations

$$p := p_{\boldsymbol{w}}(\boldsymbol{y} \mid \boldsymbol{x}), \quad p_y^i := p_{\boldsymbol{w}}(y \mid \boldsymbol{x}, \boldsymbol{y}_{1:i-1}), \quad p^i := p_{\boldsymbol{w}}(y_i \mid \boldsymbol{x}, \boldsymbol{y}_{1:i-1}), \quad Y_i \sim p_{\boldsymbol{w}}(\cdot \mid \boldsymbol{x}, \boldsymbol{y}_{1:i-1}).$$

We have

$$\nabla^2 p = p \nabla \log p \nabla \log p^\top + p \nabla^2 \log p.$$

Therefore,

$$\|\nabla^2 p\|_2 \leq p \|\nabla \log p\|_2^2 + p \|\nabla^2 \log p\|_2$$

$$\leq p\left(\sum_{i=1}^N \|\nabla \log p^i\|_2\right)^2 + p \sum_{i=1}^N \|\nabla^2 \log p^i\|_2 \tag{15}$$

Further, from Lemma A.6 with $R = 1$, recall that $\|\nabla \log p^i\|_2 \leq -2 \log p^i$. Also, from the proof of Lemma A.4, recall that

$$\begin{aligned}
\|\nabla^2 \log p^i\|_2 &= \left\|\mathbb{E}_{Y_i}[\phi_{Y_i} \phi_{Y_i}^\top] - \mathbb{E}_{Y_i}[\phi_{Y_i}] \mathbb{E}[\phi_{Y_i}]^\top\right\|_2 \\
&\leq \mathbb{E}_{Y_i}\left[\|\phi_{Y_i}\|_2^2\right] - \|\mathbb{E}_{Y_i}[\phi_{Y_i}]\|_2^2 & \text{(Since } \|\mathrm{Cov}_{Y_i}(\phi_{Y_i})\|_2 \leq \mathrm{Tr}(\mathrm{Cov}_{Y_i}(\phi_{Y_i}))) \\
&= \mathbb{E}_{Y_i}\left[\|\phi_{Y_i} - \mathbb{E}_{Y_i}[\phi_{Y_i}]\|_2^2\right] \\
&\leq 2 \mathbb{E}_{Y_i}[\|\phi_{Y_i} - \mathbb{E}_{Y_i}[\phi_{Y_i}]\|_2] & \text{(Since } \|\phi_{Y_i} - \mathbb{E}_{Y_i}[\phi_{Y_i}]\|_2 \leq 2) \\
&\leq 2\|\phi_{y_i} - \mathbb{E}_{Y_i}[\phi_{Y_i}]\|_2 + 2 \sum_{y \neq y_i} p_y^i \|\phi_y - \mathbb{E}[Y_i]\phi_{Y_i}\|_2 \\
&\leq 4(1 - p_i) + 4 \sum_{y \neq y_i} p_y^i & \text{(From the proof of Lemma A.6)} \\
&\leq 8(1 - p_i) \\
&\leq -8 \log p_i,
\end{aligned}$$

where $\phi_y := \phi(\boldsymbol{x}, \boldsymbol{y}_{1:i-1}, y)$. Plugging these estimates back into (15) yields

$$\left\|\nabla^2 p\right\|_2 \lesssim p\left(\left(\sum_{i=1}^N -\log p_i\right)^2 - \sum_{i=1}^N \log p_i\right) = e^{-\log p}\left((-\log p)^2 - \log p\right).$$

Using the fact that the function $x \mapsto e^{-x}(x^2 + x)$ is bounded over $x \geq 0$ completes the proof. $\qquad\square$

To apply Proposition A.5, we need to lower bound $q_\tau$ where $\tau \sim \text{Unif}\{0, \ldots, T-1\}$. In the case of on-policy PG, this means lower bounding $p_{\boldsymbol{w}_\tau}$ itself, which is the objective of Proposition A.5. We thus need another tool to lower bound $p_{\boldsymbol{w}_\tau}$, and use this to achieve an improved conditional lower bound on $p_{\boldsymbol{w}_\tau}$ using A.5. We establish this lower bound by using the fact that on-policy PG is effectively doing stochastic gradient ascent on the function $\boldsymbol{w} \mapsto \mathbb{E}[p_{\boldsymbol{w}}(\boldsymbol{y}^*(\boldsymbol{x}) \mid \boldsymbol{x})]$. Therefore, this quantity should not decrease significantly from its initial value along the PG dynamics if the learning rate is sufficiently small. This fact is established by the following lemma.

**Lemma A.10.** *Let $(\boldsymbol{w}_t)$ denote the on-policy (PG-OR) iterates where $q_t = p_{\boldsymbol{w}_t}$, initialized from the base policy $p_{\boldsymbol{w}_0} = q_0$. Suppose Assumption 2.1 holds and $\|\boldsymbol{w}_0\|_2 \lesssim \log(Tk\gamma^2)/\gamma^2$ for simplicity. Define $\tau \sim \text{Unif}(\{0, \ldots, T-1\})$. Then, for any constant learning rate $\eta \leq 1/(2N)$, we have*

$$\mathbb{E}[p_{\boldsymbol{w}_\tau}(\boldsymbol{y}^*(\boldsymbol{x}) \mid \boldsymbol{x})] \geq \mathbb{E}[q_0(\boldsymbol{y}^*(\boldsymbol{y}^*(\boldsymbol{x}) \mid \boldsymbol{x})] - \frac{C\eta N \log(Tk\gamma^2)^2}{\gamma^2},$$

*where $C > 0$ is an absolute constant.*

*Proof.* Let $F(\boldsymbol{w}) := -\mathbb{E}[p_{\boldsymbol{w}}(\boldsymbol{y}^*(\boldsymbol{x}) \mid \boldsymbol{x})]$ and recall $\ell_t(\boldsymbol{w}) := -r(\boldsymbol{x}^{(t)}, \boldsymbol{y}^{(t)}) \log p_{\boldsymbol{w}}(\boldsymbol{y}^*(\boldsymbol{x}^{(t)}) \mid \boldsymbol{x}^{(t)})$. Also define $p_t := p_{\boldsymbol{w}_t}(\boldsymbol{y}^*(\boldsymbol{x}^{(t)}) \mid \boldsymbol{x}^{(t)})$ and $r_t := r(\boldsymbol{x}^{(t)}, \boldsymbol{y}^{(t)})$ for brevity. Recall that using the 0-1 structure of the reward, we can write (PG-OR) as

$$\boldsymbol{w}_{t+1} = \boldsymbol{w}_t - \eta \nabla \ell_t(\boldsymbol{w}_t).$$

Then

$$\mathbb{E}[\nabla \ell_t(\boldsymbol{w}_t) \mid \boldsymbol{w}_t] = \mathbb{E}\left[r(\boldsymbol{x}^{(t)}, \boldsymbol{y}^{(t)})\nabla \log p_{\boldsymbol{w}_t}(\boldsymbol{y}^*(\boldsymbol{x}^{(t)}) \mid \boldsymbol{x}^{(t)}) \mid \boldsymbol{w}_t\right] = \nabla F(\boldsymbol{w}_t),$$

as expected from policy gradient, where we used $\mathbb{E}\left[r(\boldsymbol{x}^{(t)}, \boldsymbol{y}^{(t)}) \mid \boldsymbol{x}^{(t)}, \boldsymbol{w}_t\right] = p_{\boldsymbol{w}_t}(\boldsymbol{y}^*(\boldsymbol{x}^{(t)}) \mid \boldsymbol{x}^{(t)})$. Further, Lemma A.9 proves that $\left\|\nabla^2 F(\boldsymbol{w})\right\|_2 \leq L$ where $L = \mathcal{O}(1)$ for all $\boldsymbol{w}$. Hence, using the smoothness of $F$, we can write

$$F(\boldsymbol{w}_{t+1}) \leq F(\boldsymbol{w}_t) - \langle F(\boldsymbol{w}_t), \eta \nabla \ell_t(\boldsymbol{w}_t)\rangle + \frac{L\eta^2}{2}\|\nabla \ell_t(\boldsymbol{w}_t)\|_2^2$$

$$= F(\boldsymbol{w}_t) - \langle F(\boldsymbol{w}_t), \eta \nabla \ell_t(\boldsymbol{w}_t)\rangle + \frac{L\eta^2 r_t}{2}\|\nabla \log p_t\|_2^2,$$

where we used $r^2 = r$. By taking expectation from both sides, we obtain

$$\mathbb{E}[F(\boldsymbol{w}_{t+1})] \leq \mathbb{E}[F(\boldsymbol{w}_t)] - \eta \mathbb{E}\left[\|\nabla F(\boldsymbol{w}_t)\|_2^2\right] + \frac{L\eta^2}{2}\mathbb{E}\left[p_t\|\nabla \log p_t\|_2^2\right]$$

$$\leq \mathbb{E}[F(\boldsymbol{w}_t)] + \frac{L\eta^2 N}{2}\mathbb{E}[-p_t \log p_t] \qquad\qquad \text{(By Lemma A.4)}$$

$$\leq \mathbb{E}[F(\boldsymbol{w}_0)] + \frac{L\eta^2 N}{2}\sum_{t=0}^{T-1}\mathbb{E}[-p_t \log p_t].$$

Therefore, using $\mathbb{E}[p_{\boldsymbol{w}_\tau}(\boldsymbol{y}^*(\boldsymbol{x}) \mid \boldsymbol{x})] = \frac{1}{T}\sum_{t=0}^{T-1}\mathbb{E}[p_{\boldsymbol{w}_t}(\boldsymbol{y}^*(\boldsymbol{x}) \mid \boldsymbol{x})]$, we have

$$\mathbb{E}[p_{\boldsymbol{w}_\tau}(\boldsymbol{y}^*(\boldsymbol{x}) \mid \boldsymbol{x})] \geq \mathbb{E}[q_0(\boldsymbol{y}^*(\boldsymbol{x}) \mid \boldsymbol{x})] - \frac{L\eta^2 N}{2}\sum_{t=0}^{T-1}\mathbb{E}[-p_t \log p_t]. \tag{16}$$

To bound $\sum_{t=0}^{T-1}\mathbb{E}[-p_t \log p_t]$, we can follow the exact same path as the proof of Proposition A.5, while keeping $\eta \leq 1/(2N)$ in all equations, which yields

$$\sum_{t=0}^{T-1}\mathbb{E}[-p_t \log p_t] \lesssim \frac{\log(Nk\gamma^2\eta T)^2}{\gamma^2\eta} + \frac{\|\boldsymbol{w}_0\|_2^2}{\eta T}. \tag{17}$$

Using the bound $\|\boldsymbol{w}_0\|_2 \lesssim \log(Tk\gamma^2)/\gamma$, we obtain

$$\sum_{t=0}^{T-1} \mathbb{E}[-p_t \log p_t] \lesssim \frac{\log(Tk\gamma^2)^2}{\gamma^2 \eta}.$$

Substituting this bound into (16) results in

$$\mathbb{E}[p_{\boldsymbol{w}_\tau}(\boldsymbol{y}^*(\boldsymbol{x}) \mid \boldsymbol{x})] \geq \mathbb{E}[q_0(\boldsymbol{y}^*(\boldsymbol{x}) \mid \boldsymbol{x})] - \frac{CN\eta \log(Tk\gamma^2)^2}{\gamma^2},$$

for some absolute constant $C > 0$, where we used the fact that $L = \mathcal{O}(1)$. $\qquad\square$

With the tools above, we can state the proof of the proposition.

*Proof of Proposition A.8.* Let $\alpha \in (0,1)$ to be fixed later. For convenience, we use the notation $p_{\boldsymbol{w}_\tau} := p_{\boldsymbol{w}_\tau}(\boldsymbol{y}^*(\boldsymbol{x}) \mid \boldsymbol{x})$ where $(\boldsymbol{x}, \boldsymbol{y}^*(\boldsymbol{x}))$ is an independently drawn test sample. We rely on the following decomposition

$$
\begin{aligned}
\mathbb{P}(1 - p_{\boldsymbol{w}_\tau} \geq \varepsilon) &= \mathbb{P}(1 - p_{\boldsymbol{w}_\tau} \geq \varepsilon \mid p_{\boldsymbol{w}_\tau} \geq \alpha) + \mathbb{P}(1 - p_{\boldsymbol{w}_\tau} \geq \varepsilon \mid p_{\boldsymbol{w}_\tau} < \alpha)\mathbb{P}(p_{\boldsymbol{w}_\tau} < \alpha) \\
&\leq \mathbb{P}(1 - p_{\boldsymbol{w}_\tau} \geq \varepsilon \mid p_{\boldsymbol{w}_\tau} \geq \alpha)\mathbb{P}(p_{\boldsymbol{w}_\tau} \geq \alpha) + \mathbb{P}(p_{\boldsymbol{w}_\tau} < \alpha) \\
&\leq \frac{1}{\varepsilon} \mathbb{E}[1 - p_{\boldsymbol{w}_\tau} \mid p_{\boldsymbol{w}_\tau} \geq \alpha]\mathbb{P}(p_{\boldsymbol{w}_\tau} \geq \alpha) + \frac{\mathbb{E}[1 - p_{\boldsymbol{w}_\tau}]}{1 - \alpha} && \text{(By Markov's Inequality)} \\
&\leq \frac{1}{\alpha\varepsilon} \mathbb{E}[p_{\boldsymbol{w}_\tau}(1 - p_{\boldsymbol{w}_\tau}) \mid p_{\boldsymbol{w}_\tau} \geq \alpha]\mathbb{P}(p_{\boldsymbol{w}_\tau} \geq \alpha) + \frac{\mathbb{E}[1 - p_{\boldsymbol{w}_\tau}]}{1 - \alpha} \\
&\leq \frac{1}{\alpha\varepsilon} \mathbb{E}[-p_{\boldsymbol{w}_\tau} \log p_{\boldsymbol{w}_\tau} \mid p_{\boldsymbol{w}_\tau} \geq \alpha]\mathbb{P}(p_{\boldsymbol{w}_\tau} \geq \alpha) + \frac{1 - p_{\boldsymbol{w}_\tau}}{1 - \alpha} && \text{(Using } 1 - x \leq -\log x) \\
&\leq \frac{1}{\alpha\varepsilon} \mathbb{E}[-p_{\boldsymbol{w}_\tau} \log p_{\boldsymbol{w}_\tau}] + \frac{\mathbb{E}[1 - p_{\boldsymbol{w}_\tau}]}{1 - \alpha} && \text{(By the law of total expectation)} \\
&\leq \frac{C \log(Tk\gamma^2)^2}{\alpha\varepsilon\gamma^2\eta T} + \frac{\mathbb{E}[1 - p_{\boldsymbol{w}_\tau}]}{1 - \alpha} && \text{(Using (17))} \\
&\leq \frac{C \log(Tk\gamma^2)^2}{\alpha\varepsilon\gamma^2\eta T} + \frac{1 - \mathbb{E}[q_0(\boldsymbol{y}^*(\boldsymbol{x}) \mid \boldsymbol{x})]}{1 - \alpha} + \frac{C\eta N \log(Tk\gamma^2)^2}{\gamma^2(1 - \alpha)}, && \text{(By Lemma A.10)}
\end{aligned}
$$

where $C > 0$ is an absolute constant. For a fixed $\alpha$, the optimal (constant) learning rate minimizing the above bound is given by $\eta = \sqrt{(1 - \alpha)/(\alpha NT\varepsilon)}$. We will have to choose $T$ sufficiently large so that this learning rate is also smaller than $1/(2N)$, needed to obtain (17). By using this optimal $\eta$, we obtain

$$\mathbb{P}(1 - p_{\boldsymbol{w}_\tau} \geq \varepsilon) \leq 1 - \mathbb{E}[q_0] + \frac{\alpha(1 - \mathbb{E}[q_0])}{1 - \alpha} + \tilde{\mathcal{O}}\left(\sqrt{\frac{N}{\alpha(1 - \alpha)\gamma^4 T\varepsilon}}\right),$$

where we used $q_0 := q_0(\boldsymbol{y}^*(\boldsymbol{x}) \mid \boldsymbol{x})$ for brevity. Next, we must choose $\alpha$.

- If $1 - \mathbb{E}[q_0] \ll \delta$, then to achieve probability of error $1 - \mathbb{E}[q_0] + \delta$, the optimal $\alpha$ is $1/2$, and $T = \tilde{\mathcal{O}}(N/(\gamma^2\varepsilon\delta^2))$ iterations are sufficient. Note that the learning rate will satisfy $\eta \asymp \sqrt{1/(NT\varepsilon)} = \tilde{\Theta}(\gamma^2\delta/N) \ll 1/N$.

- If $1 - \mathbb{E}[q_0] \gg \delta$, then a suitable choice of $\alpha$ is $\alpha = (1 - \mathbb{E}[q_0])/(2\delta)$, and $T = \tilde{\mathcal{O}}(N(1 - \mathbb{E}[q_0])/(\gamma^2\varepsilon\delta^3))$ iterations are sufficient. Note that the learning rate will satisfy $\eta \asymp \sqrt{\delta/((1 - \mathbb{E}[q_0])NT\varepsilon)} = \tilde{\Theta}(\gamma^2\delta/N) \ll 1/N$.

Overall, we can conclude that with a learning rate of $\eta = \tilde{\Theta}(\gamma^2\delta/N)$ and with

$$T = \tilde{\mathcal{O}}\left(\frac{N}{\gamma^4\varepsilon\delta^2} \cdot \min\left(1, \frac{1 - \mathbb{E}[q_0]}{\delta}\right)\right)$$

iterations, we can achieve $p_{\boldsymbol{w}_\tau}(\boldsymbol{y}^*(\boldsymbol{x}) \mid \boldsymbol{x}) \geq 1 - \varepsilon$ with probability at least $q_0(\boldsymbol{y}^*(\boldsymbol{x}) \mid \boldsymbol{x}) - \delta$. $\qquad\square$

## A.6. Policy Gradient with Clipped Importance Weights

In this section, we study the effect of clipped importance weights on the convergence of policy gradients. Namely, we study updates of the form

$$\boldsymbol{w}_{t+1} = \boldsymbol{w}_t + \eta_t r(\boldsymbol{x}^{(t)}, \boldsymbol{y}^{(t)}) \rho_t \nabla \log p_{\boldsymbol{w}_t}(\boldsymbol{y}^{(t)} \mid \boldsymbol{x}^{(t)}), \tag{18}$$

where $\boldsymbol{y}^{(t)} \sim q_t(\cdot \mid \boldsymbol{x}^{(t)})$ and

$$\rho_t = \mathrm{Clip}\,\Big(\frac{p_{\boldsymbol{w}_t}(\boldsymbol{y}^{(t)} \mid \boldsymbol{x}^{(t)})}{q_t(\boldsymbol{y}^{(t)} \mid \boldsymbol{x}^{(t)})}, \zeta^{-1}, \zeta\Big), \quad \zeta \geq 1.$$

We begin by adapting the conditional convergence results of (PG-OR) to use clipped importance weights, as outlined below.

**Theorem A.11** (PG-OR with Clipped Importance Weights). *Let $(\boldsymbol{w}_t)_{t=0}^{T-1}$ denote the iterates of PG with outcome rewards and clipped importance weights as in (18). Suppose Assumption 2.1 holds. For simplicity, assume $\|\boldsymbol{w}_0\|_2 \lesssim 1/\gamma \cdot \log(NTk\gamma^2)$. Let $\tau \sim \mathrm{Unif}(\{0, \ldots, T-1\})$ and $\boldsymbol{x}$ denote an i.i.d. test sample. For any $\alpha \in (0, 1]$, define the event $\mathcal{E}_\alpha := \{\min_t q_t(\boldsymbol{y}^*(\boldsymbol{x}) \mid \boldsymbol{x}) \geq \alpha\}$ and $\pi_\alpha := \mathbb{P}(\mathcal{E}_\alpha)$. Then*

*1. For the constant learning rate $\eta_t = 1/(2\zeta N)$ we have*

$$\mathbb{E}[p_{\boldsymbol{w}_\tau}(\boldsymbol{y}^*(\boldsymbol{x}) \mid \boldsymbol{x}) \mid \mathcal{E}_\alpha] \geq 1 - \tilde{\mathcal{O}}\Big(\frac{\zeta^2 N}{\pi_\alpha \alpha \gamma^2 T}\Big).$$

*Further, the same bound holds for the averaged iterate $\bar{\boldsymbol{w}}_T^{\mathrm{PG}} = \frac{1}{T}\sum_{t=0}^{T-1} \boldsymbol{w}_t$.*

*2. For the adaptive learning rate $\eta_t = (4\zeta + 2\rho_t \|\nabla \log p_{\boldsymbol{w}_t}(\boldsymbol{y}^{(t)} \mid \boldsymbol{x}^{(t)})\|_2)^{-1}$ we have*

$$\mathbb{E}[p_{\boldsymbol{w}_\tau}(\boldsymbol{y}^*(\boldsymbol{x}) \mid \boldsymbol{x}) \mid \mathcal{E}_\alpha] \geq 1 - \tilde{\mathcal{O}}\Big(\frac{\zeta^2}{\pi_\alpha \alpha \gamma^2 T}\Big).$$

*Proof.* The proof of both statements follow by closely tracking the proof of Theorem 3.1 and Theorem 3.2 respectively. Since they are similar, we only state the proof of the second case.

We can recreate the online bound of Proposition A.5 by retracing its proof steps and for learning rate $\eta_t = 1/\big(2(\zeta\lambda + \rho_t \|\nabla \log p_{\boldsymbol{w}_t}(\boldsymbol{y}^{(t)} \mid \boldsymbol{x}^{(t)})\|_2)\big)$ obtain

$$\mathbb{E}\Bigg[\frac{1}{T}\sum_{t=0}^{T-1} q_t(\boldsymbol{y}^*(\boldsymbol{x}^{(t)}) \mid \boldsymbol{x}^{(t)}) \min(\tfrac{\lambda}{2}, -\log p_{\boldsymbol{w}_t}(\boldsymbol{y}^*(\boldsymbol{x}^{(t)}) \mid \boldsymbol{x}^{(t)}))\Bigg] \lesssim \frac{\zeta^2 \log(NTk\gamma^2)^2}{\gamma^2 T}. \tag{19}$$

Specifically, this time we define $\ell_t(\boldsymbol{w}) := -r_t \rho_t \log p_{\boldsymbol{w}}(\boldsymbol{y}^*(\boldsymbol{x}^{(t)}) \mid \boldsymbol{x}^{(t)})$. Importantly, a consequence of this definition is that while $\rho_t$ depends on the weight $\boldsymbol{w}_t$, this weight is frozen and there is no gradient from $\rho_t$ appearing in $\nabla \ell_t(\boldsymbol{w})$. Namely $\ell_t(\boldsymbol{w})$ is a still convex function of $\boldsymbol{w}$, and satisfies $\|\nabla \ell_t\|_2 \leq 2\ell_t$ thanks to Lemma A.6. Further, notice that (18) is equivalent to $\boldsymbol{w}_{t+1} = \boldsymbol{w}_t - \eta_t \nabla \ell_t$ due to the 0-1 reward structure, and recall that we use $\tilde{\ell}_t(\boldsymbol{w}) := -\log p_{\boldsymbol{w}}(\boldsymbol{y}^*(\boldsymbol{x}^{(t)}) \mid \boldsymbol{x}^{(t)})$ to denote negative log-likelihood. Therefore, we can invoke the second case of Proposition A.1 with the learning rate given above, which yields

$$\frac{1}{T}\sum_{t=0}^{T-1} \frac{r_t \rho_t \tilde{\ell}_t(\boldsymbol{w}_t)}{1 + 2\zeta^{-1}\lambda^{-1}\rho_t r_t \tilde{\ell}_t(\boldsymbol{w}_t)} \leq \frac{2}{T}\sum_{t=0}^{T-1} r_t \rho_t \tilde{\ell}_t(\boldsymbol{v}) + \frac{2\zeta\lambda\|\boldsymbol{v} - \boldsymbol{w}_0\|_2^2}{T}.$$

Using the bound $\rho_t \in [\zeta^{-1}, \zeta]$ and $r_t \leq 1$, we can lower bound the RHS above and write

$$\frac{1}{T}\sum_{t=0}^{T-1} \frac{r_t \zeta^{-1} \tilde{\ell}_t(\boldsymbol{w}_t)}{1 + 2\lambda^{-1}\tilde{\ell}_t(\boldsymbol{w}_t)} \leq \frac{2}{T}\sum_{t=0}^{T-1} r_t \rho_t \tilde{\ell}_t(\boldsymbol{v}) + \frac{2\zeta\lambda\|\boldsymbol{v} - \boldsymbol{w}_0\|_2^2}{T}.$$

Next, we take expectation from both sides and use the fact that $\mathbb{E}[r_t \mid \boldsymbol{x}^{(t)}, \boldsymbol{w}_t] = q_t(\boldsymbol{y}^*(\boldsymbol{x}^{(t)}) \mid \boldsymbol{x}^{(t)}) =: q_t$ and that

$$\mathbb{E}\Big[r_t \rho_t \mid \boldsymbol{x}^{(t)}, \boldsymbol{w}_t\Big] \leq q_t \max(p_{\boldsymbol{w}_t}/q_t, \zeta^{-1}) \leq 1.$$

We can therefore write

$$\frac{\zeta^{-1}}{T} \mathbb{E}\left[\sum_{t=0}^{T-1} \frac{q_t \tilde{\ell}_t(\boldsymbol{w}_t)}{1 + 2\lambda^{-1}\tilde{\ell}_t(\boldsymbol{w}_t)}\right] \leq \frac{2}{T} \sum_{t=0}^{T-1} \tilde{\ell}_t(\boldsymbol{v}) + \frac{2\zeta\lambda\|\boldsymbol{v} - \boldsymbol{w}_0\|_2^2}{T}.$$

By following the remaining steps from the proof of Proposition A.7, we arrive at (19). By letting $\lambda = 2$, using the inequality $1 - x \leq -\log x$, and employing the same online-to-batch conversion of replacing $\boldsymbol{x}^{(t)}$ with an independent copy $\boldsymbol{x}$, we can show

$$\mathbb{E}\left[\frac{1}{T} \sum_{t=0}^{T-1} q_t(\boldsymbol{y}^*(\boldsymbol{x}) \,|\, \boldsymbol{x})(1 - p_{\boldsymbol{w}_t}(\boldsymbol{y}^*(\boldsymbol{x}) \,|\, \boldsymbol{x}))\right] \lesssim \frac{\zeta^2 \log(NTk\gamma^2)^2}{\gamma^2 T}$$

or equivalently

$$\mathbb{E}[q_\tau(\boldsymbol{y}^*(\boldsymbol{x}) \,|\, \boldsymbol{x})(1 - p_{\boldsymbol{w}_\tau}(\boldsymbol{y}^*(\boldsymbol{x}) \,|\, \boldsymbol{x}))] \lesssim \frac{\zeta^2 \log(NTk\gamma^2)^2}{\gamma^2 T}.$$

Combining the above bound with the fact that

$$\alpha \mathbb{E}[1 - p_{\boldsymbol{w}_\tau}(\boldsymbol{y}^*(\boldsymbol{x}) \,|\, \boldsymbol{x}) \,|\, \mathcal{E}_\alpha]\mathbb{P}(\mathcal{E}_\alpha) \leq \mathbb{E}[q_\tau(\boldsymbol{y}^*(\boldsymbol{x}) \,|\, \boldsymbol{x})(1 - p_{\boldsymbol{w}_\tau}(\boldsymbol{y}^*(\boldsymbol{x}) \,|\, \boldsymbol{x}))]$$

finishes the proof. $\qquad\square$

Using the above theorem, we can also adapt the unconditional post-training guarantees of Corollary 3.4 and 3.6 to use clipped importance weights. We state this result without presenting its proof, which can simply be obtained by going through the proofs of Corollary 3.4 (in Appendix A.3) and the proof of Corollary 3.6 (in Appendix A.4) respectively.

**Corollary A.12.** *Suppose Assumption 2.1 holds and a base policy $q_0$ is given. Let $(\boldsymbol{w}_t)_{t=0}^{T-1}$ denote the iterates of PG with outcome rewards and clipped importance weights as in (18), with the adaptive learning rate of Theorem A.11. Let $T$ denote the number of PG iterations, and $Q$ denote the number of reward queries. Let $\tau \sim \mathrm{Unif}(\{0, \ldots, T-1\})$.*

1. *Suppose $q_t = \frac{1}{2}q_0 + \frac{1}{2}\mathrm{Unif}$ for all $t$. Then, for any $\varepsilon \in (0, 1)$, with $Q = T = \tilde{\mathcal{O}}\left(\frac{\zeta^2 \min\left(\mathcal{Q}_{q_0}((1-o(1))\varepsilon)^{-1}, k^N\right)}{\gamma^2\varepsilon}\right)$ iterations and reward queries, we achieve $\mathbb{E}[p_{\boldsymbol{w}_\tau}(\boldsymbol{y}^*(\boldsymbol{x}) \,|\, \boldsymbol{x})] \geq 1 - \varepsilon$.*

2. *Suppose $q_t = q$ where $q$ is the output of Algorithm 1 with $m = \min\left(\lceil \mathcal{Q}_{q_0}((1 - o(1))\varepsilon)^{-1}\rceil, k^N\right)$. Then, for any $\varepsilon \in (0, 1)$, with $Q = \tilde{\mathcal{O}}((m + \varepsilon^{-1})/\gamma^2)$ reward queries and $T = \tilde{\mathcal{O}}(\zeta^2/(\gamma^2\varepsilon))$ iterations, we achieve $\mathbb{E}[p_{\boldsymbol{w}_\tau}(\boldsymbol{y}^*(\boldsymbol{x}) \,|\, \boldsymbol{x})] \geq 1 - \varepsilon$.*

We observe that when using Algorithm 1 for behavior policy, only the number of iterations will scale with $\zeta^2$, while using a simple mixture of the base and uniform policies as in $q_t = \frac{1}{2}q_0 + \frac{1}{2}\mathrm{Unif}$ results in both the number of reward queries and iterations to scale with $\zeta^2$.

## B. Proofs of Section 4

Here we will state the proof of the results that appeared in Section 4.

### B.1. Proof of Theorem 4.1

To prove Theorem 4.1, similar to the previous section, we first prove a general statement that handles any input sequence.

**Proposition B.1.** *Let $(\boldsymbol{w}_t)$ denote the (PG-PR) iterates with $\boldsymbol{y}^{(t)} \sim q_t(\cdot \,|\, \boldsymbol{x}^{(t)})$. Suppose the sequence $(\boldsymbol{x}^{(t)}, \boldsymbol{y}^*(\boldsymbol{x}^{(t)}))$ is chosen arbitrarily (in particular not necessarily i.i.d.), with the only constraint being Assumption 2.1. Assume $\eta_t = 1/(2(\lambda + \|\nabla\ell_t(\boldsymbol{w}_t)\|_2)$ for all $t$ and some $\lambda > 0$, and $\|\boldsymbol{w}_0\|_2 \leq 1/\gamma \cdot \log(NkT\gamma^2/\lambda)$. Then, for any $i \in [N]$, we have*

$$\mathbb{E}_{\boldsymbol{y}^{(0)},\ldots,\boldsymbol{y}^{(T-1)}}\left[\frac{1}{T} \sum_{t=0}^{T-1} q_t(\boldsymbol{y}_{1:i}^*(\boldsymbol{x}^{(t)}) \,|\, \boldsymbol{x}^{(t)}) \min\left(\tfrac{\lambda}{2}, -\log p_{\boldsymbol{w}_t}(\boldsymbol{y}^*(\boldsymbol{x}^{(t)})_{1:i} \,|\, \boldsymbol{x}^{(t)})\right)\right] \leq \frac{20\lambda \log(NkT\gamma^2/\lambda)^2}{\gamma^2 T}. \qquad (20)$$

*Proof.* The proof is similar to that of Proposition A.7. Namely, using the fact that $A_t^{(j)} \neq 0$ if and only if $\boldsymbol{y}_{1:j}^{(t)} = \boldsymbol{y}^{*(t)}_{1:j}$ (where we recall the shorthand notation $\boldsymbol{y}^{*(t)} := \boldsymbol{y}^*(\boldsymbol{x}^{(t)})$). Then, we can write (PG-PR) as

$$\boldsymbol{w}_{t+1} = \boldsymbol{w}_t + \eta_t \sum_{j=1}^{N} A_j^{(t)} \log p_{\boldsymbol{w}_t}(y^*{}_j^{(t)} \mid \boldsymbol{x}^{(t)}, \boldsymbol{y}^{*(t)}_{1:j-1})$$

$$= \boldsymbol{w}_t - \eta_t \nabla \ell_t(\boldsymbol{w}_t)$$

where we recall $\ell_t(\boldsymbol{w}_t) := \sum_{j=1}^{N} A_j^{(t)} \log p_{\boldsymbol{w}_t}(y^*{}_j^{(t)} \mid \boldsymbol{x}^{(t)}, \boldsymbol{y}^{*(t)}_{1:j-1})$. Once again, by non-negativity of the advantages $A_j^{(t)} \geq 0$, $\ell_t$ is convex. Therefore, we attempt at applying the second case of Proposition A.1. For that, we need to establish a bound on $\|\nabla \ell_t\|_2$, which we do so by the triangle inequality,

$$\|\nabla \ell_t(\boldsymbol{w})\|_2 \leq \sum_{j=1}^{N} A_j^{(t)} \left\| \log p_{\boldsymbol{w}}(y^*{}_j^{(t)} \mid \boldsymbol{x}^{(t)}, \boldsymbol{y}^{*(t)}_{1:j-1}) \right\|_2 \leq -2 \sum_{j=1}^{N} A_j^{(t)} \log p_{\boldsymbol{w}}(y^*{}_j^{(t)} \mid \boldsymbol{x}^{(t)}, \boldsymbol{y}^{*(t)}_{1:j-1}) = 2\ell_t(\boldsymbol{w}),$$

where the second inequality follows from Lemma A.6. Using the above and the learning rate schedule $\eta_t = 1/(2(\lambda + \|\nabla \ell_t(\boldsymbol{w}_t)\|_2))$ we can invoke the second case of Proposition A.1 to obtain

$$\frac{1}{T} \sum_{t=0}^{T-1} \frac{\ell_t(\boldsymbol{w}_t)}{1 + 2\lambda^{-1}\ell_t(\boldsymbol{w}_t)} \leq \frac{2}{T} \sum_{t=0}^{T-1} \ell_t(\boldsymbol{v}) + \frac{2\lambda \|\boldsymbol{v} - \boldsymbol{w}_0\|_2^2}{T} \tag{21}$$

for any $\boldsymbol{v} \in \mathbb{R}^D$. Further, we use the following definitions

$$\ell_t^i(\boldsymbol{w}) := -\sum_{j=1}^{i} A_j^{(t)} \log p_{\boldsymbol{w}}(y^*{}_j^{(t)} \mid \boldsymbol{x}^{(t)}, \boldsymbol{y}^{*(t)}_{1:j-1}),$$

$$\tilde{\ell}_t^i(\boldsymbol{w}) := -\log p_{\boldsymbol{w}}(\boldsymbol{y}^{*(t)}_{1:i} \mid \boldsymbol{x}^{(t)}),$$

$$\tilde{\ell}_t(\boldsymbol{w}) := -\log p_{\boldsymbol{w}}(\boldsymbol{y}^{*(t)} \mid \boldsymbol{x}^{(t)}).$$

Since $\ell_t^i \leq \ell_t$ and $\ell_t^i \leq \tilde{\ell}_t^i$, we have

$$\frac{1}{T} \sum_{t=0}^{T-1} \frac{\ell_t^i(\boldsymbol{w}_t)}{1 + 2\lambda^{-1}\tilde{\ell}_t^i(\boldsymbol{w}_t)} \leq \frac{1}{T} \sum_{t=0}^{T-1} \frac{\ell_t^i(\boldsymbol{w}_t)}{1 + 2\lambda^{-1}\ell_t^i(\boldsymbol{w}_t)}$$

$$\leq \frac{1}{T} \sum_{t=0}^{T-1} \frac{\ell_t(\boldsymbol{w}_t)}{1 + 2\lambda^{-1}\ell_t(\boldsymbol{w}_t)}$$

$$\leq \frac{2}{T} \sum_{t=0}^{T-1} \ell_t(\boldsymbol{v}) + \frac{2\lambda \|\boldsymbol{v} - \boldsymbol{w}_0\|_2^2}{T}, \tag{22}$$

where the last inequality follows from (21). Next, we take expectation over the randomness in the draw of $\boldsymbol{y}^{(0)}, \ldots, \boldsymbol{y}^{(T-1)}$. Note that $\boldsymbol{w}_t$ is independent of $\boldsymbol{y}^{(t)}$, and $\mathbb{E}\left[A_j^{(t)} \mid \boldsymbol{w}_t\right] = q_t(\boldsymbol{y}_{1:j}^{(t)} \mid \boldsymbol{x}^{(t)})$ for all $j \in [N]$. Therefore,

$$\mathbb{E}_{\boldsymbol{y}^{(0)}, \ldots, \boldsymbol{y}^{(T-1)}}\left[\ell_t^i(\boldsymbol{w}_t)\right] = \mathbb{E}_{\boldsymbol{y}^{(0)}, \ldots, \boldsymbol{y}^{(T-1)}}\left[\sum_{j=1}^{i} -q_t(\boldsymbol{y}^{*(t)}_{1:j} \mid \boldsymbol{x}^{(t)}) \log p_{\boldsymbol{w}_t}(y^*{}_j^{(t)} \mid \boldsymbol{x}^{(t)}, \boldsymbol{y}^{*(t)}_{1:j-1})\right]$$

$$\geq \mathbb{E}_{\boldsymbol{y}^{(0)}, \ldots, \boldsymbol{y}^{(T-1)}}\left[-q_t(\boldsymbol{y}^{*(t)}_{1:i} \mid \boldsymbol{x}^{(t)}) \log p_{\boldsymbol{w}_t}(\boldsymbol{y}^{*(t)}_{1:i} \mid \boldsymbol{x}^{(t)})\right]$$

$$= \mathbb{E}_{\boldsymbol{y}^{(0)}, \ldots, \boldsymbol{y}^{(T-1)}}\left[-q_t(\boldsymbol{y}^{*(t)}_{1:i} \mid \boldsymbol{x}^{(t)})\tilde{\ell}_t^i(\boldsymbol{w}_t)\right]. \tag{23}$$

By combining (22), (23), and the fact that $\mathbb{E}[\ell_t(\boldsymbol{v})] \leq \tilde{\ell}_t(\boldsymbol{v})$, we obtain

$$\mathbb{E}_{\boldsymbol{y}^{(0)}, \ldots, \boldsymbol{y}^{(T-1)}}\left[\frac{1}{T} \sum_{t=0}^{T-1} \frac{\tilde{\ell}_t^i(\boldsymbol{w}_t)}{1 + 2\lambda^{-1}\tilde{\ell}_t^i(\boldsymbol{w}_t)}\right] \leq \frac{2}{T} \sum_{t=0}^{T-1} \tilde{\ell}_t(\boldsymbol{v}) + \frac{2\lambda \|\boldsymbol{v} - \boldsymbol{w}_0\|_2^2}{T}.$$

The remainder of the proof follows exactly as in that of Proposition A.7. Namely, we let $\boldsymbol{v} = R_v \boldsymbol{w}^*$ for which we have have $\tilde{\ell}_t(\boldsymbol{v}) \leq Nke^{-R_v\gamma}$. We choose $R_v = 1/\gamma \cdot \log(NkT\gamma^2/\lambda)$ and notice that $\frac{\tilde{\ell}_t^i}{1+2\lambda^{-1}\tilde{\ell}_t^i} \geq \frac{1}{2}\min(\lambda/2, \tilde{\ell}_t^i)$. Combined with the bound on $\|\boldsymbol{w}_0\|_2$, the proof is complete. □

Note that the above guarantee is more general than that of Proposition A.7. While it can recover the same bound for $i = N$, it can cover cases where $i < N$, where it shows that (PG-PR) can boost the accuracy on the first $i$ tokens if the behavior policy assigns a non-trivial probability to the first $i$ ground-truth tokens. As done before, we can use an online-to-batch conversion to turn the above bound into a conditional guarantee on test error.

**Proposition B.2.** *Let* $(\boldsymbol{w}_t)$ *denote the* (PG-PR) *iterates with adaptive learning rate* $\eta_t = (4 + 2\|\nabla\ell_t(\boldsymbol{w}_t)\|_2)^{-1}$. *Suppose Assumption 2.1 holds, and* $\|\boldsymbol{w}_0\|_2 \leq 1/\gamma \cdot \log(NkT\gamma^2/2)$. *For any* $i \in [N]$ *and any* $\alpha \in (0,1]$, *define the event* $\mathcal{E}_{i,\alpha} := \{\inf_t q_t(\boldsymbol{y}^*(\boldsymbol{x})_{1:i} \,|\, \boldsymbol{x}) \geq \alpha\}$ *and* $\pi_{i,\alpha} := \mathbb{P}(\mathcal{E}_{i,\alpha})$. , *in expectation over the training randomness, over the test sample* $\boldsymbol{x}$, *and over the choice* $\tau \sim \mathrm{Unif}(\{0,\ldots,T-1\})$), *we have*

$$\mathbb{E}[p_{\boldsymbol{w}_\tau}(\boldsymbol{y}^*(\boldsymbol{x})_{1:i} \,|\, \boldsymbol{x}) \,|\, \mathcal{E}_{i,\alpha}] \geq 1 - \tilde{\mathcal{O}}\left(\frac{1}{\pi_{i,\alpha}\alpha\gamma^2 T}\right). \tag{24}$$

*Proof.* Similar to the proof of Theorem 3.2, we take expectation from both sides of (20), and use the independence of $\boldsymbol{x}^{(t)}$ from $\boldsymbol{w}_t$ to replace it with an independent test sample $\boldsymbol{x}$, and write

$$\mathbb{E}\left[\frac{1}{T}\sum_{t=0}^{T-1} q_t(\boldsymbol{y}_{1:i}^*(\boldsymbol{x}) \,|\, \boldsymbol{x}) \min\left(\frac{\lambda}{2}, -\log p_{\boldsymbol{w}_t}(\boldsymbol{y}_{1:i}^*(\boldsymbol{y}) \,|\, \boldsymbol{x})\right)\right] \lesssim \frac{\lambda\log(NkT\gamma^2/\lambda)^2}{\gamma^2 T}.$$

Using $1 - x \leq -\log x$, we can write

$$\mathbb{E}\left[\frac{1}{T}\sum_{t=0}^{T-1} q_t(\boldsymbol{y}_{1:i}^*(\boldsymbol{x}) \,|\, \boldsymbol{x}) \min\left(\frac{\lambda}{2}, 1 - p_{\boldsymbol{w}_t}(\boldsymbol{y}_{1:i}^*(\boldsymbol{y}) \,|\, \boldsymbol{x})\right)\right] \lesssim \frac{\lambda\log(NkT\gamma^2/\lambda)^2}{\gamma^2 T}.$$

By choosing $\lambda = 2$, we can remove the minimum and write

$$\mathbb{E}\left[q_\tau(\boldsymbol{y}_{1:i}^*(\boldsymbol{x}) \,|\, \boldsymbol{x})\left(1 - p_{\boldsymbol{w}_\tau}(\boldsymbol{y}_{1:i}^*(\boldsymbol{y}) \,|\, \boldsymbol{x})\right)\right] \lesssim \frac{\log(NkT\gamma^2)^2}{\gamma^2 T},$$

where we replaced the average with expectation over $\tau$. Next, we can use the law of total expectation and note that on $\mathcal{E}_{i,\alpha}$ we have $q_\tau(\boldsymbol{y}_{1:i}^*(\boldsymbol{x}) \,|\, \boldsymbol{x}) \geq \alpha$ for all $t$, which yields

$$\mathbb{E}[1 - p_{\boldsymbol{w}_\tau}(\boldsymbol{y}_{1:i}^*(\boldsymbol{y}) \,|\, \boldsymbol{x}) \,|\, \mathcal{E}_{i,\alpha}]\mathbb{P}(\mathcal{E}_{i,\alpha}) \lesssim \frac{\log(NkT\gamma^2)^2}{\gamma^2 T}$$

which completes the proof. □

With this proposition, we can present the proof of the main theorem of this section.

*Proof of Theorem 4.1.* The proof is roughly similar to the argument presented in Appendix A.4. This time, we define

$$\mathcal{E}_\alpha := \{\min_{j\in[N]} q_0(y_j^*(\boldsymbol{x}) \,|\, \boldsymbol{x}, \boldsymbol{y}_{1:j-1}^*(\boldsymbol{x})) \geq \alpha\},$$

$$\tilde{\mathcal{E}}_{\tilde{\alpha}} := \{q(\boldsymbol{y}^*(\boldsymbol{x}) \,|\, \boldsymbol{x}) \geq \tilde{\alpha}\},$$

for $\alpha, \tilde{\alpha} \in (0,1]$. Using the law of total expectation and Proposition B.2, we have

$$\mathbb{E}[1 - p_{\boldsymbol{w}_\tau}(\boldsymbol{y}^*(\boldsymbol{x}) \,|\, \boldsymbol{x})] = \mathbb{E}\left[1 - p_{\boldsymbol{w}_\tau}(\boldsymbol{y}^*(\boldsymbol{x}) \,|\, \boldsymbol{x}) \,|\, \tilde{\mathcal{E}}_{\tilde{\alpha}}\right]\mathbb{P}\left(\tilde{\mathcal{E}}_{\tilde{\alpha}}\right) + \mathbb{E}\left[1 - p_{\boldsymbol{w}_\tau}(\boldsymbol{y}^*(\boldsymbol{x}) \,|\, \boldsymbol{x}) \,|\, \tilde{\mathcal{E}}_{\tilde{\alpha}}^c\right]\mathbb{P}\left(\tilde{\mathcal{E}}_{\tilde{\alpha}}^c\right)$$

$$\leq \tilde{\mathcal{O}}\left(\frac{1}{\tilde{\alpha}\gamma^2 T}\right) + \mathbb{P}\left(\tilde{\mathcal{E}}_{\tilde{\alpha}}^c\right). \tag{25}$$

Let

$$m := \lceil 2(\log N + 1) \cdot m^* \rceil, \quad m^* := \min\left(\mathcal{Q}_{q_0}^{\mathrm{TL}}\left((1 - \tfrac{1}{\log(1/\varepsilon)}\varepsilon)^{-1}, k\right)\right.$$

Once again, we proceed by considering each case in the minimum above separately.

- Suppose the minimum is attained by the first term. Note that

$$q(y_j^*(\boldsymbol{x}) \,|\, \boldsymbol{x}, \boldsymbol{y}_{1:j-1}^*(\boldsymbol{x})) \geq 1 - (1 - q_0(y_j^*(\boldsymbol{x}) \,|\, \boldsymbol{x}, \boldsymbol{y}_{1:j-1}^*(\boldsymbol{x})/2)^m$$

for all $j$. Therefore, on $\mathcal{E}_\alpha$ we have $q(y_j^*(\boldsymbol{x}) \,|\, \boldsymbol{x}, \boldsymbol{y}_{1:j-1}^*(\boldsymbol{x})) \geq 1 - e^{-\alpha m/2}$ and

$$q(\boldsymbol{y}^*(\boldsymbol{x}) \,|\, \boldsymbol{x}) \geq 1 - Ne^{-\alpha m/2} \geq 1 - N^{-\alpha m^*(\log N + 1)}.$$

By letting $\alpha = 1/m^*$ we obtain $q(\boldsymbol{y}^*(\boldsymbol{x}) \,|\, \boldsymbol{x}) \geq 1 - e^{-1}$. Thus, if we let $\tilde{\alpha} := 1 - e^{-1}$, we have

$$\mathbb{P}\Big(\tilde{\mathcal{E}}_{\tilde{\alpha}}^c\Big) \leq \mathbb{P}(\mathcal{E}_\alpha^c) = \mathcal{Q}_{q_0}^{-1}(1/m^*) \leq \Big(1 - \frac{1}{\log(1/\varepsilon)}\Big)\varepsilon.$$

Going back to (25), we obtain

$$\mathbb{E}[1 - p_{\boldsymbol{w}_\tau}(\boldsymbol{y}^*(\boldsymbol{x}) \,|\, \boldsymbol{x})] \leq \tilde{\mathcal{O}}\Big(\frac{1}{\gamma^2 T}\Big) + \Big(1 - \frac{1}{\log(1/\varepsilon)}\Big)\varepsilon.$$

Choosing $T = \tilde{\mathcal{O}}(1/(\gamma^2\varepsilon))$ is sufficient to ensure the first term remains bounded by $\varepsilon/\log(1/\varepsilon)$, and the overall expected error remains bounded by $\varepsilon$.

- Now suppose the minimum is attained by the second term. Then,

$$q(y_j^*(\boldsymbol{x}) \,|\, \boldsymbol{x}, \boldsymbol{y}_{1:j-1}^*(\boldsymbol{x})) \geq 1 - (1 - 1/(2k))^m \geq 1 - e^{-m/(2k)}.$$

And $q(\boldsymbol{y}^*(\boldsymbol{x}) \,|\, \boldsymbol{x}) \geq 1 - Ne^{-m/(2k)} \geq 1 - Ne^{-\log N - 1}$. Thus, for $\tilde{\alpha} = 1 - e^{-1}$ we have $\mathbb{P}\Big(\tilde{\mathcal{E}}_{\tilde{\alpha}}^c\Big) = 0$. Hence (25) reads

$$\mathbb{E}[1 - p_{\boldsymbol{w}_\tau}(\boldsymbol{y}^*(\boldsymbol{x}) \,|\, \boldsymbol{x})] \leq \tilde{\mathcal{O}}\Big(\frac{1}{\gamma^2 T}\Big),$$

and choosing $T = \tilde{\mathcal{O}}(1/(\gamma^2\varepsilon))$ is sufficient.

The argument for the number of reward queries is also similar to that of Appendix A.4, adapted to Algorithm 2. Note that if $\boldsymbol{y}^{(t)} = \boldsymbol{y}^*(\boldsymbol{x}^{(t)})$ for $\boldsymbol{y}^{(t)} = p_{\boldsymbol{w}_t}$, then Algorithm 2 only makes one reward query, and (PG-PR) does not need any additional reward queries. Otherwise, Algorithm 2 makes at most $1 + mN$ reward queries, and (PG-PR) makes at most $N$ reward queries. Thus, we can write the total number of reward queries as

$$Q = T + (m+1)N\,\mathbb{E}\left[\sum_{t=1}^{T} \mathbb{1}[\hat{\boldsymbol{y}}^{(t)} \neq \boldsymbol{y}^*(\boldsymbol{x}^{(t)})]\right]$$

$$= T + (m+1)N\sum_{t=1}^{T} \mathbb{E}\Big[1 - p_{\boldsymbol{w}_t}(\boldsymbol{y}^*(\boldsymbol{x}^{(t)}) \,|\, \boldsymbol{x}^{(t)})\Big]$$

$$= T + (m+1)N\sum_{t=1}^{T} \mathbb{E}[1 - p_{\boldsymbol{w}_t}(\boldsymbol{y}^*(\boldsymbol{x}) \,|\, \boldsymbol{x})] \qquad \text{(Using independent of } \boldsymbol{x}^{(t)} \text{ from } \boldsymbol{w}_t\text{)}$$

$$= T + (m+1)NT\mathbb{P}\Big(\tilde{\mathcal{E}}_{\tilde{\alpha}}^c\Big) + (m+1)N\tilde{\mathcal{O}}(\gamma^{-2}) \qquad \text{(Using (25) and } \tilde{\alpha} = 1 - e^{-1}\text{).}$$

Using $\mathbb{P}\Big(\tilde{\mathcal{E}}_{\tilde{\alpha}}^c\Big) \leq (1 - o(1))\varepsilon$ and $T = \tilde{\mathcal{O}}(1/(\gamma^2\varepsilon))$, we have

$$Q = T + (m+1)N\tilde{\mathcal{O}}(\gamma^{-2}) = \tilde{\mathcal{O}}\Big(\frac{mN + \varepsilon^{-1}}{\gamma^2}\Big).$$

Plugging in $m$ completes the proof. $\qquad\qquad\qquad\qquad\qquad\qquad\qquad\qquad\qquad\qquad\qquad\qquad\qquad$ □

### B.2. Proof of Corollary 4.2

The first case of the corollary directly follows from Proposition 2.2.

For the second case, note that as a consequence of the assumption on the feature map, for a given $\boldsymbol{x}$ we have

$$p_{\boldsymbol{w}}(y_i^*(\boldsymbol{x}) \,|\, \boldsymbol{x}, \boldsymbol{y}_{1:i-1}^*(\boldsymbol{x})) = p_{\boldsymbol{w}}(\boldsymbol{y}^*(\boldsymbol{x}) \,|\, \boldsymbol{x})^{1/N} \tag{26}$$

for all $i$. Further, by going back to Proposition A.5 in the SGD setting (thus removing the base policy in (11)), we have $\mathbb{E}\left[-\log p_{\bar{\boldsymbol{w}}_n^{\text{SGD}}}(\boldsymbol{y}^*(\boldsymbol{x}) \,|\, \boldsymbol{x})\right] \leq \mathcal{O}(N \log(nk\gamma^2)/(\gamma^2 n))$, where we used the Jensen inequality. Recall $q_0 = p_{\bar{\boldsymbol{w}}_n^{\text{SGD}}}$. Define $Z(\boldsymbol{x}) := \min_{i \in N} q_0(y_i^*(\boldsymbol{x}) \,|\, \boldsymbol{x}, \boldsymbol{y}_{1:i-1}^*(\boldsymbol{x}))$. By (26), we have $Z(\boldsymbol{x}) = q_0(\boldsymbol{y}^*(\boldsymbol{x}) \,|\, \boldsymbol{x})^{1/N}$. Consequently we have

$$\begin{aligned} \mathbb{P}(Z(\boldsymbol{x}) < \alpha) = \mathbb{P}\left(q_0(\boldsymbol{y}^*(\boldsymbol{x}) \,|\, \boldsymbol{x})^{1/N} < \alpha\right) &= \mathbb{P}\left(-\frac{1}{N}\log q_0(\boldsymbol{y}^*(\boldsymbol{x}) \,|\, \boldsymbol{x}) \geq \log(1/\alpha)\right) \\ &\leq \frac{\mathbb{E}[-\log q_0(\boldsymbol{y}^*(\boldsymbol{x}) \,|\, \boldsymbol{x})]}{N \log(1/\alpha)} \qquad \text{(By Markov's Inequality)} \\ &\leq \frac{\log(nk\gamma^2)}{n\gamma^2 \log(1/\alpha)}. \end{aligned}$$

Recall that $\mathcal{Q}_{q_0}^{\text{TL}}(\varepsilon) = \sup\{\alpha \in [0,1] : \mathbb{P}(Z \leq \alpha) \leq \varepsilon\}$. Thus we have

$$\mathcal{Q}_{q_0}^{\text{TL}}(\varepsilon) \geq \exp\left(-\frac{\log(n\gamma^2 k)}{n\gamma^2 \varepsilon}\right).$$

Recall from Theorem 4.1 that while the number of iterations is always given by $T = \tilde{\mathcal{O}}(1/(\gamma^2\varepsilon))$, the number of reward queries is given by

$$Q = \tilde{\mathcal{O}}\left(\frac{N \min\left(\mathcal{Q}_{q_0}^{\text{TL}}(\varepsilon)^{-1}, k\right) + \varepsilon^{-1}}{\gamma^2}\right).$$

When $n \geq \frac{\log(k/\varepsilon)}{\gamma^2\varepsilon}$, one can verify that $\mathcal{Q}_{q_0}^{\text{TL}}(\varepsilon) \geq e^{-3}$. Thus, we can choose this $\mathcal{Q}_{q_0}^{\text{TL}}$ in the minimum above. Otherwise, we can choose $k$. This finishes the proof. $\qquad \square$

## C. Proofs of Section 5

Our minimax lower bounds rely on the following fact.

**Lemma C.1.** *Suppose $y$ is a random variable drawn uniformly from $\{1, \dots, m\}$. To estimate $y$, an algorithm is allowed $l \leq m$ rounds of guesses, where at round $t$ the algorithm produces a guess $\hat{y}_t$ and receives the answer $\mathbb{1}[\hat{y}_t = y]$. Let $\hat{y}_l$ denote the final guess of the algorithm. For any such algorithm we have*

$$\mathbb{P}(\hat{y}_l \neq y) \geq \frac{m-l}{m}.$$

*Proof.* Define $A_l = \cap_{i=1}^{l}\{\hat{y}_i \neq y\}$. We will prove by induction that $\mathbb{P}(A_l) = (m-l)/m$ for $l \leq m$. Note that $\hat{y}_1$ is independent of $y$, therefore $\mathbb{P}(A_1) = (m-1)/m$. For $l \geq 2$, we have

$$\mathbb{P}(A_l) = \mathbb{P}(\hat{y}_l \neq y \,|\, A_{l-1})\mathbb{P}(A_{l-1}). \tag{27}$$

Further,

$$\begin{aligned} \mathbb{P}(\hat{y}_l \neq y \,|\, A_{l-1}) &= \mathbb{P}(\hat{y}_l \neq y \,|\, A_{l-1}, \hat{y}_l \in \{\hat{y}_i\}_{i=1}^{l-1})\mathbb{P}(\hat{y}_l \in \{\hat{y}_i\}_{i=1}^{l-1}) \\ &\quad + \mathbb{P}(\hat{y}_l \neq y \,|\, A_{l-1}, \hat{y}_l \notin \{\hat{y}_i\}_{i=1}^{l-1})\mathbb{P}(\hat{y}_l \notin \{\hat{y}_i\}_{i=1}^{l-1}) \\ &\geq \mathbb{P}(\hat{y}_l \neq y \,|\, A_{l-1}, \hat{y}_l \notin \{\hat{y}_i\}_{i=1}^{l-1}) \\ &\geq \frac{m-l}{m-(l-1)}, \end{aligned}$$

where we used the fact that $\mathbb{P}\big(\hat{y}_l \neq y \mid A_{l-1}, \hat{y}_l \in \{\hat{y}_i\}_{i=1}^{l-1}\big) = 1$. Plugging back into (27) and using the induction hypothesis $\mathbb{P}(A_{l-1}) \geq (m - (l-1))/m$, we obtain

$$\mathbb{P}(A_l) \geq \frac{m-l}{m}.$$

This completes the proof of the induction. The proof of the lemma is completed by noticing that $\mathbb{P}(\hat{y}_l \neq y) \geq \mathbb{P}(A_l)$. $\qquad\square$

*Remark* C.2. Note that the above lower bound is sharp. An optimal algorithm draws $\hat{y}_l$ uniformly from $\{1, \ldots, m\} \setminus \{\hat{y}_1, \ldots, \hat{y}_{l-1}\}$.

## C.1. Proof of Theorem 5.3

The proof proceeds by randomizing the data distribution and providing bounds in expectation with respect to this randomness, which implies the existence of a worst-case distribution with the same bound.

**Setup and Definitions.** Define $I := \lfloor 1/\gamma^2 \rfloor$ and $m := \lfloor 1/\alpha \rfloor$, and let $e_i$ be the $i$th standard basis vector of $\mathbb{R}^D$. Define the four sets $\mathcal{X}_i = \{x^{iI/4+1}, \ldots, x^{(i+1)I/4}\}$, where we assume without loss of generality that $I$ is divisible by 4. We define $\mathcal{D}$ as follows: let

$$p_1 := (1 - \varepsilon^*)(1 - \delta), \quad p_2 := (1 - \varepsilon^*)\delta, \quad p_3 := \varepsilon^*(1 - \delta), \quad p_4 := \varepsilon^*\delta$$

for some $\delta \in (0, 1)$ to be fixed later. With probability $p_i$, we draw $x$ uniformly from $\mathcal{X}_i$. Let $\mathbb{Y}_m \subseteq \mathcal{Y}^N$ be an arbitrary set of $m$ distinct sequences of length $N$. To determine the ground-truth labels $y^*(x)$, we draw them independently from $\mathrm{Unif}(\mathbb{Y}_m)$ for $x \in \mathcal{X}_1 \cup \mathcal{X}_2$ and from $\mathrm{Unif}(\mathcal{Y}^N)$ for $x \in \mathcal{X}_3 \cup \mathcal{X}_4$. We will also use the notation $(m_i)_{i=1}^4$ where $m_i = m$ for $i \in \{1, 2\}$ and $m_i = k^N$ for $i \in \{3, 4\}$. With this definition, the label for $x \in \mathcal{X}_i$ is drawn uniformly from $m_i$ sequences. Note that $m_i \geq 2$ for all $i$.

We define $q(\cdot \mid x)$ to be $\mathrm{Unif}(\mathcal{Y}_m)$ for $x \in \mathcal{X}_1 \cup \mathcal{X}_2$ and $\mathrm{Unif}(\mathcal{Y}^N)$ for $x \in \mathcal{X}_3 \cup \mathcal{X}_4$. Therefore, $q(y^*(x) \mid x) = m^{-1}$ if $x \in \mathcal{X}_1 \cup \mathcal{X}_2$ and $q(y^*(x) \mid x) = k^{-N}$ otherwise. One can then verify that $q$ satisfies

$$\mathcal{Q}_q(\varepsilon) = \begin{cases} k^{-N} & \text{if} \quad \varepsilon < \varepsilon^* \\ m^{-1} & \text{if} \quad \varepsilon \geq \varepsilon^*. \end{cases}$$

For every $j \in [N]$ and $i \in [I]$, the feature map $\phi$ is defined as

$$\phi(x^i, y_{1:j}) = \phi(x^i, y_j) = e_{k(i-1)+y_j}.$$

As a result, $\langle \phi(x^i, y_{1:j}), \phi(x^u, y_{1:v}) \rangle = \mathbb{1}[i = u]\mathbb{1}[y_j = y_v]$. It is straightforward to verify that $w^* := \gamma \sum_{i=1}^I \phi(x^i, y^*(x^i))$ satisfies Assumption 2.1.

**Lower Bound on Number of Samples.** Let $m(x)$ denote the number of reward queries made on context $x$. Let $x \sim \mathcal{D}$ independent of $S$. We have

$$\mathbb{P}(\hat{y}(S)(x) \neq y^*(x)) = \sum_{i=1}^4 \mathbb{P}(\hat{y}(S)(x) \neq y^*(x) \mid x \in \mathcal{X}_i)\mathbb{P}(x \in \mathcal{X}_i). \tag{28}$$

Moreover,

$$\begin{aligned} \mathbb{P}(\hat{y}(S)(x) \neq y^*(x) \mid x \in \mathcal{X}_i) &= \mathbb{E}[\mathbb{P}(\hat{y}(S)(x) \neq y^*(x) \mid S, x \in \mathcal{X}_i) \mid x \in \mathcal{X}_i] \\ &\geq \mathbb{E}\left[\mathbb{1}[x \notin S] \cdot \frac{m_i - 1}{m_i} + \mathbb{1}[x \in S] \cdot \frac{m_i - 1 - m(x)}{m_i} \vee 0 \mid x \in \mathcal{X}_i\right] \\ &= \mathbb{P}(x \notin S \mid x \in \mathcal{X}_i) \cdot \frac{m_i - 1}{m_i} + \mathbb{E}\left[\frac{\mathbb{1}[x \in S](m_i - 1 - m(x))}{m_i} \vee 0 \mid x \in \mathcal{X}_i\right]. \end{aligned} \tag{29}$$

To obtain a lower bound on the number of samples, we remove the second term from (29) to obtain

$$\mathbb{P}(\hat{y}(S)(x) \neq y^*(x) \mid x \in \mathcal{X}_i) \geq \frac{m_i - 1}{m_i}\mathbb{P}(x \notin S \mid x \in \mathcal{X}_i)$$

where the second inequality follows from Lemma C.1. Note that if $\boldsymbol{x} \in \mathcal{X}_i$, then the probability that it is drawn once is $\frac{4p_i}{I}$, and by a union bound, the probability that it is drawn at least once is at most $\frac{4Mp_i}{I}$. Therefore $\mathbb{P}(\boldsymbol{x} \notin S \,|\, \boldsymbol{x} \in \mathcal{X}_i) \geq (1 - \frac{4p_iM}{I}) \vee 0$. Thus

$$\mathbb{P}(\hat{\boldsymbol{y}}(S)(\boldsymbol{x}) \neq \boldsymbol{y}^*(\boldsymbol{x})) \geq \sum_{i=1}^{4} p_i \cdot \frac{m_i - 1}{m_i} \cdot \left(1 - \frac{4p_iM}{I}\right) \vee 0.$$

Since each term is non-negative, we can remove $i = 1$ and $i = 3$ terms, from the lower bound, and write

$$\mathbb{P}(\hat{\boldsymbol{y}}(S)(\boldsymbol{x}) \neq \boldsymbol{y}^*(\boldsymbol{x})) \geq \frac{p_2}{2}\left(1 - \frac{4p_2M}{I}\right) + \frac{p_4}{2}\left(1 - \frac{4p_4M}{I}\right) \geq \frac{p_2 + p_4}{2}\left(1 - \frac{4(p_1 \vee p_2)M}{I}\right),$$

where we used $m_i \geq 2$. For the probability of error to be less than $\varepsilon$ we need we need

$$M \geq \frac{I}{4(p_2 \vee p_4)}\left(1 - \frac{2\varepsilon}{p_2 + p_4}\right).$$

Recall $p_2 = (1 - \varepsilon^*)\delta$ and $p_4 = \varepsilon^*\delta$. For $\varepsilon \leq 1/2 - c$ for some absolute constant $c > 0$, let $\delta = 2\varepsilon(1 + 2c) \leq 1$. Then, the above implies

$$M \geq \frac{I}{4\delta((1 - \varepsilon^*) \vee \varepsilon^*)} \cdot \frac{2c}{1 + 2c} \geq \frac{cI}{4\delta(1 + 2c)} \gtrsim \frac{1}{\gamma^2\varepsilon},$$

which completes the proof of the sample lower bound.

**Lower Bound on Number of Reward Queries.**  For this part, we choose $\delta = 1/2$. To prove the lower bound on the number of reward queries, we go back to (29), and note that

$$\mathbb{P}(\hat{\boldsymbol{y}}(S)(\boldsymbol{x}) \neq \boldsymbol{y}^*(\boldsymbol{x}) \,|\, \boldsymbol{x} \in \mathcal{X}_i) \geq \frac{m_i - 1}{m_i} - \mathbb{E}\left[\frac{\mathbb{1}[\boldsymbol{x} \in S]m(\boldsymbol{x})}{m_i} \,|\, \boldsymbol{x} \in \mathcal{X}_i\right].$$

In particular, using the fact that $m_1 = m_2 = m$ and $m_3 = m_4 = k^N$, we can write,

$$\mathbb{P}(\hat{\boldsymbol{y}}(S)(\boldsymbol{x}) \neq \boldsymbol{y}^*(\boldsymbol{x}) \,|\, \boldsymbol{x} \in \mathcal{X}_1 \cup \mathcal{X}_2) \geq \frac{m - 1}{m} - \frac{\mathbb{E}[\mathbb{1}[\boldsymbol{x} \in S]m(\boldsymbol{x}) \,|\, \boldsymbol{x} \in \mathcal{X}_1 \cup \mathcal{X}_2]}{m},$$

and

$$\mathbb{P}(\hat{\boldsymbol{y}}(S)(\boldsymbol{x}) \neq \boldsymbol{y}^*(\boldsymbol{x}) \,|\, \boldsymbol{x} \in \mathcal{X}_3 \cup \mathcal{X}_4) \geq \frac{k^N - 1}{k^N} - \frac{\mathbb{E}[\mathbb{1}[\boldsymbol{x} \in S]m(\boldsymbol{x}) \,|\, \boldsymbol{x} \in \mathcal{X}_3 \cup \mathcal{X}_4]}{k^N},$$

Define $Z := \mathbb{E}[\mathbb{1}[\boldsymbol{x} \in S]m(\boldsymbol{x}) \,|\, \boldsymbol{x} \in \mathcal{X}_1 \cup \mathcal{X}_2]$ and $Z' := \mathbb{E}[\mathbb{1}[\boldsymbol{x} \in S]m(\boldsymbol{x}) \,|\, \boldsymbol{x} \in \mathcal{X}_3 \cup \mathcal{X}_4]$. Using (28), we obtain

$$\mathbb{P}(\hat{\boldsymbol{y}}(S)(\boldsymbol{x}) \neq \boldsymbol{y}^*(\boldsymbol{x})) \geq (1 - \varepsilon^*)\left(\frac{m - 1 - Z}{m}\right) \vee 0 + \varepsilon^*\left(\frac{k^N - 1 - Z'}{k^N}\right) \vee 0, \tag{30}$$

where we used the non-negativeness of each conditional probability. In particular, each term on the RHS must be bounded by the LHS, i.e.

$$\mathbb{P}(\hat{\boldsymbol{y}}(S)(\boldsymbol{x}) \neq \boldsymbol{y}^*(\boldsymbol{x})) \geq (1 - \varepsilon^*)\left(\frac{m - 1 - Z}{m}\right) \tag{31}$$

when considering only $\boldsymbol{x} \in \mathcal{X}_1 \cup \mathcal{X}_2$, and

$$\mathbb{P}(\hat{\boldsymbol{y}}(S)(\boldsymbol{x}) \neq \boldsymbol{y}^*(\boldsymbol{x})) \geq \varepsilon^*\left(\frac{k^N - 1 - Z'}{k^N}\right) \tag{32}$$

when considering only $\boldsymbol{x} \in \mathcal{X}_3$. Before proceeding, we observe that the number of reward queries is equal to $Q = \frac{(Z+Z')I}{2}$. This is because

$$Q = \mathbb{E}\left[\sum_{i=1}^{I} \mathbb{1}[\boldsymbol{x}^i \in S]m(\boldsymbol{x}^i)\right]$$

$$= \mathbb{E}\left[\sum_{\boldsymbol{x}^i \in \mathcal{X}_1 \cup \mathcal{X}_2} \mathbb{1}[\boldsymbol{x}^i \in S]m(\boldsymbol{x}^i)\right] + \mathbb{E}\left[\sum_{\boldsymbol{x}^i \in \mathcal{X}_3 \cup \mathcal{X}_4} \mathbb{1}[\boldsymbol{x}^i \in S]m(\boldsymbol{x}^i)\right]$$

$$= \frac{I}{2}\,\mathbb{E}[\mathbb{1}[\boldsymbol{x} \in S]m(\boldsymbol{x}) \,|\, \boldsymbol{x} \in \mathcal{X}_1 \cup \mathcal{X}_2] + \frac{I}{2}\,\mathbb{E}[\mathbb{1}[\boldsymbol{x} \in S]m(\boldsymbol{x}) \,|\, \boldsymbol{x} \in \mathcal{X}_3 \cup \mathcal{X}_4],$$

where we recall $\boldsymbol{x} \sim \mathcal{D}$ in the above.

Now, consider the following cases:

- First, consider the case where $(1 + 1/\log(1/\varepsilon))\varepsilon \geq \varepsilon^*$. We have

$$
\varepsilon \geq (1 - \varepsilon^*)\left(\frac{m - 1 - Z}{m}\right) + \varepsilon^*\left(\frac{k^N - 1 - Z'}{k^N}\right)
$$
$$
\geq \frac{m - 1 - (Z + Z')}{m}
$$

  which implies $Z + Z' \geq m(1 - \varepsilon - m^{-1}) \gtrsim m$ where we used $\varepsilon < 1/2$ and $m^{-1} < 1/2$. Therefore, the number of reward queries is $Q = I(Z + Z')/2 \gtrsim mI \gtrsim \alpha^{-1}/\gamma^2$. Note that for the range of $\varepsilon$ here we have $\mathcal{Q}_q((1 + 1/\log(1/\varepsilon))\varepsilon) = \alpha$, therefore $Q \gtrsim \mathcal{Q}_q((1 + 1/\log(1/\varepsilon))\varepsilon)^{-1}/\gamma^2$.

- Second, assume $(1 + 1/\log(1/\varepsilon))\varepsilon < \varepsilon^*$. Using (32), we have

$$
\varepsilon \geq \varepsilon^*\left(\frac{k^N - 1 - Z'}{k^N}\right)
$$

  and consequently

$$
Z' \geq k^N\left(1 - k^{-N} - \frac{\varepsilon}{\varepsilon^*}\right) \geq k^N\left(1 - k^{-N} - \frac{1}{1 + 1/\log(1/\varepsilon)}\right).
$$

  We want the multiplier of $k^N$ on the RHS to be at least some absolute constant $c$, and W.L.O.G. we assume $N$ is sufficiently large such that $k^{-N} \leq c$. We need to guarantee $1 + 1/\log(1/\varepsilon) \geq (1 - 2c)^{-1}$. This inequality is always satisfied for $\varepsilon < 1/2$ as long as $c$ is a sufficiently small absolute constant, e.g. $c = 0.2$ satisfies this condition by numerical calculation. With this, we have $Z' \geq ck^N$ and $Q = I(Z + Z')/2 \gtrsim k^N I \gtrsim \mathcal{Q}_q((1 + 1/\log(1/\varepsilon))\varepsilon)^{-1}/\gamma^2$, which completes the proof.

$\qquad\qquad\qquad\qquad\qquad\qquad\qquad\qquad\qquad\qquad\qquad\qquad\qquad\qquad\qquad\qquad\qquad\qquad\qquad\qquad\qquad\quad\square$

## C.2. Proof of Theorem 5.4

We use the same setup as that of Appendix C.1. Let $\tilde{\pi} := \pi_\alpha$ if $\varepsilon \geq 1 - \pi_\alpha$, and $\tilde{\pi} := 1 - \varepsilon(1 + c)$ if $\varepsilon(1 + 2c) \leq 1 - \pi_\alpha$ for some absolute constant $c > 0$. We define $p_1 := \tilde{\pi}(1 - \delta)$, $p_2 := \tilde{\pi}\delta$, and $p_3 := 1 - \tilde{\pi}$, where $\delta$ will be chosen later. Note that with this new definition, we still have $\mathbb{P}_{\boldsymbol{x}}(q(\boldsymbol{y}^*(\boldsymbol{x}) \mid \boldsymbol{x}) \geq \alpha) \geq \pi_\alpha$.

Here, we use $S$ to denote the previous samples as well the learner's prediction on them and the obtained reward. We begin with the same decomposition as in (28). This time, for each conditional probability, we write

$$
\mathbb{P}(\hat{\boldsymbol{y}}(S)(\boldsymbol{x}) \neq \boldsymbol{y}^*(\boldsymbol{x}) \mid \boldsymbol{x} \in \mathcal{X}_i) = \mathbb{E}[\mathbb{P}(\hat{\boldsymbol{y}}(S)(\boldsymbol{x}) \neq \boldsymbol{y}^*(\boldsymbol{x}) \mid S, \boldsymbol{x} \in \mathcal{X}_i) \mid \boldsymbol{x} \in \mathcal{X}_i]
$$
$$
\geq \mathbb{E}\left[\frac{m_i - 1 - m(\boldsymbol{x})}{m_i} \mid \boldsymbol{x} \in \mathcal{X}_i\right] \vee 0,
$$

where $m(\boldsymbol{x})$ is the number of times $\boldsymbol{x}$ has appeared in $S$, and we used Lemma C.1 for the inequality. Moreover, when $\boldsymbol{x} \in \mathcal{X}_i$, the probability that an individual draw in $S$ is equal to $\boldsymbol{x}$ is $4p_i/I$, hence $\mathbb{E}[m(\boldsymbol{x}) \mid \boldsymbol{x} \in \mathcal{X}_i] = \frac{4Tp_i}{I}$ where we recall $T$ is the number of rounds (hence samples). Therefore

$$
\mathbb{P}(\hat{\boldsymbol{y}}(S)(\boldsymbol{x}) \neq \boldsymbol{y}^*(\boldsymbol{x}) \mid \boldsymbol{x} \in \mathcal{X}_i) \geq \left(\frac{(m_i - 1)I - 4Tp_i}{m_i I}\right) \vee 0. \tag{33}
$$

**Lower Bound on the Number of Samples.**   We consider the following two cases:

- Consider the case $(1 + 1/\log(1/\varepsilon))\varepsilon \geq \varepsilon^*$. Using (33) and (28), we have

$$\mathbb{P}(\hat{\boldsymbol{y}}(S)(\boldsymbol{x}) \neq \boldsymbol{y}^*(\boldsymbol{x})) \geq \sum_{i=1}^{4} p_i \cdot \left( \frac{(m_i - 1)I - 3Tp_i}{m_i I} \right) \vee 0. \tag{34}$$

$$\geq p_2 \left( \frac{(m-1)I - 4Tp_2}{mI} \right) + p_4 \left( \frac{(k^N - 1)I - 4Tp_4}{k^N I} \right) \tag{35}$$

$$\geq (p_2 + p_4)\left( \frac{(m-1)I - 4T(p_2 \vee p_4)}{mI} \right) \tag{36}$$

where we used the fact that $k^N \geq m$. Recall $p_2 = (1 - \varepsilon^*)\delta$, and $p_4 = \varepsilon^*\delta$. Then we can rearrange the above terms to write

$$T \geq \frac{mI}{4\delta((1-\varepsilon^*) \vee \varepsilon^*)}\left( 1 - m^{-1} - \frac{\varepsilon}{\delta} \right) \geq \frac{mI}{8\delta}\left( 1 - m^{-1} - \frac{\varepsilon}{\delta} \right).$$

By choosing $\delta = 2\varepsilon < 1$ and using the fact that $m \geq 2$, we get $T \gtrsim mI/\varepsilon \gtrsim m/(\gamma^2\varepsilon)$. Note that for $(1 + 1/\log(1/\varepsilon))\varepsilon \geq \varepsilon^*$ we have $\mathcal{Q}_q(\varepsilon) = m^{-1}$. Hence we can write $T \gtrsim \mathcal{Q}_q((1 + 1/\log(1/\varepsilon))\varepsilon)^{-1}/(\gamma^2\varepsilon)$, which completes the proof of this case.

- For the case where $(1 + 1/\log(1/\varepsilon))\varepsilon < \varepsilon^*$, we use (34) with $i = 4$, where we recall $m_4 = k^N$ and $p_4 = \varepsilon^*\delta$. We then have

$$\varepsilon \geq \varepsilon^*\delta\left( \frac{(k^N - 1)I - 4T\delta\varepsilon^*}{k^N I} \right)$$

and consequently

$$T \geq \frac{k^N I}{4\varepsilon^*\delta}\left( 1 - k^{-N} - \frac{\varepsilon}{\varepsilon^*\delta} \right).$$

The argument is similar to that of Appendix C.1. Let $c$ denote some absolute constant and assume $N$ is sufficiently large such that $k^{-N} \leq c$. Then, we can choose $\delta = (\varepsilon/\varepsilon^*) \cdot (1 - 2c)^{-1}$ to guarantee $T \gtrsim k^N I/\varepsilon$. We only need to make sure $\delta \leq 1$, and it is sufficient to show $1 + 1/\log(1/\varepsilon) \geq (1 - 2c)^{-1}$. For $\varepsilon < 1/2$, this inequality always holds with $c = 0.2$. With this, we have $T \gtrsim k^N/(\gamma^2\varepsilon)$, and we conclude the proof of this case by recalling that for $(1 + 1/\log(1/\varepsilon))\varepsilon < \varepsilon^*$ we have $\mathcal{Q}_q((1 + 1/\log(1/\varepsilon))\varepsilon) = k^{-N}$.

**Lower Bound on the Number of Mistakes.** For the lower bound on the number of mistakes, we choose $\delta = 1/2$ and unify $\mathcal{X}_1$ with $\mathcal{X}_2$ and $\mathcal{X}_3$ with $\mathcal{X}_4$. For sample $t$, define $S_t$ to be the collection of previous samples as well as their corresponding predictions and rewards. Then,

$$\mathbb{P}\left( \hat{\boldsymbol{y}}(S_t)(\boldsymbol{x}^{(t)}) \neq \boldsymbol{y}^*(\boldsymbol{x}^{(t)}) \right) = \mathbb{P}\left( \hat{\boldsymbol{y}}(S_t)(\boldsymbol{x}^{(t)}) \neq \boldsymbol{y}^*(\boldsymbol{x}^{(t)}) \mid \boldsymbol{x}^{(t)} \in \mathcal{X}_1 \cup \mathcal{X}_2 \right)(1 - \varepsilon^*)$$
$$+ \mathbb{P}\left( \hat{\boldsymbol{y}}(S_t)(\boldsymbol{x}^{(t)}) \neq \boldsymbol{y}^*(\boldsymbol{x}^{(t)}) \mid \boldsymbol{x}^{(t)} \in \mathcal{X}_3 \cup \mathcal{X}_4 \right)\varepsilon^*.$$

Using Lemma C.1 and arguments similar to the above paragraph, we have

$$\mathbb{P}\left( \hat{\boldsymbol{y}}(S_t)(\boldsymbol{x}^{(t)}) \neq \boldsymbol{y}^*(\boldsymbol{x}^{(t)}) \mid \boldsymbol{x}^{(t)} \in \mathcal{X}_1 \cup \mathcal{X}_2 \right) \geq \frac{m - 1 - 2(t-1)(1 - \varepsilon^*)}{mI} \vee 0,$$

and

$$\mathbb{P}\left( \hat{\boldsymbol{y}}(S_t)(\boldsymbol{x}^{(t)}) \neq \boldsymbol{y}^*(\boldsymbol{x}^{(t)}) \mid \boldsymbol{x}^{(t)} \in \mathcal{X}_1 \cup \mathcal{X}_2 \right) \geq \frac{k^N - 1 - 2(t-1)\varepsilon^*}{k^N I} \vee 0. \tag{37}$$

Therefore,

$$\mathbb{E}\left[ \sum_{t=1}^{T} \hat{\boldsymbol{y}}(S_t)(\boldsymbol{x}^{(t)}) \neq \boldsymbol{y}^*(\boldsymbol{x}^{(t)}) \right] = \sum_{t=1}^{T} \mathbb{P}\left( \hat{\boldsymbol{y}}(S_t)(\boldsymbol{x}^{(t)}) \neq \boldsymbol{y}^*(\boldsymbol{x}^{(t)}) \right)$$

$$\geq \sum_{t=1}^{T} (1 - \varepsilon^*) \cdot \frac{(m-1)I - 2(t-1)(1-\varepsilon^*)}{mI} \vee 0 + \sum_{t=1}^{T} \varepsilon^* \cdot \frac{(k^N - 1)I - 2(t-1)\varepsilon^*}{k^N I} \vee 0$$

Note that for all $t \lesssim k^N I/\varepsilon^*$, the terms inside the second sum above are $\Omega(\varepsilon^*)$. Therefore, the sum above for $T \geq k^N/\varepsilon^*$ is at least $k^N I \gtrsim k^N/\gamma^2$, completing the proof. $\qquad\square$

## C.3. Proof of Theorem 5.5

We reuse $I := \lfloor 1/\gamma^2 \rfloor$ as well as the basis $(\boldsymbol{e}_i)$ and the definition of the feature map $\phi$ from Appendix C.1. This time, we define $\mathcal{X}_1 = \{\boldsymbol{x}^1, \ldots, \boldsymbol{x}^{I/2}\}$ and $\mathcal{X}_2 = \{\boldsymbol{x}^{I/2+1}, \ldots, \boldsymbol{x}^I\}$, where we assume without loss of generality that $I$ is even. We construct the distribution as follows. With probability $p_1 := 1 - \delta$, we draw $\boldsymbol{x}$ uniformly from $\mathcal{X}_1$, and with probability $p_2 = \delta$, we draw it uniformly from $\mathcal{X}_2$. We randomize the distribution of the labels, and draw $\boldsymbol{y}^*(\boldsymbol{x}^i) \sim \text{Unif}(\mathcal{Y}^N)$ independently. We recall that $\boldsymbol{w}^* = \gamma \sum_{i=1}^I \phi(\boldsymbol{x}^i, \boldsymbol{y}^*(\boldsymbol{x}^i))$ satisfies Assumption 2.1.

Let $S$ denote the training set, and $q(S)(\cdot \mid \boldsymbol{x})$ be the output of the algorithm. Consider the decomposition

$$\mathbb{P}(q(S)(\boldsymbol{y}^*(\boldsymbol{x}) \mid \boldsymbol{x}) \le \alpha) = \sum_{i=1}^2 p_i \mathbb{P}(q(S)(\boldsymbol{y}^*(\boldsymbol{x}) \mid \boldsymbol{x}) \mid \boldsymbol{x} \in \mathcal{X}_i).$$

We choose $i = 2$, and use the non-negativity of probability to write

$$\mathbb{P}(q(S)(\boldsymbol{y}^*(\boldsymbol{x}) \mid \boldsymbol{x}) \le \alpha) \ge p_2 \mathbb{P}(q(S)(\boldsymbol{y}^*(\boldsymbol{x}) \mid \boldsymbol{x}) \le \alpha \mid \boldsymbol{x} \in \mathcal{X}_2)$$
$$\ge p_2 \mathbb{P}(q(S)(\boldsymbol{y}^*(\boldsymbol{x}) \mid \boldsymbol{x}) \le \alpha \mid \boldsymbol{x} \in \mathcal{X}_2, \boldsymbol{x} \notin S) \mathbb{P}(\boldsymbol{x} \notin S \mid \boldsymbol{x} \in \mathcal{X}_2). \tag{38}$$

By a union bound, we have $\mathbb{P}(\boldsymbol{x} \notin S \mid \boldsymbol{x} \in \mathcal{X}_2) \ge 1 - \frac{2p_2 n}{I}$. Further, note that conditioned on $\boldsymbol{x} \notin S$, $\boldsymbol{y}^*(\boldsymbol{x})$ is independent from $q(S)(\cdot \mid \boldsymbol{x})$. Therefore,

$$\mathbb{P}(q(S)(\boldsymbol{y}^*(\boldsymbol{x}) \mid \boldsymbol{x}) > \alpha, \boldsymbol{x} \in \mathcal{X}_2, \boldsymbol{x} \notin S) \le \alpha^{-1} \mathbb{E}[q(S)(\boldsymbol{y}^*(\boldsymbol{x}) \mid \boldsymbol{x}) \mid \boldsymbol{x} \in \mathcal{X}_2, \boldsymbol{x} \notin S] \quad \text{(Markov Inequality)}$$

$$\le \alpha^{-1} \mathbb{E}\left[ \frac{1}{k^N} \sum_{\boldsymbol{y} \in \mathcal{Y}^N} q(S)(\boldsymbol{y} \mid \boldsymbol{x}) \mid \boldsymbol{x} \in \mathcal{X}_2, \boldsymbol{x} \notin S \right] \quad \text{(Expectation over } \boldsymbol{y}^*(\boldsymbol{x}))$$

$$\le \alpha^{-1} k^{-N},$$

where for the last inequality we used the fact that the distribution $q(S)(\cdot \mid \boldsymbol{x})$ must sum to 1. Plugging this back into (38), we obtain

$$\mathbb{P}(q(S)(\boldsymbol{y}^*(\boldsymbol{x}) \mid \boldsymbol{x}) \le \alpha) \ge \delta(1 - \alpha^{-1} k^{-N})(1 - 2\delta n/I).$$

By choosing $\delta = I/(4n)$, we have

$$\mathbb{E}_{\boldsymbol{y}^*}[\mathbb{P}(q(S)(\boldsymbol{y}^*(\boldsymbol{x}) \mid \boldsymbol{x}) \le \alpha \mid \boldsymbol{y}^*)] \ge \frac{I(1 - \alpha^{-1} k^{-N})}{8n} \gtrsim \frac{1 - \alpha^{-1} k^{-N}}{\gamma^2 n},$$

where we choose $\boldsymbol{y}^*$ to denote the vector of all sequences $(\boldsymbol{y}^*(\boldsymbol{x}^i))_{i=1}^I$. The fact that the expectation over labels provides this lower bound, implies the existence of at least one set of label assignments that satisfies this lower bound, namely

$$1 - \pi_\alpha = \mathbb{P}(q(S)(\boldsymbol{y}^*(\boldsymbol{x}) \mid \boldsymbol{x}) \le \alpha \mid \boldsymbol{y}^*) \gtrsim \frac{1 - \alpha^{-1} k^{-N}}{\gamma^2 n}.$$

$\square$

