# OpenReview forum: "Post-Training with Policy Gradients: Optimality and the Base Model Barrier"
_ICML.cc/2026/Conference — ICML 2026 spotlight_

### Official Review · Reviewer_UQxp · 2026-03-04

**Soundness:** 4
**Presentation:** 2
**Significance:** 3
**Originality:** 3
**Overall Recommendation:** 4
**Confidence:** 3

**Summary:**

This paper mainly studies the sample complexity of current policy gradient RL post-training algorithms w.r.t. the base model distribution, with linear autoregressive models. The authors establish minimax-optimal reward-query bounds that are governed by the likelihood quantile, which captures how much probability mass the base model assigns to correct solutions.
The results highlight a support barrier: with only outcome rewards, pushing performance beyond the base model’s effective support can require exponentially many reward queries in the sequence length.
The authors further analyze the process-reward setting and show that token-level reward can mitigate this exponential barrier.

**Compliance With Llm Reviewing Policy:**

Affirmed.

**Final Justification:**

I maintain my original recommendation of weak accept. The paper
  is technically strong and offers a useful theoretical
  perspective on how the base model distribution affects the
  sample complexity of policy-gradient post-training, especially
  through the likelihood-quantile view and the contrast between
  outcome and process rewards. I find the soundness strong, and
  the originality/significance good, since the results clarify an
  important limitation of RLVR-style training. The rebuttal addressed my main technical questions
  about the biased off-policy estimator, the role of baselines,
  and the experimental setup, and the proposed clarifications/
  additional experiments are helpful. However, the relaxed
  assumptions still seem somewhat impractical for real LLM post-
  training, so the rebuttal resolves my questions but does not
  substantially change my evaluation. Overall, I remain positive
  but keep the weak accept score.

**Key Questions For Authors:**

About the off-policy IS ratio: In Section 3, the paper suggests setting the ratio to 1 for a better (minimax-optimal) convergence rate, and also uses a simple estimator without a baseline. Could the authors clarify these two design choices and their practical implications? In particular, setting the IS ratio to 1 makes the gradient estimate biased. Does this mean that, in practice, when the ratio has very high variance, it can be preferable to optimize a biased surrogate rather than the corrected off-policy objective? Also, the paper argues that adding a baseline does not improve the rate under Assumption 2.1. Could the authors explain why baseline-based variance reduction does not translate into better sample complexity in this setting (e.g., does it only affect constants)?

**Limitations:**

yes

**Strengths And Weaknesses:**

## Strengths
1. Provides rigorous and comprehensive theoretical analysis on how the base model distribution controls post-training sample complexity.
2. Provides a useful lens on the debate of whether RLVR incentivizes reasoning capability beyond the base model.

## Weaknesses
1. Linear model assumption. The results are derived for linear autoregressive models, so it’s not obvious how much carries over to real LLM post-training with strong model mismatch, which limits the applicability of this paper.
2. Assumption 2.1 is strong. It assumes a unique correct sequence and a prefix-level margin at every step against all wrong tokens, which feels much stricter than typical RLVR tasks where multiple solutions exist and correctness is more flexible.
3. Experiment. Key setup details are mostly in the appendix, and the synthetic construction seems tailored to the theory; also, PG uses extra implementation choices (batch baseline, ±1reward) that should be emphasized since they can affect the observed behavior.

---

> ### Author Rebuttal · Authors · 2026-03-31
>
> We thank the reviewer for their time and their detailed evaluation of our work. We address their questions and concerns below.
>
> ---
> > *Linear model assumption. The results are derived for linear autoregressive models … Assumption 2.1 is strong. It assumes a unique correct sequence and a prefix-level margin at every step …*
>
> We would like to highlight the role of each core assumption and provide certain relaxations below.
>
> 1. The correct next token is $\gamma$-separated from any incorrect next token using the feature map $\phi$ on partially correct tokens so far. This assumption is seemingly strong but **not critical**, as we can relax it as follows.
>
>
> >   **Assumption:** For every $\varepsilon > 0$, there exists $\boldsymbol{w}$ such that $\Vert \boldsymbol{w} \Vert \leq \mathcal{R}(\varepsilon)$ and $\mathbb{E}[-\log p\_{\boldsymbol{w}}(\boldsymbol{y}^\*(\boldsymbol{x}) | \boldsymbol{x})] \leq \varepsilon$.
>
>
> With this new condition, we can replace $1/(\gamma^2 \varepsilon)$ in all upper bound statements with $\mathcal{R}(\varepsilon)^2/\varepsilon$. Note that the overall message, including the definition of LQ, remains unchanged.
>
> 2. The feature map $\phi$ is frozen. Such a condition is currently inevitable for a tractable analysis of post-training dynamics; see e.g. prior works Chen et al. (2025), Foster et al. (2025) where they point out that empirically, post-training might stay in the lazy regime where only last layer features evolve (Malladi et al., 2025).
>
> 3. Each $\boldsymbol{x}$ is associated with a unique correct response $\boldsymbol{y}^\*(\boldsymbol{x})$. We agree that extending this to a setting where we only have a subset $\mathcal{Y}^\*(\boldsymbol{x})$ of correct responses per input $\boldsymbol{x}$ is a very interesting future direction. A priori, it is not clear which correct solution will be preferred by the model after post-training. We expect that redefining the notion of Likelihood Quantile, where likelihood refers to the probability of generating any correct response, would be useful in that setting. We will highlight this as an open problem in the final version.
>
>
>
>
> ---
> > *Experiment. Key setup details are mostly in the appendix, and the synthetic construction seems tailored to the theory …*
>
>
> We thank the reviewer for their suggestion. In the final version, we will move the description of the experimental setup to the main text, and include a link to the code repository in the de-anonymyzed version.
>
> To provide intuitions beyond the synthetic task above, we have conducted additional experiments, and we will include figures illustrating the Likelihood Quantile function of *Qwen3-8B on the MATH500 dataset*.
>
> We have also carried out additional simulation studies where we compare the *evolution of the average likelihood of initially off-support samples* between *PG with process rewards* and *PG with outcome rewards*, demonstrating how unlike PG-OR, PG-PR is able to go beyond the base model support.
>
> We will include these additional experiments in the final version.
>
>
> ---
>
> > *In Section 3, the paper suggests setting the ratio to 1 for a better (minimax-optimal) convergence rate …  Also, the paper argues that adding a baseline does not improve the rate under Assumption 2.1. Could the authors explain why …*
>
> We thank the reviewer for the interesting questions. We provide some intuition on each design choice below.
>
> *On Importance Weights*: We expect that in certain settings, the high variance of the importance weights will slow down convergence. The reason why biased gradient estimation in our setting is sufficient is due to the particular problem structure; our ultimate goal is to maximize log-likelihood of correct responses. In this setting, PG without importance weights can be seen as an online convex optimization of a weighted negative log-likelihood loss, where the weights come from the behavior policy.
>
> *On Baseline*: Beyond minimax optimality, another intuition for why baselines will not significantly help in our setting is the following. There is a significant information asymmetry between correct and incorrect answer guesses. If a sampled answer is incorrect, it is only telling the model to avoid a single response in an exponentially large space of potential response set. Therefore, *intuitively*, gradient updates on negative samples can be ignored compared to gradient updates on positive samples, without affecting the convergence rate’s order of magnitude.
>
> ---
>
> F. Chen et al. “The coverage principle: How pre-training enables post-training.” ICLR 2026.
>
> D. Foster et al. “Is a good foundation necessary for efficient reinforcement learning? The computational role of the base model in exploration.” COLT 2025.
>
> S. Malladi et al. “A kernel-based view of language model fine-tuning.” ICLR 2023.

---

> > ### Author Rebuttal · Reviewer_UQxp · 2026-04-03
> >
> > Thank the authors for the reply, my concerns are fully resolved. Overall, I think the paper offers a novel and interesting perspective on the role of the base model in RLVR training. However, I still find the $\gamma$-margin assumption, even in its relaxed form, somewhat impractical, which limits the paper's applicability. Therefore, I will maintain my original score.

---

### Official Review · Reviewer_o3tZ · 2026-03-08

**Soundness:** 3
**Presentation:** 3
**Significance:** 3
**Originality:** 3
**Overall Recommendation:** 4
**Confidence:** 3

**Summary:**

This paper provides a theoretical characterization of how the quality of the base model fundamentally limits the effectiveness of post-training with policy-gradient methods. In a linearly parameterized autoregressive setting, the authors compare outcome-reward and process-reward post-training both statistically and computationally. Their analysis shows that outcome-reward training suffers from a base model barrier, whereas process-reward training can overcome this limitation under more favorable conditions. To formalize this phenomenon, the paper introduces the Likelihood Quantile (LQ), which offers a refined way to measure how improvable a base model is. The theoretical results are further supported by minimax optimality guarantees and numerical experiments.

**Compliance With Llm Reviewing Policy:**

Affirmed.

**Final Justification:**

The author's response was clear: My concerns about the superiority of PG-PR was resolved. Considering the simplicity of their setting (linear model), I maintained my initial positive score.

**Key Questions For Authors:**

* Do the authors have an intuition for why PG-PR does not seem to outperform SGD with adaptive learning rates? Does this suggest that one would need a richer problem setting to see a clearer separation? More broadly, do the authors conjecture that PG-PR may be fundamentally unable to outperform well-tuned SGD in a generalizable sense?
* Why is the binary (0)-(1) reward in Eq. (4) considered optimal in this framework? In particular, why is it sufficient to use this terminal reward formulation, rather than an undiscounted sum of per-step rewards over the trajectory?

**Limitations:**

yes

**Strengths And Weaknesses:**

Strengths:
* The paper offers an appealing and fairly deep theoretical characterization of how the base model constrains the effectiveness of post-training.
* It is interesting that the paper compares outcome-reward and process-reward training from both computational and statistical perspectives, and shows that (i) outcome-reward training suffers from a base model barrier, while (ii) process-reward training can break this barrier.
* To explain this rigorously, the authors introduce Likelihood Quantile (LQ) as a way to quantify the improvability of the base model. This seems to be a finer and clearer notion than the “coverage” measures used in some prior works.
* The paper strengthens its contribution by establishing minimax optimality results.
* The theory is complemented by numerical experiments, which help illustrate the main phenomena.

Weaknesses:
* The superiority of PG-PR is only established relative to constant-step-size SGD, not to SGD with adaptive or near-optimal learning rates.
* The theoretical setting is still restricted, especially in that it assumes (i) linear parameterization, and (ii) binary-valued rewards.

---

> ### Author Rebuttal · Authors · 2026-03-31
>
> We thank the reviewer for their time and their detailed evaluation of our work. We address their questions and concerns below.
>
> ---
> > *The superiority of PG-PR is only established relative to constant-step-size SGD, not to SGD with adaptive or near-optimal learning rates … Do the authors have an intuition for why PG-PR does not seem to outperform SGD with adaptive learning rates?*
>
> We thank the reviewer for pointing out a possible confusion that we need to better clarify in the final version. First, we remark that in most settings, PG-PR outperforms PG-OR with *the same base model*, because $\mathcal{Q}^{\mathrm{TL}}\_q(\varepsilon) \geq \mathcal{Q}\_q(\varepsilon)$. This gap can be significant. The PG-PR guarantee of Theorem 4.1 *never suffers exponential dependency on $N$*, unlike the PG-OR guarantee of Corollary 3.6.
>
> The specific result the reviewer is referring to here is Corollary 4.2. Here, the goal is to find a setting where SGD pre-training + PG-PR post-training has a strictly better *sample and reward query complexity* compared to: (1) only SGD pre-training, and (2) only PG-PR post-training. We can show this when the SGD pre-training part is done with a constant learning rate, in which case we have the following supervised sample complexity $n$ and reward query complexity $Q$:
>
> | Algorithm | $n$ | $Q$ |
> | --- | --- | --- |
> | Only SGD | $N/(\gamma^2 \varepsilon)$ | - |
> | Only PG-PR | - | $(Nk + \varepsilon^{-1}) / \gamma^2$ |
> | SGD + PG-PR | $1/(\gamma^2 \varepsilon)$ | $(N + \varepsilon^{-1})/\gamma^2$ |
>
> Thus, SGD pre-training + PG-PR post-training has an order of magnitude improvement in both the sample complexity $n$ when compared to SGD and the reward query complexity $Q$ when compared to PG-PR.
>
> If SGD is run with adaptive learning rate however, the supervised sample complexity becomes $1/(\gamma^2 \varepsilon)$ both for SGD alone and SGD + PG-PR. That said, running PG-PR after SGD will continue to improve the accuracy, without hitting an exponential barrier like PG-OR. We will clarify this in the final version.
>
> ---
>
> > *The theoretical setting is still restricted, especially in that it assumes (i) linear parameterization, and (ii) binary-valued rewards.*
>
>
> Point (i): While we agree that this assumption is generally restrictive compared to practice, such a condition is currently inevitable for a tractable analysis of post-training dynamics; see e.g. prior works Chen et al. (2025), Foster et al. (2025) where they point out that empirically, post-training might stay in the lazy regime where only last layer features evolve (Malladi et al., 2025).
>
> Point (ii): While the study of general rewards is certainly an interesting direction for future work, we believe that for our setting, binary rewards are less of a severe restriction, as they are often used in practice for certain RLVR reasoning domains (Guo et al., 2025).
>
> ---
>
> > *Why is the binary (0)-(1) reward in Eq. (4) considered optimal in this framework? In particular, why is it sufficient to use this terminal reward formulation, rather than an undiscounted sum of per-step rewards over the trajectory?*
>
> This is due to the particular structure of the problem, where our inherent goal is to increase the accuracy of generating a correct response, and defining the process reward is simply a tool to achieve this. Intuitively, our analysis depends on the fact that at each step, the policy can either pick a correct action, in which case it needs to receive a positive reward and some gradient signal, or an incorrect action which must yield zero reward and no gradient update. The value of this positive reward itself is not important, and thus can be chosen to be 1 instead of the sum of future returns. This is in contrast with the broader multi-turn RL setups where maximizing the reward is the inherent goal, in which case one needs to use the entire return from the action.
>
> We remark that an interesting open question is whether adding a baseline to the calculation of per-action reward (and thus having gradient update on incorrect actions) can improve the convergence rate. We think that the answer is likely negative, since this is not the case in the outcome reward setting where we have precise lower bounds.
>
>
>
> We would be happy to clarify any questions that may arise during the discussion period.
>
> ---
>
> F. Chen et al. “The coverage principle: How pre-training enables post-training.” ICLR 2026.
>
> D. Foster et al. “Is a good foundation necessary for efficient reinforcement learning? The computational role of the base model in exploration.” COLT 2025.
>
> D. Guo et al. “DeepSeek-R1: Incentivizing Reasoning Capability in LLMs via Reinforcement Learning.” Nature 2025.

---

> > ### Author Rebuttal · Reviewer_o3tZ · 2026-04-02
> >
> > Thank you for detailed explanation. My concerns have been addressed and I maintain my positive score.

---

### Official Review · Reviewer_Au5Q · 2026-03-13

**Soundness:** 4
**Presentation:** 4
**Significance:** 4
**Originality:** 4
**Overall Recommendation:** 5
**Confidence:** 3

**Summary:**

This paper provides theoretical results on how the base model affects the post-trained models.  Specifically, this paper studies policy gradient (PG) algorithms for post-training an autoregressive linear model under the separability assumption.
The paper proves that with an outcome reward model, PG can efficiently increase the likelihood of samples inside the support of the base model. To go outside the support of the base model and achieve a significantly smaller test error, the number of reward queries depends on the proposed likelihood quantile (LQ) and can be exponential in the sequence length. This issue can be alleviated using a process reward model, where the authors introduce a token-level likelihood quantile and show that it scales more favorably with sequence length. The authors also prove lower bounds to show the tightness of the results.

**Compliance With Llm Reviewing Policy:**

Affirmed.

**Key Questions For Authors:**

1. Regarding Weakness 1 on the assumptions, could you please give more discussion on the assumptions, when they are realistic, and to what extent they can be relaxed?
2. Could you please provide more discussion on the relationship of the proposed likelihood quantile to existing coverage notions?

**Limitations:**

yes

**Strengths And Weaknesses:**

Strengths:
1. This paper theoretically shows how the base model affects the post-trained model, which explains the empirical findings, and shows that learning good process models could help move beyond the base model.
2. This paper is well-written and technically sound.

Weaknesses:
1. There are many assumptions in the paper that are restrictive and not realistic in the real LLM settings:
(1) This paper studies autoregressive linear models with frozen feature representations, where only the final linear layer is trained
(2) The separability (margin) assumption is strong
But this is OK to provide formal theoretical results.
2. The experiment evaluation is limited to synthetic experiments. It would be better to provide more experimental results.

---

> ### Author Rebuttal · Authors · 2026-03-31
>
> We thank the reviewer for their time and their detailed evaluation of our work. We address their questions and concerns below.
>
> ---
> > *There are many assumptions in the paper that are restrictive ... Regarding Weakness 1 on the assumptions, could you please give more discussion ...*
>
> We would like to highlight the role of each core assumption and provide certain relaxations below.
>
> 1. The correct next token is $\gamma$-separated from any incorrect next token using the feature map $\phi$ on partially correct tokens so far. This assumption is seemingly strong but **not critical**, as we can relax it as follows.
>
>
> >   **Assumption:** For every $\varepsilon > 0$, there exists $\boldsymbol{w}$ such that $\Vert \boldsymbol{w} \Vert \leq \mathcal{R}(\varepsilon)$ and $\mathbb{E}[-\log p\_{\boldsymbol{w}}(\boldsymbol{y}^\*(\boldsymbol{x}) | \boldsymbol{x})] \leq \varepsilon$.
>
>
> With this new condition, we can replace $1/(\gamma^2 \varepsilon)$ in all upper bound statements with $\mathcal{R}(\varepsilon)^2/\varepsilon$. Note that the overall message, including the definition of LQ, remains unchanged.
>
> 2. The feature map $\phi$ is frozen. Such a condition is currently inevitable for a tractable analysis of post-training dynamics; see e.g. prior works Chen et al. (2025), Foster et al. (2025) where they point out that empirically, post-training might stay in the lazy regime where only last layer features evolve (Malladi et al., 2025).
>
> 3. Each $\boldsymbol{x}$ is associated with a unique correct response $\boldsymbol{y}^\*(\boldsymbol{x})$. We agree that extending this to a setting where we only have a subset $\mathcal{Y}^\*(\boldsymbol{x})$ of correct responses per input $\boldsymbol{x}$ is a very interesting future direction. A priori, it is not clear which correct solution will be preferred by the model after post-training. We expect that redefining the notion of Likelihood Quantile, where likelihood refers to the probability of generating any correct response, would be useful in that setting. We will highlight this as an open problem in the final version.
>
>
>
>
> ---
> > *The experiment evaluation is limited to synthetic experiments. It would be better to provide more experimental results.*
>
> We thank the reviewer for their suggestion. To provide intuitions beyond the synthetic task above, we have conducted additional experiments, and we will include figures illustrating the Likelihood Quantile function of *Qwen3-8B on the MATH500 dataset*.
>
> We have also carried out additional simulation studies where we compare the *evolution of the average likelihood of initially off-support samples* between *PG with process rewards* and *PG with outcome rewards*, demonstrating how PG-PR is able to go beyond the base model support, unlike PG-OR.
>
> These additional experiments will be included in the final version.
>
>
> ---
> > *Could you please provide more discussion on the relationship of the proposed likelihood quantile to existing coverage notions?*
>
> We thank the reviewer for this suggestion; this is indeed a valuable addition to the paper. While we touch upon this relationship after introducing Likelihood Quantile in line 260, we will elaborate further with the following.
>
> We characterize the relationship of LQ with three variants of *coverage* considered in prior work. The Coverage Profile used by Chen et al. (2025) is given by
> $$\operatorname{Cov}\_N(q^\* \Vert q) = \mathbb{P}\_{x,y \sim q^\*} [\frac{q^\*(y | x)}{q(y|x)} \geq N],$$
> where $q^*$ is the ground-truth distribution. This notion is directly related to the model’s Best-of-$N$ performance.
>
> Other popular variants of coverage include the following $L\_2$ and $L\_\infty$ notions (see e.g. Huang et al., (2025)):
> $$\operatorname{Cov}\_{L\_2}(q^\* \Vert q) = \mathbb{E}\_{x,y \sim q^\*}[\frac{q^\*(y|x)}{q(y|x)}], \quad \operatorname{Cov}\_{L\_\infty}(q^* \Vert q) =\sup\_{x,y} \frac{q^*(y|x)}{q(y|x)}.$$
>
> In our setting, $q^\*(\cdot | x)$ is a point mass at $y^\*(x)$ and using standard inequalities we have the following for all $\alpha \in (0,1)$ and $\varepsilon \in (0,1)$:
> $$\mathcal{Q}\_{q}^{-1}(\alpha) = \operatorname{Cov}_{1/\alpha}(q^\* \Vert q), \quad \mathcal{Q}\_q(\varepsilon) \geq \frac{\varepsilon}{\operatorname{Cov}\_{L\_2}(q^* \Vert q)}, \quad \mathcal{Q}\_q(0) \geq \frac{1}{\operatorname{Cov}\_{L\_\infty}(q^* \Vert q)}.$$
>
>
>
> We would be happy to clarify any questions that may arise during the discussion period.
>
>
> ---
>
> F. Chen et al. “The coverage principle: How pre-training enables post-training.” ICLR 2026.
>
> D. Foster et al. “Is a good foundation necessary for efficient reinforcement learning? The computational role of the base model in exploration.” COLT 2025.
>
> S. Malladi et al. “A kernel-based view of language model fine-tuning.” ICLR 2023.
>
> A. Huang et al. "Is Best-of-N the Best of Them? Coverage, Scaling, and Optimality in Inference-Time Alignment." ICML 2025.

---

> > ### Author Rebuttal · Reviewer_Au5Q · 2026-04-03
> >
> > Thanks for your response. I will maintain my positive score.

---

### Official Review · Reviewer_wHcK · 2026-03-14

**Soundness:** 2
**Presentation:** 2
**Significance:** 2
**Originality:** 3
**Overall Recommendation:** 4
**Confidence:** 3

**Summary:**

This paper gives a theoretical analysis of post-training linear autoregressive models under a token-level margin assumption. Its main question is when policy-gradient-style RL can genuinely improve a pretrained base model, versus merely sharpening probability mass that was already present. To formalize that dependence on the base model, the paper introduces the Likelihood Quantile (LQ) for whole-sequence success and a token-level analogue for process rewards. The positive results say that PG can efficiently boost performance on samples where the base model already assigns nontrivial likelihood, while the negative results argue that, with outcome rewards only, improving substantially beyond the base model can require exponentially many reward queries in sequence length in the worst case. The paper then argues that process rewards can avoid that exponential dependence by reducing the problem to token-level coverage, and it complements these upper bounds with lower bounds intended to show near-minimax optimality. The experiments are small synthetic illustrations of these ideas, showing slow or no improvement on very low-likelihood samples and the evolution of the likelihood-quantile curves during training.

**Compliance With Llm Reviewing Policy:**

Affirmed.

**Key Questions For Authors:**

- Do you intend the feature boundedness assumption to hold for **all** token-prefix-action triples, i.e. $\sup_{x,\text{prefix},a}\|\phi(x,\text{prefix},a)\|_2 \le R$? If so, can you restate all positive theorems with that assumption explicitly?

- In Remark 3.3, is the learner allowed to make a separate exploratory query from $q_t$ while evaluating mistakes under $p_{w_t}$, or should the statement instead be interpreted as an offline/statistical result?

**Strengths And Weaknesses:**

Strength:
1. The paper isolates an important issue in post-training: when policy-gradient-style RL can actually improve a model beyond what is already present in the base distribution.

2. The likelihood quantile viewpoint is a nice way to formalize how improvement depends on base-model coverage, both at the sequence level and token level. The paper does a nice job contrasting what is possible under good base support with hardness when success events have exponentially small base probability. The token-level/process-reward analysis gives an intuitive and theoretically motivated explanation for why denser supervision can avoid the worst-case sequence-level barrier.

Weaknesses:
- Altough this is a theory paper, many assumptions are too strong and not aligned with practical scenarios. For example, the reward functions enforces strict string matching, which is not practical because in real world llm tasks there are many functionally equivalent responses such as different code implementations. Furthermore, the main positive theory currently rests on an unstated stronger assumption: The stated assumption only bounds the norm of correct-path features where the proofs of the upper bounds require a bound over all prefixes/actions.

---

> ### Author Rebuttal · Authors · 2026-03-31
>
> We thank the reviewer for their time and their detailed evaluation of our work. We address their questions and concerns below.
>
> ---
>
> > *Although this is a theory paper, many assumptions are too strong and not aligned with practical scenarios. For example, the reward function enforces strict string matching, …*
>
> We would like to highlight the role of each core assumption and provide certain relaxations below.
>
> 1. The correct next token is $\gamma$-separated from any incorrect next token using the feature map $\phi$ on partially correct tokens so far. This assumption is seemingly strong but **not critical**, as we can relax it as follows.
>
>
> >   **Assumption:** For every $\varepsilon > 0$, there exists $\boldsymbol{w}$ such that $\Vert \boldsymbol{w} \Vert \leq \mathcal{R}(\varepsilon)$ and $\mathbb{E}[-\log p\_{\boldsymbol{w}}(\boldsymbol{y}^\*(\boldsymbol{x}) | \boldsymbol{x})] \leq \varepsilon$.
>
>
> With this new condition, we can replace $1/(\gamma^2 \varepsilon)$ in all upper bound statements with $\mathcal{R}(\varepsilon)^2/\varepsilon$. Note that the overall message, including the definition of LQ, remains unchanged.
>
> 2. The feature map $\phi$ is frozen. Such a condition is currently inevitable for a tractable analysis of post-training dynamics; see e.g. prior works Chen et al. (2025), Foster et al. (2025) where they point out that empirically, post-training might stay in the lazy regime where only last layer features evolve (Malladi et al., 2025).
>
> 3. Each $\boldsymbol{x}$ is associated with a unique correct response $\boldsymbol{y}^\*(\boldsymbol{x})$. We agree that extending this to a setting where we only have a subset $\mathcal{Y}^\*(\boldsymbol{x})$ of correct responses per input $\boldsymbol{x}$ is a very interesting future direction. A priori, it is not clear which correct solution will be preferred by the model after post-training. We expect that redefining the notion of Likelihood Quantile, where likelihood refers to the probability of generating any correct response, would be useful in that setting. We will highlight this as an open problem in the final version.
>
>
>
> ---
> > *Do you intend the feature boundedness assumption to hold for all token-prefix-action triples?*
>
> Yes. We thank the reviewer for pointing this out. We will fix Assumption 2.1 in the final version to reflect the requirement $\Vert \phi(\boldsymbol{x},\boldsymbol{y}_{1:i}(\boldsymbol{x})) \Vert \leq 1$. We remark that this assumption is often satisfied in practice thanks to normalization layers in the case of deep networks and bounded input features in the case of shallow ones.
>
> ---
> > *In Remark 3.3, is the learner allowed to make a separate exploratory query from $q_t$ while evaluating mistakes under $p_{w_t}$, or should the statement instead be interpreted as an offline/statistical result?*
>
> This is in fact an important aspect that we need to clarify in the paper, and we thank the reviewer again for bringing this to our attention. At iteration $t$, the online learner will draw a sample response from $q\_t$ to update its parameters. However, it will draw a separate response from $p\_{\boldsymbol{w}\_t}$ to make a prediction about the current prompt. This second response is only used to make predictions or to measure the quality of the mode, not to update the weights.
>
>
> We would be happy to clarify any questions that may arise during the discussion period.
>
> ---
> References:
>
> F. Chen et al. “The coverage principle: How pre-training enables post-training.” ICLR 2026.
>
> D. Foster et al. “Is a good foundation necessary for efficient reinforcement learning? The computational role of the base model in exploration.” COLT 2025.
>
> S. Malladi et al. “A kernel-based view of language model fine-tuning.” ICLR 2023.

---

> > ### Author Rebuttal · Reviewer_wHcK · 2026-04-07
> >
> > Thank you for the response. The questions are addressed, so I'd like to keep my positive score.

---

### Decision · Program_Chairs · 2026-04-30

**Decision:**

Accept (spotlight)

**Comment:**

All reviewers agree this is an interesting and sound paper.  While some reviewers quibble with the restrictions imposed by the assumptions, the authors convincingly rebut this point by (a) pointing to recent literature making similar assumptions and (b) pointing out that some assumptions have to be made in order to make the problem analytically tractable.